# Deep phenotyping of skin tissue remodeling in patients with systemic sclerosis treated with CD19-CAR T cells

Aleix Rius Rigau[1,2,16], Meilin Xu[3,4,16], Ziyuan Liu[1,2,16], Sara Chenguiti Fakhouri[1,2,16], Janina Auth[1,2], Panagiotis Garantziotis[1,2], Andrea Zoli[1,2,5], Manoj Kumar Selvaraju[6], Maria Gabriella Raimondo [1,2], Carlo Tur [1,2,5], Tim Filla[3,4], Paula Gehringer[1,2], Markus Eckstein [7], Fabian Müller [2,8], Armin Atzinger[9], Moritz Ronicke[2,10], Arif Ekici [11], Rafael Schmid[12], Andreas Wirsching[1,2], Melanie Hagen[1,2], Sebastian Böltz [1,2], Tobias Krickau[13], Raymund E. Horch [12], Carola Berking [2,10], Ricardo Grieshaber-Bouyer [1,2], Andreas Ramming [1,2], Pooja Gupta [14], Aline Bozec [1,2], Andreas Mackensen [2,8], Jörg HW Distler [3,4,15], Georg Schett [1,2], Yi-Nan Li [3,4,15,17] & Christina Bergmann [1,2,17] ✉

Systemic sclerosis (SSc) is an autoimmune disease characterized by vasculopathy and fibrotic remodeling of the skin and internal organs. Fibrotic tissue changes are considered hardly reversible with current therapies, suggesting that new strategies are required to modulate the disease-associated molecular and cellular phenotype to enable regeneration of affected tissues. Here, analyzing skin biopsy samples from patients with SSc who had received CD19-CAR T cell therapy as part of the CASTLE study or named patient use, we demonstrate structural regeneration of SSc skin structure, as evidenced by recovery of skin papillae. Consistent with these histological changes, cyclic in situ hybridization and imaging mass cytometry analyses suggested that fibroblast populations shifted towards a physiological state, both in terms of composition and function. Moreover, we describe signs of vascular repair and changes in epidermal cell function. These results suggest that B cell depletion using CD19-CAR T cell therapy may lead to skin tissue remodeling in SSc and highlight its potential for tissue regeneration in fibrotic diseases.

Systemic Sclerosis (SSc) is an autoimmune fibrotic disease characterized by the highest case-related mortality among connective tissue diseases[1–3]. Autoimmunity, vasculopathy, and fibrotic tissue remodeling are cardinal features of SSc that interact and potentially amplify each other to drive disease progression and organ dysfunction. Current pharmaceutical therapies may at best slow down disease progression but hardly halt it or even induce regeneration of the affected tissues. In the early stages of SSc, autoimmunity prevails with the occurrence of specific autoantibodies[4,5] and vasculopathy manifests as

Raynaud's phenomenon and digital ulcerations[1]. In the course of the disease, excessive fibroblast activation results in progressive fibrotic remodeling of the skin, lungs, and heart.

Skin fibrosis manifests in almost all patients with SSc, starting from the distal and extending to the proximal extremities in the more severe diffuse-cutaneous form of SSc (dcSSc). Extensive skin fibrosis is associated with an increased risk of fibrotic remodeling of inner organs[1,6]. Physiologically, the skin is composed of a polarized, layered structure including the epidermis, the papillary and reticular dermis,

---

and the subcutaneous tissue[7]. Skin papillae are complex structures at the dermal-epidermal interface that are limited by the epidermal rete ridges and shape the skin surface[8]. Dermal papillae contain capillaries and nerve endings and are highly functional, as they mediate mechanical signals supporting sensory and haptic functions. They also promote structural stability and supply the epidermis[7,9]. Of note, they play a central role in regenerative as opposed to reparative (scarring) healing of the skin after injury: hence, fibroblasts extracted from the papillary dermis promote the organization of a normal stratified epidermis in a three-dimensional organ culture model more efficiently than fibroblasts from the reticular dermis[10–12].

In SSc, the papillary dermal structures disappear with progression of the disease, reflecting both vascular loss and fibrotic tissue remodeling, as the skin transitions to a "reticularized" phenotype[13,14] with accumulation and increased alignment of collagen fibers throughout the dermis[14–16]. These skin changes and papillary loss as characteristic hallmarks of SSc skin have been considered permanent for a long time. Studies performed on skin biopsies in patients who had received autologous stem-cell transplantation (ASCT) have shown that broad immunosuppression can reduce the extent of skin fibrosis, as assessed by the semi-quantitative normalized fibrosis score[17], and obtain improvements of capillaroscopy results[18]. However, whether the reversal of the dermal phenotype and recovery of the dermal papillae are possible remains unknown.

We hypothesize that B cell driven autoimmunity may be a key driver of cutaneous remodeling in SSc, as growing evidence implicates B cells in orchestrating fibrotic tissue remodeling in SSc. Dysregulation of circulating B cells is evident at several levels in SSc, including aberrant B cell receptor (BCR) signaling characterized by enhanced activating and diminished inhibitory signals[19–21] and disturbed effector function resulting in hypergammaglobulinemia and autoantibody secretion. Distinct autoantibodies are associated with defined clinical SSc phenotypes[2,22], e.g., anti-scl70 antibodies are associated with a high risk of disease progression and inner organ involvement. Other autoantibodies exert direct pathogenic effects: agonistic antibodies targeting anti-platelet-derived growth factor receptor (PDGFR) or fibrillin promote fibroblast activation[23,24], while anti-endothelial cell antibodies, anti-endothelin−1-type A receptor, and anti-angiotensin-II type 1 receptor antibodies contribute to endothelial cell damage and vasculopathy[20,25,26]. In addition to autoantibody-mediated mechanisms, B cells shape the fibrotic milieu through the secretion of pro-fibrotic and pro-inflammatory cytokines such as IL6[27,28], whereas regulatory B cell subsets that restrain inflammation, e.g., via IL10, are decreased in SSc[29]. B cells also infiltrate the organs affected in SSc and may mediate tissue remodeling[20,30]: A recent exploratory study using a three-dimensional humanized skin model showed that infiltrating B cells of SSc patients show a hyperactivated phenotype, increased immunoglobulin production and the capacity to induce profibrotic transcriptomic programs in resident fibroblasts, supporting a direct role for B cells in tissue-level disease propagation[31].

The rationale of targeting B cells in SSc is further supported by clinical trials of the CD20-depleting antibody rituximab (RTX), which have shown improvements in both skin fibrosis and interstitial lung disease (ILD)[32–34] and exploratory analyses suggest that clinical response may be linked to dermal remodeling of pathways such as the upregulation of dermal DKK-1[35]. However, despite encouraging results, many patients still progress under conventional B cell–targeted therapy, highlighting a critical limitation of current B cell-directed therapies. CD20-targeting antibodies incompletely deplete tissue-resident B cells[36–38] and fail to eliminate antibody-producing plasmablasts. While ASCT may overcome this limitation, it carries significant procedure-related risk. We recently demonstrated that CD19-directed chimeric antigen receptor (CAR) T cells achieve deep and broad B cell depletion across tissues[36,37]. Consistent with this, we and others have reported

profound clinical responses in patients with SSc treated with CD19-CAR T cell therapy[39–41]. Based on these lines of evidence, we hypothesize that depletion of B cells by CAR T cell therapy may interfere with the B cell mediated fibrotic and vascular remodeling, thereby reshaping diseased tissue architecture in SSc.

The aim of the present study was to provide a proof of concept for skin tissue changes on the histological, molecular and cellular level before and after CD19-CAR T cell therapy by using RNA sequencing, imaging mass cytometry (IMC) and cyclic in situ hybridization (cISH) (Supplementary Fig. 1).

## Results

### Signs of regeneration in SSc skin upon CD19-CAR T cell therapy
Sequential skin biopsies were performed at the forearm of patients with dcSSc before and one, six and twelve months after infusion of CD19-CAR T cells. Skin biopsies of patients who received standard-of-care treatments (standard of care group, SOC) and a patient with stable dcSSc without any specific treatment (natural disease course group, NDG) were investigated as exploratory controls. The baseline characteristics of all patients are summarized in Supplementary table 1 and Supplementary Fig. 2A. The course of peripheral CAR T cell expansion is visualized in Supplementary Fig. 1C. In line with previous studies, peripheral B cells were completely depleted within 9 days in all patients after the application of CD19-CAR T cells. Their reconstitution started between d62 and d253 (Supplementary Fig. 2B). Interestingly, while B cells counts were heterogenous at baseline, no CD19 + B cells or CD20 + B cells were detected in any of the available follow up skin samples (Supplementary Fig. 3). The serum-titers of anti-Scl70 antibodies declined throughout the observation (Supplementary Fig. 2D). To investigate the effects of CD19-CAR T cell therapy on the skin structure, we first analyzed effects of CD19-CAR T cell therapy on the dermal papillae (DP) on histologic skin sections (Fig. 1A): The frequency of DP significantly increased, which was not observed during follow-up examinations SOC group and samples of a NDG patient. In addition, DP height increased by up to 2.4-fold (12 mo, patient 2). Recovery of the skin wrinkles upon CD19-CAR T cell therapy as indirect readout for rete ridges is exemplified in Fig. 1B[8]. In line with the regression of fibrotic skin remodeling, the alignment of collagen fibers[16] decreased upon CD19-CAR T cell therapy, but not with standard-of-care treatments (Fig. 1A). No significant sex-related differences were detected in dermal papillary measurements (data shown in the source data file). Therefore, given the exploratory sample size, data from both sexes were aggregated for all further analyses. The changes of the modified Rodnan Skin score (mRSS), which was documented in parallel with the skin biopsies in each group, is visualized in Fig. 1C and in Supplementary Data 3.

To assess how CD19-CAR T cell therapy affects transcriptomic patterns of the skin, we first performed bulk RNA sequencing (bulk-RNA-Seq). Using gene set enrichment analysis, we observed the downregulation of gene signatures associated with B cell activation and CD22-mediated B cell regulation, as expected upon CD19-CAR T cell treatment (Fig. 1D). In line with regeneration of the skin, we observed an increased enrichment of gene signatures associated with the recovery of vasculopathy (e.g., "angiogenesis", "vasculature development") with extracellular matrix remodeling (e.g., "extracellular matrix structure organization" and "wound healing") and terms related to epidermis remodeling (e.g., "skin epidermis development" and "epithelial cell differentiation").

We next characterized the transcriptional changes of single cells upon CD19-CAR T cell therapy by cISH using Xenium Prime-5k spatial transcriptomics assay. We recovered a total of 74 636 cells; at 1-month follow-up: 25 178 cells; 6-month follow-up: 13 618 cells; 12-month follow-up: 12 053 cells, with their spatial localization at subcellular resolution and transcriptomic profiles (Supplementary Fig. 4). We first

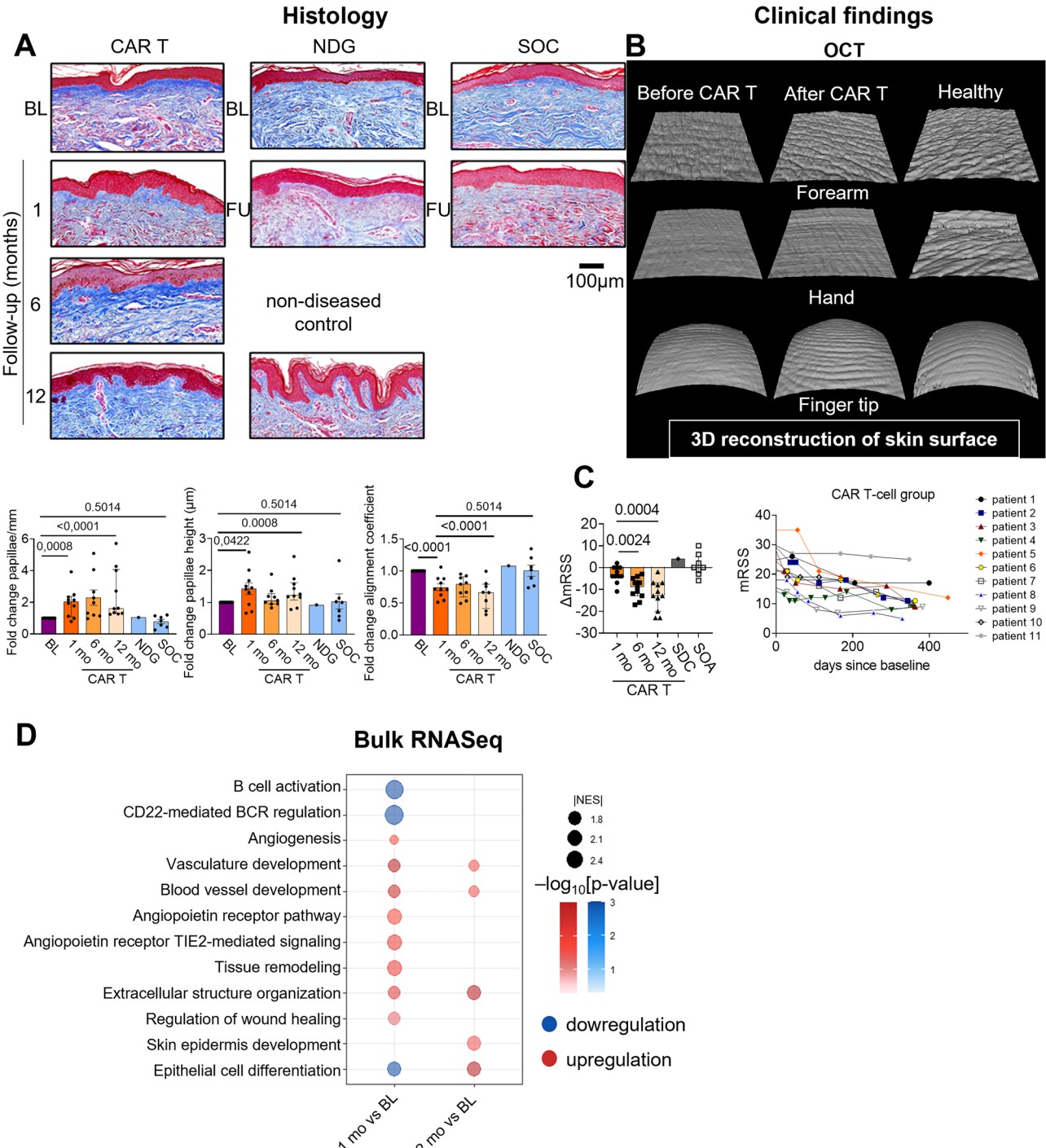

**Fig. 1 | Effects of CD19-CAR T cell therapy on the skin microanatomy and skin vessels in dcSSc compared to the natural disease course in patients with previous immunosuppression and upon standard-of-care treatments.**
**A** Representative Masson-Trichrome staining images (CAR T patient 5, NDG patient 1 and SOC patient 4). Papillae quantifications (papillae/mm and papillary height) per section as well as collagen alignment coefficient are shown as bar graphs. The data are presented as median with interquartile range (IQR). *p*- values < 0.0125 were considered significant after Bonferroni correction. Baseline (*n* = 18), CAR T: 1-month follow-up (*n* = 10), CAR T: 6-month follow-up (*n* = 9), CAR T: 12-month follow-up (*n* = 10), natural disease course group: follow up (*n* = 1) and standard-of-care treatments: follow up (*n* = 7). **B** Representative images of 3D reconstruction of the skin surface of different body areas as captured with dynamic optical coherence tomography before and 2 months after CAR T cell therapy (patient 11). **C** mRSS change from baseline to the respective follow-up in the different treatment groups.

Median mRSS improvement after CAR T cell therapy is shown at 1 month (*n* = 11), 6 months (*n* = 11) and 12 months (*n* = 11). Median follow-up time in the NDG group (*n* = 1) was 619 days after baseline and in the SOC group (*n* = 8) 240 days after baseline. The data are presented as median with IQR. **D** Dot plot showing the enriched pathways detected by gene set enrichment analysis (GSEA) of RNASeq data from skin biopsies collected at 1 month (*n* = 5) or 12 months (*n* = 3) after CAR T cell therapy, compared to baseline (*n* = 6). Only pathways with an adjusted *p*-value < 0.05 are plotted. The size of the dots indicates the absolute value of normalized enrichment score. The colors represent the direction of regulation (red, upregulated; blue, downregulated in post-treatment samples). The color intensity denotes the adjusted *p*- value using the Benjamini-Hochberg procedure. Ctrl control; FU Follow-up; BL baseline; mo month; mRSS modified Rodnan Skin score; NDG natural disease course group; SOC standard of care; NES normalized enrichment score. 1 A-C Mann-Whitney nonparametric t-test was used.

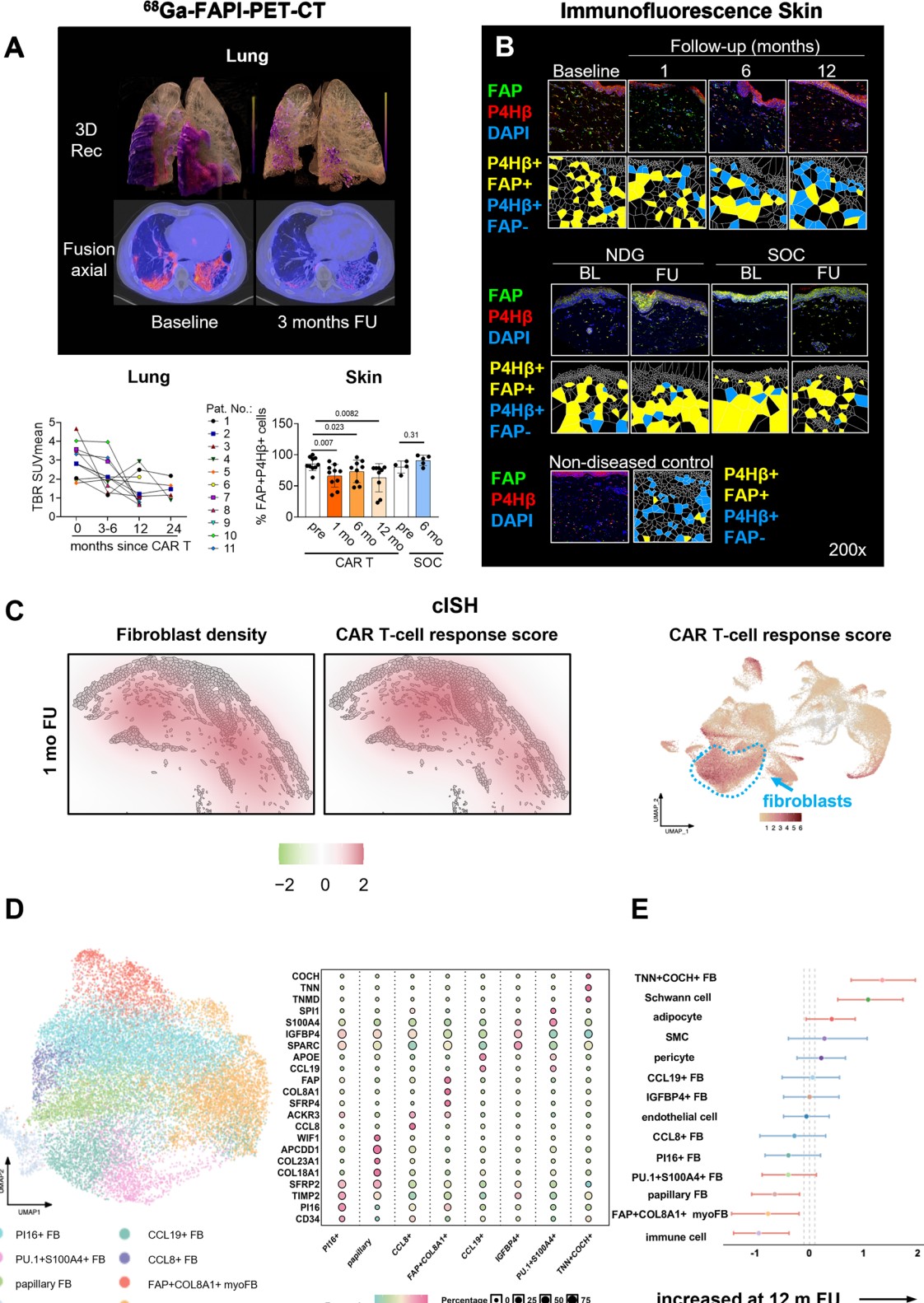

**A** $^{68}$Ga-FAPI-PET-CT

**B** Immunofluorescence Skin

**C** cISH

**D**

**E** increased at 12 m FU →

annotated major cell types in the skin tissue based on known marker gene expression (Supplementary Fig. 4A and Supplementary Fig. 4B[42]). To identify the major changes triggered by CD19-CAR T cell therapy, we computed a CAR T response score using the differentially expressed genes (DEG) obtained from bulk RNA-Seq (1 month after CAR T compared to baseline) and applied it to individual cell types. We

observed high CAR T cell response scores in different cell types including vascular cells and epithelial cells (Supplementary Fig. 4C) as well as particularly high response scores in fibroblasts (Fig. 2C and Supplementary Data 1). Based on our findings, we next focused on the functional characterization and phenotyping of fibroblasts before and after treatment with CD19-CAR T cells.

**Fig. 2 | Effects of CD19-CAR T cell treatment on fibroblast activation in vivo.**
**A** Representative fusion images of the FAPI PET scan and the corresponding CT scan before and 3 months after CD19-CAR T cell therapy (patient 4). 3D-reconstruction was kindly provided by Dr. Engel (Siemens Healthineers). Quantification of FAPI uptake is visualized as Spaghetti plot (before CAR T: $n = 11$, 3-6 mo-FU $n = 11$, 12 mo-FU: $n = 7$, 24 mo-FU $n = 5$). **B** Immunofluorescence for fibroblast activation protein (FAP) and the fibroblast marker prolyl-4-hydroxylase β (P4Hβ) and DAPI (CAR T patient 2, NDG patient 1, SOC patient 4). Quantification of FAP+ fibroblasts as percent of total fibroblasts shown as bar graph with median andIQR. Before CAR T ($n = 10$), 1 mo-FU ($n = 10$), 6 mo-FU ($n = 9$), 12 mo-FU ($n = 9$), before start of standard-of-care therapy ($n = 4$), FU SOC ($n = 5$). **A,B**: Mann-Whitney test was used,$p$- values < 0.025 were considered significant after Bonferroni-correction. Representative images and voronoi tesselations are shown. **C** Kernel density maps showing the spatial distribution of fibroblast density and CAR T cell response score in the skin (patient 9 at 1-mo FU). Color scales indicate relative densityfrom low

(green) to high (red). UMAP plot displays CAR T cell response scores, based on the same UMAP shown in Supplementary Fig. 3A. Yellow color represents low and red high CAR T cell response score. **D** UMAP plot showing fibroblast subpopulations detected by cISH,. Dot plot characterizes eight FB subpopulations based on er known marker genes. Dot size: proportion of expressing cells,dot color: te average expression levels. BL $n = 7$, 1-month FU $n = 8$, 6-month FU $n = 4$, 12-month FU $n = 4$. FAP + COL8A1+FBcorrespond to myofibroblasts. **E** Forest plot showing differential composition of non-epithelial component between baseline ($n = 7$) and 12-mo FU ($n = 4$) by a compositional regression model with donor-specific random intercepts using sccomp. Points represent the posterior mean estimates with error bars indicating 95% of credible intervals (2.5%–97.5%). Red bars indicate significant difference in cell proportion (FDR < 0.05)Dashed line: default threshold for the minimum effect size of ±0.1. FU Follow-up; BL baseline; mo month; NDG natural disease course group; Rec Reconstruction; SOC standard of care; NES normalized enrichment score. FB Fibroblast; FDR False discovery rate.

## Effects of CD19-CAR T cell therapy on fibroblast function and phenotype

Using positron emission tomography (PET) with a fibroblast activation protein inhibitor (FAPI) tracer[43,44], we evaluated the effect of CD19-CAR T cell therapy on fibroblast activation in vivo. We observed a rapid decrease in FAPI uptake in the lungs in the temporal context with CD19-CAR T cell therapy (Fig. 2A and Supplementary Fig. 5) indicating substantial reduction of fibroblast activation. Lung function parameters stabilized or improved during the observational period (Supplementary Fig. 4D). Quantification of FAPI uptake on the skin tissue is technically not possible due to the relatively thin skin layer and the signal detection threshold of the scanner. However, consistent with the in vivo FAP imaging of the lungs, we observed a reduction of FAP+ fibroblasts in skin biopsies after CD19-CAR T cell therapy as assessed by co-immunofluorescence staining (Fig. 2B).

When mapping the CAR T cell response score across the skin, CAR T cell responses were particularly prominent in regions with higher fibroblast density (Fig. 2C). In line with these findings and with dermal remodeling described in Fig. 1, we observed an enrichment of gene sets related to "extracellular matrix regulation"[45], "extracellular matrix degradation" and "extracellular matrix elasticity" when comparing the RNASeq data of one and twelve months after CD19-CAR T cell treatment to baseline (Supplementary Fig. 6). Gene signatures that relate to papillary and reticular fibroblasts in non-diseased skin have recently been described[10,46,47]. Interestingly, we observed significant enrichment of papillary and reticular gene patterns in bulk Seq-derived DEG upon CD19-CAR T cell treatment, suggesting remodeling towards the physiological state (Supplementary Fig. 7). Next, we focused on the detailed phenotyping of fibroblast populations using cISH and identified eight fibroblast subpopulations (Fig. 2D). Aligned with the spatial genomic atlas[42], the papillary fibroblasts (expressing *COL18A1* and *COL23A1*) and TNN + COCH+ fibroblasts are located right beneath the skin epidermal layer and adjacent to hair follicles, respectively, confirming their spatial localization (Supplementary Fig. 8). The expression profiles of papillary fibroblasts also resembled their reported phenotype (Fig. 2D)[48]. Similarly, the PI16+ fibroblasts exhibited a universal fibroblast progenitor phenotype associated with regenerative functions[49]. We also recovered other fibroblast populations of SSc skin described in previous studies, including CCL19 + [50], PU.1 + S100A4 + [51], IGFBP4 + [52], CCL8+ (expressing *ACKR3*[53,54];), and FAP + COL8A1 + , which co-expressed the myofibroblast marker SFRP4[55,56] (Fig. 2D). We further validated the cISH-identified fibroblast populations with the spatial genomics atlas by scANVI-based integration. The results indicated good correspondence of the cISH-identified population to the reported phenotype. For instance, papillary fibroblasts correspond to *F1: Superficial* in the atlas; CCL19+ and PU.1 + S100A4+ fibroblasts mapped to inflammatory population *F3: FRC-like*; PI16+ fibroblasts mapped to *F2: Universal* progenitor population; and FAP + COL8A1+

myofibroblasts to disease-specific myofibroblasts (Supplementary Fig. 9).

Next, we investigated the functional phenotypes of these fibroblast populations using fast gene set enrichment analysis (FGSEA). FGSEA revealed strong enrichment of fibrosis-related terms such as "collagen formation" and "extracellular matrix organization" in FAP + COL8A1+ and fibroblasts (Supplementary Fig. 10A). Terms related to immune response regulation including "T cell activation" and "IFN-γ response" were particularly prominent in CCL19 + , CCL8 + , and PU.1 + S100A4+ fibroblasts, confirming their previously reported proinflammatory functions. FAP + COL8A1+ myofibroblasts demonstrated the highest collagen expression and TGFβ activities among the fibroblast populations, indicating their pro-fibrotic phenotype (Supplementary Fig. 10B).

We next explored the changes of cell composition after CD19-CAR T cell therapy. We first examined the differences in dermal cell populations between samples obtained from 12-month follow-up and baseline, as a representation of long-term CD19-CAR T cell effect (Fig. 2E). We detected prominent decreases in pro-fibrotic FAP + COL8A1+ myofibroblasts, papillary fibroblasts, pro-inflammatory PU.1 + S100A4+ fibroblasts, and immune cells. We also found increases of TNN + COCH+ fibroblasts, Schwann cells, and adipocytes in samples from 12 months post-CAR T cell therapy compared to baseline. The increase of Schwann cells and TNN + COCH+ fibroblasts and the decrease of PU.1 + S100A4+ fibroblasts were detected between 6–12 months post-CD19-CAR T cell therapy (Supplementary Fig. 11A and 11B). We detected moderate increases of immune cells and CCL8+ fibroblasts within the first 6 months post-therapy, followed by a strong reduction in these immune and immune-modulatory populations.

Next, we focused on the analysis of the fibroblast composition in relation to the papillary and reticular dermal layer (Supplementary Fig. 12). We detected an increase of regenerative *PI16*-expressing fibroblasts in the papillary dermis consistent with changes towards a physiologic distribution[57]. In concordance with the IF staining, the reduction of FAP + COL8A1+ myofibroblasts was mainly detected in the reticular dermis. Downregulation of proinflammatory PU.1 + S100A4+ fibroblasts and upregulation of TNN + COCH+ fibroblasts were also found prominently in the reticular dermis. Additionally, we observed a reduction of CCL19+ fibroblasts and more abundant IGFBP4+ fibroblasts in the reticular dermis when comparing 12-month follow-up samples to baseline (Supplementary Fig. 12).

After describing changes in cellular compositions, we next investigated whether fibroblast populations change their functional phenotypes. Therefore, we performed differential gene expression analysis comparing 1-month or 12-month follow-up samples to baseline for each fibroblast subpopulation. FGSEA indicated a steady reduction in the enrichment score for "collagen formation" in FAP + COL8A1+ myofibroblasts following the time course after therapy (Figs. 3A, B). In

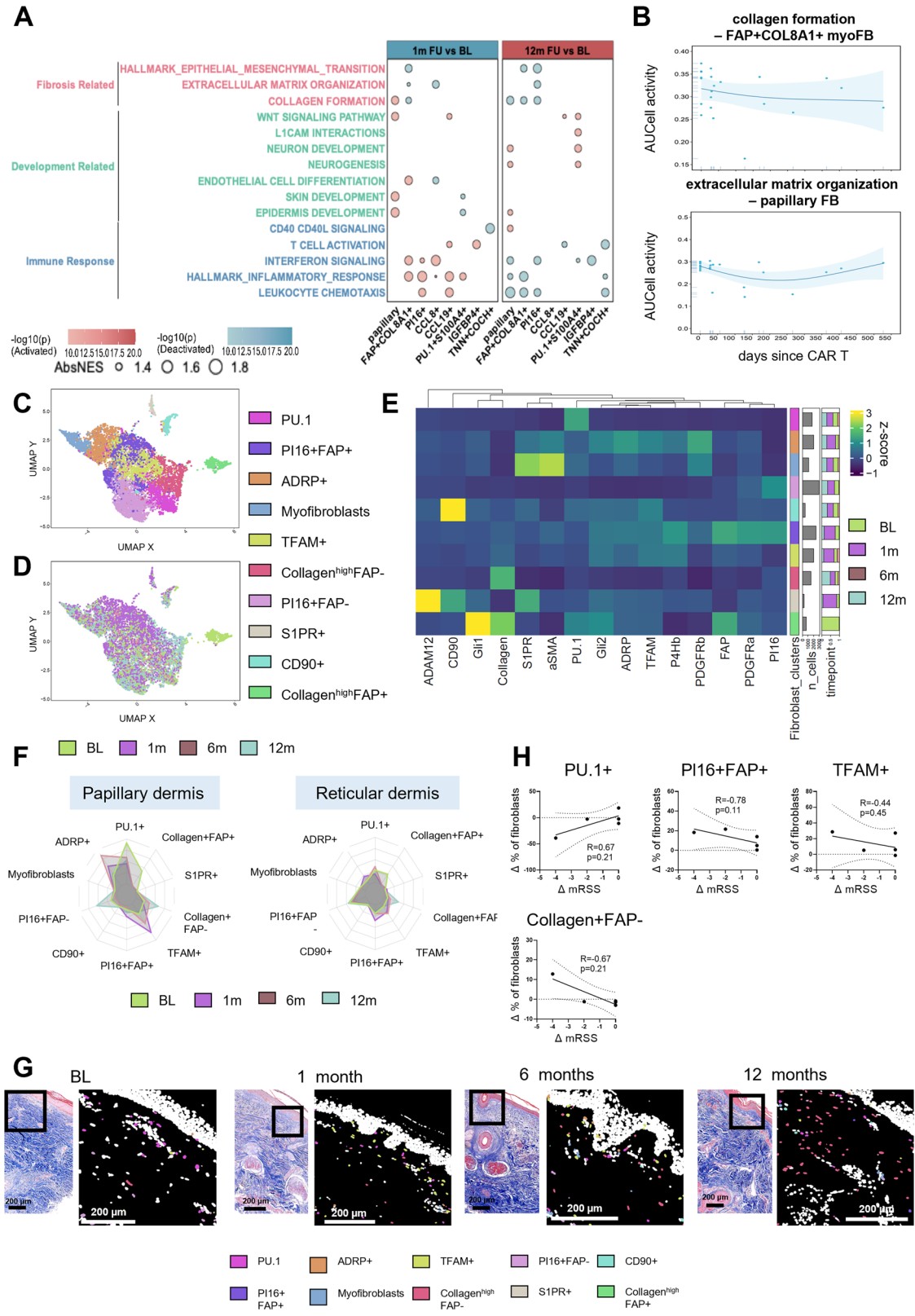

agreement with compositional analysis, we detected an upregulation of immune-related terms including "leukocyte chemotaxis", "inflammatory response" and "interferon signaling" early after CD19-CAR T cell treatment, which was no longer observed at the 12-month time point. Consistent with RNASeq data, papillary fibroblasts, which are located adjacent to the rete ridges, upregulate the functional term "epidermis development" after CD19-CAR T cell treatment (Fig. 3A).

To analyze the composition of fibroblast populations with another approach and on the protein level, we analyzed cellular changes on single cell resolution using imaging mass cytometry (IMC). We used a panel of antibodies designed to identify different fibroblast populations, which have been implicated in the pathogenesis of SSc[58], as well as markers of immune cells and vascular cells (Supplementary methods, table 1). A total of 53 853 cells were collected (baseline: 13 918

**Fig. 3 | Characterization of functional changes of fibroblast populations upon CD19-CAR T cell therapy using cISH and analysis of changes in fibroblast populations after CD19-CAR T cell therapy by imaging mass cytometry (IMC).** **A** Dot plots showing enriched pathways across fibroblast subtypes between BL and 12-month FU detected by FGSEA. Pathways of three functional categories—fibrosis-related, development-related, and immune response— are colored. Dot size represents the absolute value of normalized enrichment score (AbsNES), dot color reflects the log-transformed *p*-value. Red and blue dots represent positive and negative NES **B** Scatter plots show the dynamic changes in AUCell-derived pathway activities at the pseudo-bulk level for fibroblast subtypes. The x-axis represents the number of days after CAR T cell treatment. GAM (Generalized Additive Model) was used to fit smooth curves illustrating temporal trends, and the shaded band denotes the 95% confidence interval.**C**, **D** UMAP plots of fibroblasts of all patients characterized by IMC. The UMAP plots are colored by the identified cluster (**C**) or

the different time points (**D**) and randomly downsampled to 10.000 cells. **E** Heatmap represents the protein expression of the different clusters shown as z-score. The legend shows the number of cells as well as the % of each time point in each cluster, as indicated. **F** Radar chart showing the proportion of each fibroblast cluster of all stromal cells comparing the different time points in the papillary and reticular dermis. **G** Representative images of the IMC cell segmentation mask colored by the fibroblast clusters showing their spatial distribution and the trichrome histology staining on a consecutive cut, comparing baseline, 1 month, 6 months and 12 months after the CD19-CAR T cell infusion. The scale bars represent 200 μm. (baseline patient 6; 1 month patient 5; 6 months patient 5; 12 months patient 2). **H** Spearman's two-sided correlation on the mRSS with percentage of the selected fibroblast clusters on the whole dermis. All values represent the difference between 1 month and baseline (*n* = 5). p- and R-values are indicated.**C–H** baseline, *n* = 5; 1 month, *n* = 6; 6 months, *n* = 2; 12 months, *n* = 3. BL baseline.

cells; 1-month follow-up: 22 807 cells; 6-month follow-up: 3 985 cells; 12-month follow-up: 13 143 cells). The dataset is composed of 21 102 fibroblasts (defined as: E-cadherin⁻, CD31-, CD45-, SM22$^{low/-}$, SMA$^{low/-}$), 1 453 immune cells (E-cadherin-,CD31-,CD45 + ), 4 242 vascular cells (E-cadherin-,CD45-,CD31 + ), 18 781 epithelial cells (epidermis and hair follicle, defined as E-cadherin + ) and 385 vascular smooth muscle cells (E-cadherin-, CD31-, CD45-, SM22$^{high}$, SMA$^{high}$, Supplementary Fig. 13 and 14). As visualized by UMAP, distinct proteomic patterns of fibroblasts were found in skin samples from SSc patients before and after CD19-CAR T cell therapy (Figs. 3C, D). Using Phenograph, we identified fibroblast populations similar to previous studies[58]: PI16 + , FAP+ fibroblasts and PI16 + , FAP- fibroblasts, two fibroblast subsets with high levels of collagen protein expression (referred to as Collagen$^{high}$, FAP+ in contrast to Collagen$^{high}$, FAP- fibroblasts) and further five fibroblast clusters based on the prominent expression of PU.1[59], perilipin 2 (ADRP), mitochondrial transcription factor (TFAM)[58,60], sphingosine-1-phosphate receptor 1 + (S1PR) and Thy-1 cell surface antigen (THY1/CD90) (Fig. 3E). Myofibroblasts were defined as the fibroblast cluster expressing alpha smooth muscle actin (αSMA).

We analyzed how the composition of these fibroblast subsets changes upon CD19-CAR T cell therapy in the papillary versus the reticular dermis (Fig. 3F and Supplementary Fig. 15). Changes of fibroblast composition towards a more physiologic composition was observed in the skin and was particularly prominent in the papillary dermis (Figs. 3F, G, and Supplementary Fig. 15): the proportion of TFAM+ fibroblasts, which are associated with homeostatic functions and decreased in SSc[50,58], was increased in all samples, particularly in the papillary dermis by one month after CD19-CAR T cell treatment (Supplementary Fig. 15). Moreover, we observed a reduction of profibrotic PU.1+ fibroblasts in the papillary dermis, especially in patients with a high proportion of these cells at baseline (Supplementary Fig. 15). In addition, changes in the frequencies of other fibroblast populations were observed, which support the hypothesis of a regeneration of the physiological fibroblast composition: up-regulation of PI16 + FAP+ fibroblasts, down-regulation of S1PR+ fibroblasts in the papillary dermis and downregulation of myofibroblasts, particularly in the reticular dermis[56,58,61,62] (Supplementary Fig. 15).

To further support the hypothesis of a restoration of fibroblast composition to the physiological phenotype by CD19-CAR T cell therapy, we compared the dataset of fibroblasts obtained before and after CD19-CAR T cell treatment to a dataset obtained from non-diseased skin biopsies[58] stained with an antibody panel of 22 shared markers (Supplementary Fig. 16). Interestingly, the obtained composition of fibroblast clusters in the post-CAR T samples was comparable to the composition of fibroblast clusters in healthy controls.

Next, we analyzed the association of the quantitative changes of fibroblast subpopulations with clinical improvement of the skin (ΔmRSS) upon CD19-CAR T cell treatment (Fig. 3 H and Supplementary Fig. 17): the relative reduction of PU.1+ fibroblasts tended to be associated with the relative reduction in mRSS score ($R_S$ = 0.67, *p* = 0.22).

Moreover, the increase of PI16 + FAP+ fibroblasts ($R_S$ = −0.78; *p* = 0.12) and Collagen$^{high}$FAP- fibroblasts ($R_S$ = −0.67; *p* = 0.22) tended to associate with a reduction in mRSS score, consistent with dermal regeneration.

Of note, the protein markers used for identifying the fibroblast populations in IMC were also distinctly expressed in the populations detected by cISH on RNA levels (Supplementary Fig. 18). Both techniques recovered fibroblasts expressing PI16, FAP, CD90 (encoded by *THY1*), and PU.1 (encoded by *SPI1*), suggesting that these populations can be robustly recovered by both protein and RNA extraction methods. Moreover, the populations identified by both techniques consistently shifted towards a homeostatic profile upon CD19-CAR T cell therapy, such as decreasing Collagen$^{high}$FAP+ fibroblasts and increasing PI16+ fibroblasts.

In summary, these results suggest that CD19-CAR T cell treatment triggers a profound shift in the composition of fibroblast populations towards a physiological phenotype on the transcriptomic and proteomic level. In addition, we observed changes in their transcriptomic signatures with downregulation of pro-inflammatory and pro-fibrotic pathways, indicating a shift towards more homeostatic and regenerative functions.

## Increase of vessel structures in dcSSc skin upon CD19-CAR T cell therapy

SSc patients reported improvement of Raynaud's phenomenon upon CD19-CAR T cell treatment with reduction of the number and duration of attacks, as well as the pain intensity, as assessed by standardized questionnaires (Fig. 4B). Capillary density, as assessed by capillaroscopy and quantified by capillaries/distance, remained stable 1−3 months after therapy and significantly increased 6 and 12 months after the treatment (Fig. 4C). We also observed increased blood perfusion, as assessed by contrast optical coherence tomography in patients (Fig. 4D).

As assessed on sections stained for the endothelial cell marker CD31 (Figs. 4A, E, F), we observed an increase of mean length of vessel fragments and an integrated vessel length across the whole skin upon CD19-CAR T cell therapy, which was not observed in the NDG or SOC groups. Consistently, we observed higher numbers of CD31⁺LYVE1⁻ cells (vascular endothelial cells) after one month as analyzed by IMC (Supplementary Fig. 14). Vessels detected in the papillary dermis upon CD19-CAR-T cell therapy are exemplified in Fig. 4F.

As analyzed by gene set enrichment analysis (GSEA), angiogenesis-related genes were positively enriched at 1 month and 12 months after CD19-CAR T cell treatment (Fig. 4G), in line with the GSEA terms retrieved in Fig. 1D. Consistent with these results, a positive CAR T cell response score was obtained for endothelial cells and pericytes as defined by associating the RNASeq-derived DEG using cISH (Supplementary Fig. 3C).

Next, we phenotyped endothelial cell populations upon CD19-CAR T cell therapy and recovered vascular endothelial cells (VEC) and

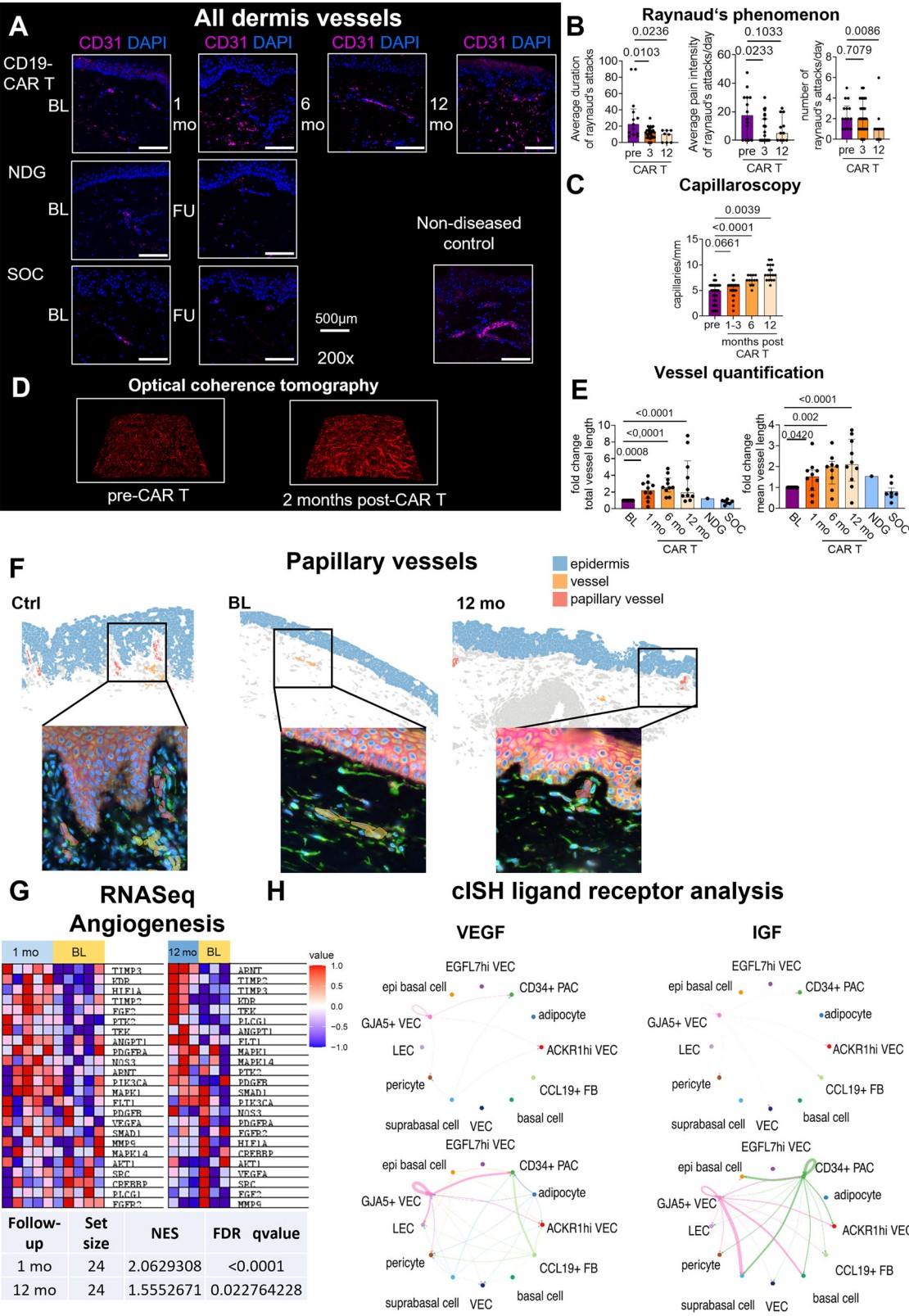

lymphatic endothelial cells (LEC) based on the expression of *PROX1*, *LYVE1*, and *VWF* (Supplementary Fig. 19A). We further subcategorized the VEC population based on the expression of arterial marker *GJA5*, venous marker *ACKR1*, pro-angiogenic factor *EGFL7*, and progenitor marker CD34[63,64]. Using FGSEA, we observed an upregulation of terms associated with immune response early after CD19-CAR T cell treatment ("leukocyte chemotaxis", "leukocyte migration" and "regulation

of T cell-mediated immunity") (Supplementary Fig. 9B) across different endothelial cell subsets, which turned negative by 12 months. These findings in endothelial cells are similar to the patterns observed in fibroblasts (Fig. 3A). Moreover, we observed an upregulation of angiogenesis-related terms, which were particularly prominent in GJA5 + VEC cells early after CD19-CAR T cell treatment and were detectable in several endothelial cell populations 12 months after CAR

**Fig. 4 | Vascular changes following CD19-CAR T cell therapy in dcSSc. A** Co-immunofluorescence staining for the vessel marker CD31 and 4′,6-Diamidino-2-Phenylindol (DAPI) (CAR T patient 2, NDG patient 1, SOC patient 4). Representative images are shown at 200-fold magnification. **B** Results of Raynaud's Questionnaires be (*n* = 4; patients 5; 8; 9; 12), 3 months (*n* = 5; patients 5; 6; 8; 9; 12) and 12 months (*n* = 4; patients 5; 6; 7; 8) after CAR T therapy. Each data point represents the results of Raynaud´s diary of one day. **C** Capillaries/mm were counted in several capillaroscopy pictures taken from at least 3 different fingers before (*n* = 3; patients 8; 9; 12), 1-3 months (*n* = 3; patients 8; 9 ;12), 6 months (*n* = 1; patient 8) and 12 months (*n* = 3; patients 8; 9; 12) after CAR T cell therapy. **D** Optical coherence tomography of forearm skin exemplified in patient 11. **E** Fold change of total and mean vessel length per section are shown as bar graphs. Measurements were normalized to the respective baseline sample. Baseline (*n* = 18), CAR T: 1-month follow-up (*n* = 10), 6-month follow-up (*n* = 9), 12-month follow-up (*n* = 10), NDG (*n* = 1) and SOC (*n* = 7).

**B–E** Mann-Whitney test was used; all data are presented as median with IQR. **F** Representative vessel patches of papillary vessel patches detected by cISH: Ctrl, BL: Patient 6, 12-month FU: Patient 1. Corresponding magnified views below display the cell segmentation staining for cISH in the same region. The scale bars represent 100 μm. **G** Gene Set Enrichment Analysis (GSEA) heatmaps and enrichment score table of a representative gene set for angiogenesis-related genes (WP1539) at 1-month and 12-month follow-up. **H** Enhanced cell communication for vascular formation after CAR T cell therapy. Circle plots showing cell-cell communication networks for VEGF (vascular endothelial growth factor) and IGF (insulin-like growth factor) signaling at BL and 12-month FU by spatially-informed ligand-receptor analysis. Line thickness indicates interaction strength and the color denotes the identity of source cell. Ctrl control; FU Follow-up; BL baseline; mo month; NDG natural disease course group; SOC standard of care; NES normalized enrichment score. FDR False discovery rate.

T cell treatment. Interestingly, we observed a strong upregulation of functional terms associated with insulin-like growth factor 1 receptor (IGF1R)-related signaling, which was also consistent with increased IGF-dependent communication between endothelial cells and fibroblast subtypes, as analyzed by spatially informed ligand-receptor analysis of cISH data using CellChat v2 (Fig. 4H) [38]. IGF regulates proliferation and migration of endothelial cells and can also reinforce the effects of VEGF-dependent signaling [39]. Consistent with this, we also detected an increased VEGF communication in endothelial cells (Fig. 4H).

### Epidermal cell remodeling during CD19-CAR T cell treatment

Consistent with the increase of dermal papillae described in Fig. 1, we observed an increase of rete ridges upon CD19-CAR T cell therapy (Fig. 5A). On the transcriptional level, we found changes in transcriptomic signatures relating to epidermis development (Fig. 5B), suggesting profound remodeling of the epidermal cell layer upon CD19-CAR T cell therapy.

As we observed prominent changes in genes related to the epidermis and keratinization in bulk-RNASeq, we annotated the epithelial cells in cISH using a similar approach. We identified basal, suprabasal, cornified, and glandular cells (Supplementary Fig. 20). Basal cells located within the epidermis were further segregated and labeled as "epi basal cells" (Supplementary Fig. 20).

By investigating the functional dynamics of epidermal cells, we detected an enrichment of gene signatures related to extracellular matrix organization and vascular development one month after CD19-CAR T cell treatment. Similar to fibroblasts, signatures associated with immune cell activation and interferon signaling were positively enriched early after CD19-CAR T cell treatment and downregulated after 12 months. These results suggest a functional modulation of keratinocytes upon CD19-CAR T cell therapy that may impact other cellular compartments. Spatially informed ligand-receptor analysis of our cISH data revealed an enhanced cell-cell communication between epi basal and suprabasal epidermal cells by bone morphogenic protein (BMP) and heparan sulfate proteoglycan (HSPG) signaling (Fig. 5E), both of which are implicated in epidermal cell remodeling[65]. Thrombospondin (THBS) signaling (Fig. 5E) has been reported to play a role in TGF-β signaling in SSc[66]. We observed attenuated THBS signaling sent out from superficial fibroblasts and epidermal basal cells at the 12-month follow-up compared to baseline, suggesting a reduction in pro-fibrotic phenotypes in the skin tissue. Interestingly, signaling via myelin protein zero (MPZ) and increased interactions between fibroblasts and epidermal cells with Schwann cells were observed, suggesting regulatory roles also in neuronal regeneration that may also contribute to regeneration of the papillae (Fig. 5E).

Our results indicate functional regeneration in epithelial cells upon CAR T cell treatment at the transcriptomic level. Further studies may be required for protein-based validation of these phenotypes.

## Discussion

Autoimmune diseases are characterized not only by immune dysregulation but also by pathological tissue remodeling. SSc exemplifies this dual process, in which autoimmune-mediated inflammation occurs alongside progressive fibrosis and vasculopathy. In this study, we addressed the fundamental question of whether deep B cell targeting can modulate pathological tissue responses in SSc, a prototypical fibrosing autoimmune disease. We performed an in-depth exploratory analysis of skin tissue of SSc patients treated with CD19-CAR T cell therapy and provide first evidence that deep B cell depletion can affect several pathogenic hallmark processes in SSc: fibrotic tissue remodeling and vasculopathy. A major finding of our study is the regeneration of dermal papillae, a highly complex and functional skin structure with a distinct vascularization, rich neuronal innervation and enrichment of specialized fibroblast subpopulations, which is disrupted in SSc. Loss of the papillae was thought to be irreversible so far. However, we demonstrate by bulkRNA seq, cISH and IMC that CD19-CAR T cell therapy induces profound changes in molecular expression patterns and cellular phenotypes across different cell types, particularly in the papillary layer of the dermis, that culminate in the formation of new dermal papillae in SSc patients. The improvement of histological outcomes persisted during the entire observation time in accordance with sustained improvements of the clinical response as assessed by mRSS.

Our results suggest several mechanisms of CD19-CAR T cell-mediated skin remodeling on the tissue level: First, the clinical improvement of mRSS is accompanied by a shift in fibroblast population composition toward a more physiologic distribution both at the transcriptomic and protein levels, as assessed by two complementary single-cell spatial imaging approaches (IMC and cISH). Across both datasets, we observe decreased proportions of fibroblast subsets associated with pro-fibrotic and pro-inflammatory activity, including reductions in PU.1⁺ fibroblasts and myofibroblasts, which showed a high collagen score and activation of TGFβ related signaling cascades as analyzed by cISH and in line with the previous literature[55,67]. Despite the previous scRNA-Seq study suggesting exclusive PU.1 expression in macrophages[55], IMC and cISH consistently detected reduction of PU.1⁺ fibroblasts. However, we could not exclude the possibility that the population was derived from fibroblast-macrophage doublets. Additional studies on fibroblast-macrophage interaction may further elucidate whether these signals reflect a genuine cellular phenotype or a close spatial association. Conversely, fibroblast populations linked to more homeostatic and potentially antifibrotic roles, such as PI16+ fibroblasts and TFAM+ fibroblasts—typically diminished in SSc— showed numerical increases following CD19-CAR T cell therapy. Of note, the shift towards regenerative fibroblast populations was particularly pronounced in the papillary dermis.

In addition to changes of fibroblast composition, we observe changes of fibroblast function as demonstrated by cISH: fibrosis-related FGSEA terms such as "extracellular matrix organization" and

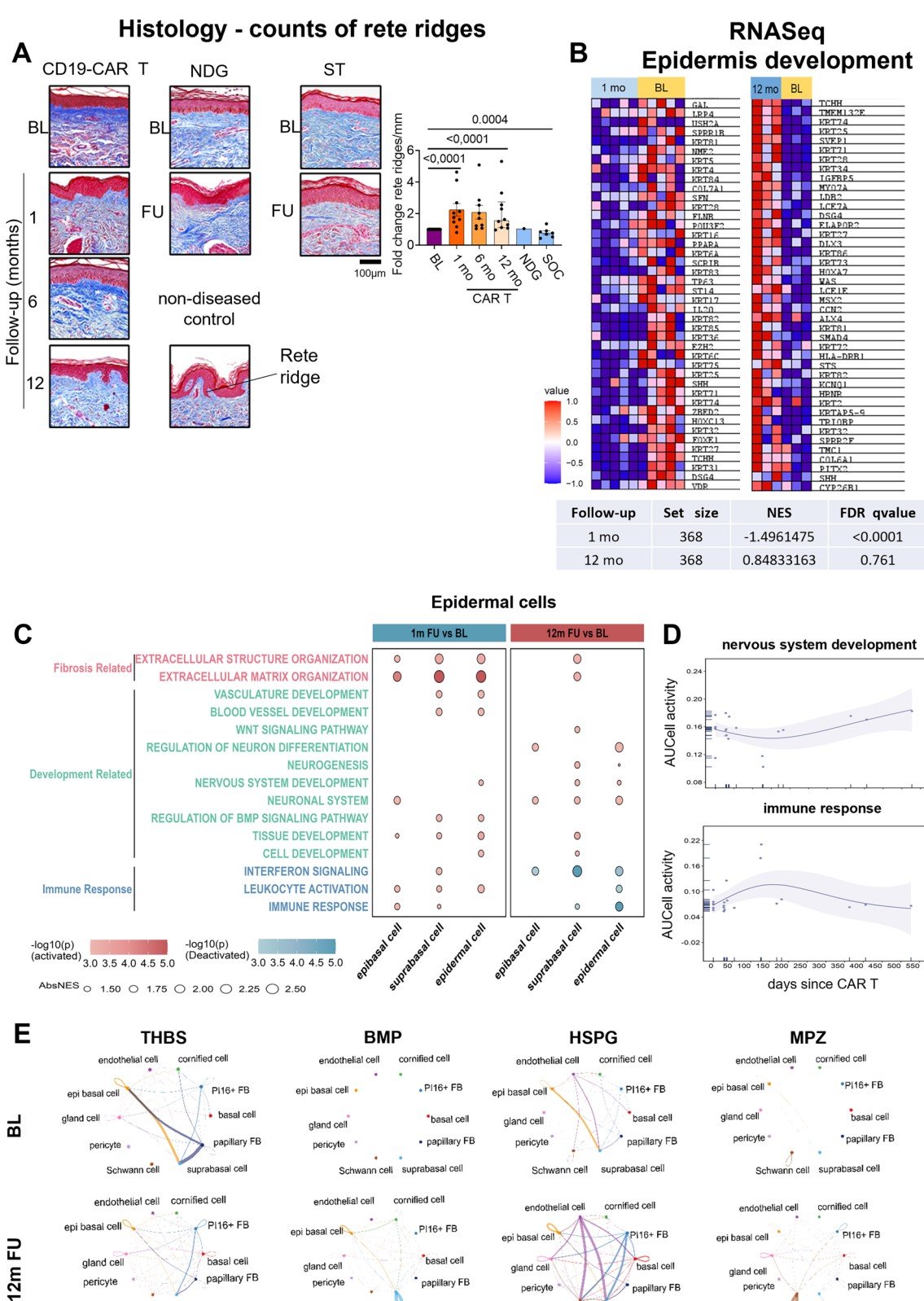

**Figure panels:** (A) Histology - counts of rete ridges; (B) RNASeq Epidermis development; (C) Epidermal cells; (D) nervous system development / immune response; (E) THBS, BMP, HSPG, MPZ signaling networks.

| Follow-up | Set size | NES | FDR qvalue |
|---|---|---|---|
| 1 mo | 368 | -1.4961475 | <0.0001 |
| 12 mo | 368 | 0.84833163 | 0.761 |

"collagen formation" decline across several fibroblast populations, particularly myofibroblasts. Interestingly, in addition to functional terms related to ECM regulation, regulatory functions are altered upon CD19-CAR T cell therapy and include, for instance, "neuron development". These results are consistent with the increase of Schwann cells (Fig. 2E) and may indicate enhanced crosstalk between nerve fibers and fibroblasts as part of the tissue regeneration process- a hypothesis that

needs further validation. In addition, transcriptomic signatures associated with endothelial cell differentiation increase in fibroblasts upon CD19-CAR T cell therapy and suggest enhanced fibroblast/vessel interaction. In line with clinical improvement of vascular symptoms such as Raynaud's phenomenon and increase of capillary density, CD19-CAR T cell therapy partially restored vascular skin structures on the histologic level. Transcriptomic analyses showed increased

**Fig. 5 | Epidermal changes following CD19-CAR T cell therapy. A** Representative Masson-Trichrome staining images (CD19-CAR T patient 5, NDG patient 1, SOC patient 4). Quantifications of rete ridges (rete ridges/mm) per section are shown as bar graphs. The data are presented as median with IQR. Mann-Whitney nonparametric test was used. *P*- values < 0.0125 were considered significant after Bonferroni correction. Baseline (*n* = 18), 1-month follow-up (*n* = 10), 6-month follow-up (*n* = 9), 12-month follow-up (*n* = 10), natural disease course group (*n* = 1) and standard-of-care treatments (*n* = 7). **B** Gene Set Enrichment Analysis (GSEA) heatmaps and enrichment score table of a representative gene sets for epidermis development-related genes (GO:0008544) at 1-month (CD19-CAR T patients 1 to 5) and 12-month follow-up (CD19-CAR T patients 1 to 3). **C** Dot plots showing enriched pathways across epidermal cells between baseline (BL) and 12-month follow-up (12 m FU) detected by FGSEA. Pathways are grouped into three functional categories—fibrosis-related, development-related, and immune response—and are colored accordingly. Dot size represents the absolute value of normalized enrichment score

(AbsNES), and dot color reflects the log-transformed *p*- value. Red and blue dots represent positive and negative enrichment scores indicating pathway activation and deactivation, respectively. **D** Scatter plots show the dynamic changes in AUCell-derived pathway activities at the pseudo-bulk level for epidermal cells. The x-axis represents the number of days after CAR T cell treatment. GAM (Generalized Additive Model) was used to fit smooth curves illustrating temporal trends, and the shaded band denotes the 95% confidence interval. **E** Circle plots showing cell–cell communication networks for THBS (Thrombospondin), BMP (Bone Morphogenetic Protein), HSPG (Heparan Sulfate Proteoglycan), and MPZ (Myelin Protein Zero) signaling pathways at baseline (BL) and 12-month follow-up (12 m FU) by spatially-informed ligand-receptor analysis. Line thickness indicates interaction strength and color denotes the identity of source cell. Ctrl control; FU Follow-up; BL baseline; mo month; NDG natural disease course group; SOC standard of care; NES normalized enrichment score. FDR False discovery rate.

enrichment of angiogenesis-related terms, which may be a central contributor to vessel structure regeneration. Consistently, increased cell-cell interaction with pro-angiogenetic signals such as VEGF and IGF were observed after CD19-CAR T cell treatment. These results indicate vascular remodeling upon CD19-CAR T cell therapy as part of the process of tissue regeneration; however, further functional studies, e.g., using vascular model systems, are warranted to describe these processes.

Compatible with the regeneration of the dermal papillae upon CD19-CAR T cell treatment, we observed an increase of rete ridges as epidermal counterpart and transcriptomic changes of cells at the dermal-epidermal interface suggesting remodeling upon CD19-CAR T cell therapy. Moreover, our findings obtained in cISH analysis including the temporal shifts IFN-responsive and ECM-related transcriptional programs, and dynamic BMP/HSPG, THBS, and MPZ signaling—suggesting that keratinocytes may act as active modulators of dermal/epidermal regeneration upon B cell depletion. A limitation is that these findings were not validated on the protein level; however, the transcriptomic data may generate several hypotheses that may be validated on the functional level in future studies, e.g., (i) BMP/HSPG-driven communication between epibasal and suprabasal layers may initiate epidermal re-stratification and (ii) MPZ-associated crosstalk between Schwann cells, fibroblasts, and epidermal cells may contribute to papillae regeneration. Such pathways may represent novel therapeutic entry points that synergize with CD19-CAR T cell therapy by enhancing epidermal–stromal repair. However, models that can adequately recapitulate the human dermal/epidermal interface are warranted to answer these research questions in the future.

Interestingly, we observe an upregulation of terms associated with immune response regulation in fibroblasts, endothelial cells and keratinocytes early (1 month) after CD19-CAR T cell therapy including "T cell activation", "interferon signaling" and "leukocyte chemotaxis". At the time of 1-month follow up biopsy, B cells were not detectable in the blood and skin tissue. Moreover, the peak expansion of CAR T cells had already subsided, and peripheral CAR T cells were no longer detectable, making a local manifestation of cytokine release syndrome unlikely. We thus speculate that these transcriptomics changes may be a reflection of the recently described local immune effector cell-associated toxicity syndrome (LICATS)[68], which is characterized by local organ specific inflammatory symptoms that occur after CD19-CAR T cell infusion and are usually transient and mild in their nature. In this cohort of SSc patients, a transient reddening of the skin was observed in one patient, so we assume the molecular findings occurred largely beyond the clinical detection threshold. The pathogenesis of LICATS is not fully defined, however, activation of autoreactive T cells that are reinfused as non-transduced T cells and subsequently expand alongside the CAR T cells or effects mediated by immune cells that repopulate after lymphodepletion, including—but not limited to—fludarabine-induced changes in myeloid cells and regulatory T cells may

play a role. By twelve months, this activation of inflammatory responses in fibroblasts and endothelial cells is no longer detectable, and in contrast, the respective signatures are even downregulated.

Taken together, our expression data argue for the modification of fibroblast-immune cell/ endothelial/immune-cell interactions as a mechanism of CD19-CAR T cell-mediated skin tissue changes in SSc. Herein, several mechanisms of B cell/fibroblast and B cell/endothelial interactions could possibly be involved: First, autoreactive B cells can secrete pro-inflammatory and pro-fibrotic cytokines such as Il6, Il13 and TGFβ[28,69] that sustain myofibroblast and endothelial cell activation. The removal of these auto-reactive B cells by CD19-CAR T cell therapy may result in a local tissue environment promoting fibrosis resolution and physiologic repair. This concept is supported by the downregulation of interferon-mediated transcriptomic signatures in fibroblasts, endothelial cells and keratinocytes upon CD19-CAR T cell therapy at the later time points (12 months) after short term upregulation at an earlier time-point. Of note, interferon mediated signaling in response to TGF-β was recently characterized as a mediator of skin tissue changes in SSc[70], and the downregulation of IFN-mediated signaling on the tissue level might thus contribute to CD19-CAR T cell mediated recovery of the skin tissue structure. The interception of B cell-mediated cytokine signaling by CD19-CAR T cells in the skin is further supported by the downregulation of THBS-mediated signaling between epidermis cells and papillary fibroblasts.

Moreover, CD19-CAR T cell therapy might affect direct B cell/ fibroblasts or B cell/endothelial cell interactions[71]. Indeed, we did not observe CD19 + B cells and CD20+cells in the available skin biopsies early (one month) or later (six or twelve months) after CD19-CAR T cell therapy given interindividual variability of numbers of B cells/mm in the pre-treatment sample. Another hypothesis for the mechanisms underlying the regeneration of skin tissue upon CD19-CAR T cell therapy is the interference with pathologic functional autoantibodies[23]. As surrogate for disease specific autoantibodies, we monitored the serum levels of anti-Scl-70 antibodies, which decreased upon CD19-CAR T cell therapy slowly but constantly, while remaining detectable. However, the effects of agonistic antibodies such as anti-PDGF receptor antibodies or anti-endothelial cell antibodies have not been investigated as part of this study. A recent study described the reduction of Fcγ-receptor-activating circulating immune complexes in AID upon CD19- CAR T cell therapy, which may also play a role[72]. These observations, together with the transient peripheral B cell depletion, may argue for a deep immune alteration resulting in regenerative tissue remodeling despite the persistence of anti-Scl70 antibodies, yet with declining titers. Both local and systemic immunologic effects may contribute to the mediation of CD19-CAR T cell mediated tissue responses. While the reduction of auto-antibodies and depletion of CD19+ cells in the lymph nodes suggest systemic effects, the role of local mediators cannot fully be excluded[37]. Further mechanistic studies are needed to clarify these contributions. Similar results on the

persistence of anti-Scl70 antibodies were also observed upon hematopoietic stem cell transplantation, which is considered a highly effective therapy in SSc: An analysis of the autoantibody repertoire of SSc-patients treated within the SCOT trial showed transient reduction of the anti-Scl70 antibody profile in two out of five Scl70 positive patients[73]. Despite the persistence of anti-Scl70 antibodies, hematopoietic stem cell transplantation provides proof-of concept for the elimination of autoreactive immune-cells resulting in the restoration of self-tolerance. Several studies showed the downregulation of interferon-related signatures in the peripheral blood upon HSCT, similar to our results on skin-related signatures, and the shift from pro-inflammatory to tolerant immune phenotypes[73–75]. In parallel, beneficial effects on skin tissue remodeling with reduced skin thickness and collagen deposition were described[17,76,77]. Overall, AHSCT is an effective therapy for SSc that can reduce the rate of disease-related death, however, it remains a complex procedure with treatment related risks[78]. CD19 CAR T cell therapy also eliminates B cells deeply and appears to have a rather favorable safety profile[39,79,80]. Given its clinical effects and the profound tissue remodeling described in this study, CAR T cell therapy might achieve immune mediated tissue regeneration with aspects similar to HSCT at a more favorable risk profile. Comparisons between the effects of CD19-CAR T cell therapy with AHSCT would be needed to further clarify this point.

The patients investigated in this study had refractory, diffuse cutaneous SSc with a progressive disease course despite multiple treatments. Interestingly, we observed a clinical and molecular response in a patient with progressive disease after seven years of disease duration, which may open avenues for further studies of the subgroup of patients who experience disease progression at later disease stages. However, other subgroups of SSc patients such as early SSc, or limited cutaneous SSc with organ involvement were not included in the target population of this study and may be investigated in future trials. However, such patients can only be treated with CAR T cells if showing high-level safety.

Our study has some limitations: while we provide detailed clinical, histological, proteomic, and transcriptomic evidence for skin regeneration toward a physiologic phenotype—including restoration of fibroblast, vascular, and epidermal compartments— a detailed analysis of the immunologic mechanisms behind the observed tissue changes including effects of B cells on other immune cells has not been performed as part of this study. Furthermore, to which extent B cell depletion in the tissue versus systemic immunologic reprogramming mediates the effects of CD19-CAR T cell treatment needs to be addressed in future studies with longer follow up and detailed investigation of immune cells.

Acknowledging the novelty of CD19-CAR T cell treatment as an only recently emerging therapy, the sample sizes are still limited. Moreover, sample sizes are uneven between the different follow up time points due to the consecutive inclusion of the patients. Thus, further investigation and studies in larger cohorts with extended longitudinal data are required to confirm the effects of CD19-CAR T cells on tissue changes in SSc. The results of the present study may help to generate hypothesis for the further exploration of the effects of CD19-CAR T cell therapy on different pathogenetic hallmarks and organ manifestations in SSc and support the definition of patient selection and outcomes in future CD19-CAR T cell related studies.

Given the complexity of the CD19-CAR T cell approach, studies including a randomized, blinded control arm were not available at the time of this manuscript. Thus, employing an external systemic sclerosis cohort represents a pragmatic alternative for contextualizing clinical outcomes, as also acknowledged in a recent recommendation of the Food and Drug Administration (FDA) for clinical trial design for cellular therapy products in small populations (FDA-2025-D-3403). Nonetheless, baseline differences between cohorts warrant careful consideration. In addition to a lower average baseline modified

Rodnan Skin Score (mRSS) in the external cohort—which may attenuate the apparent relative improvement attributable to treatment or influence the natural trajectory of skin scores—patients receiving CD19-CAR T- cells had a higher burden of prior immunomodulatory therapies, potentially reflecting more severe or refractory disease. This imbalance could affect skin responsiveness through cumulative treatment effects or altered tissue biology and may introduce residual confounding in comparative analyses. Therefore, comparative efficacy should be interpreted cautiously, taking these baseline differences into account.

SSc is a heterogeneous disease with varying organ involvement. Here we present detailed data on the effects of CD19-CAR T cell therapy on the molecular level in the skin and results of molecular imaging of the lung. However, the inclusion criteria and outcomes of this study were not directed towards the detailed assessment of other organ manifestations such as gastrointestinal system or myocardial involvement. The efficacy in these domains needs to be assessed in future trials.

Taken together, we provide first evidence that deep B cell depletion using CD19-CAR T cell treatment can substantially interfere with pathologic skin tissue remodeling in SSc including restoration of the physiologic fibroblast compositions and functionality, vascular repair and epidermal tissue remodeling that culminate in the regeneration of the dermal papillae. Deep B cell depletion may thus be the first therapy that facilitates regeneration of preexisting fibrotic tissue damage in SSc.

## Methods

### Patients

In this proof-of-concept study, samples of patients who had received CD19-CAR T-cell treatment as named patient use or within the CASTLE study were included. In line with a recent recommendation of the Food and Drug Administration (FDA-2025-D-3403) that refers to study design for cellular therapy products in small populations, control samples from a real-life cohort who matched the inclusion criteria and had received standard of care treatments according to national and international guidelines[81,82] and patients were included. The overarching inclusion criteria for the study were: fulfillment of the 2013 American College of Rheumatology (ACR)–European League Against Rheumatism (EULAR) criteria for systemic sclerosis, positivity for anti-Topoisomerase I (anti-Scl70) or anti-RNA polymerase III antibodies, and active systemic sclerosis as defined by either (1) disease duration of less than 7 years (since first non-Raynaud symptom); or (2) modified Rodnan Skin Score (mRSS) 10–35 at screening; or (3) elevated acute phase reactants (C-reactive protein ≥ 6 mg/L, erythrocyte sedimentation rate ≥28 mm/1 h, or platelet count ≥330 109/L); or (4) mRSS increase by 3 points or more or involvement of a new body area, or mRSS increase by 2 points or more in one body area, or 1 or more tendon friction rub instances over 6 months, or signs of progressive lung disease as defined by the criteria used in the INBUILD study[83] or new onset of interstitial lung disease within 6 months before baseline (≥ 10% disease extent, based on CT scan). Eligible patients had insufficient response or intolerance to at least two standard-of-care treatments, including mycophenolate mofetil, azathioprine, cyclophosphamide, nintedanib, methotrexate, rituximab, or tocilizumab. Patients with disease duration > seven years and no signs disease progression were excluded.

**CD19-CAR T-cell therapy.** We analyzed successive skin biopsies of eleven patients with dcSSc who received CD19-CAR T-cell therapy at the University Hospital Erlangen since August 2022. Four of the patients received CAR T-cell therapy at the Department of Internal Medicine 5, University Hospital Erlangen (Erlangen, Germany) as named patient use. Seven of the patients received CD19-CAR T-cells within the CASTLE phase 1/2 study (NCT06347718). Herein, patient 5

(Supplementary Table 1) was included during protocol version 4.0 and patients 6-11 were included within protocol version 5.0 (EudraCT-Nr. 2022-001366-35 CSP version 5.0 / 28.09.2023), which included the extension of the disease duration allowed at baseline from 5 years in the previous version to seven years in version 5.0.

CD19-targeting CAR T-cells were produced, and therapy was conducted as previously described[79,84]. Briefly, apheresis was conducted 13 days before CD19-targeting CAR T-cell infusion and T cells were transduced with a lentiviral vector (Miltenyi Biotec, Bergisch Gladbach, Germany) expressing a CAR directed against human CD19. Cells were expanded using the CliniMACS Prodigy device (Miltenyi Biotec, Bergisch Gladbach, Germany) for 12 days. The resulting investigational medicinal product MB-CART19.1 contained $1 \times 10^6$ CAR T cells per kg bodyweight, was produced for each patient individually and administered as a short infusion at day 0. Patients received lymphodepletion with fludarabine (25 mg/m2 intravenously on days −5, −4, and −3; 50% dose reduction in patient three due to renal insufficiency and dialysis) and cyclophosphamide (1 g/m² intravenously on day −3).

**Standard-of-care treatments.** For control, successive skin samples of patients who were matched to the inclusion criteria described above and had received standard-of-care treatments[81,82] including mycophenolate mofetil, tocilizumab, rituximab or methotrexate were investigated (ST-group). A table listing the indications for treatment change and applied standard of care therapy is visualized in the supplementary data 2. These patients are followed at the outpatient clinic at the Department of Internal Medicine 3 (Rheumatology and Immunology) according to EUSTAR recommendations[85]. Standard-of-care treatments with mycophenolate mofetil, tocilizumab or rituximab were performed according to the recommendations given by EULAR[81]. For mycophenolate mofetil, treatment was started at a dose of 250 mg twice daily and increased to up to 3 g daily[86]. Tocilizumab was administered subcutaneously at a dosage of 162 mg weekly. Rituximab was induced as two infusion therapies with 1,000 mg at intervals of 14 days followed by a maintenance therapy of 1,000 mg intravenously every 6 months.

**Natural disease course.** As an additional exploratory control, a sample from a patient with dcSSc who had stabilized upon standard-of-care therapies and did not receive immunosuppression was included as control referred to as "natural disease group".

All patients have given written informed consent for all procedures and data sharing according to CARE guidelines and in compliance with the principles of the Declaration of Helsinki. The collection and use of patient data and biomaterial are covered by license 334_18 B of the Institutional Review Board of the University Hospital Erlangen. All procedures were performed in accordance with the Good Clinical Practice guidelines of the International Council for Harmonization. The CASTLE study was approved by the ethics committee of Friedrich-Alexander University Erlangen (22-168-Az). The primary and secondary outcomes of this ongoing study are described separately.

**Clinical assessments**
MRSS was documented at baseline, one month, six months and twelve months after CAR19-CAR T-cell treatment. Raynaud's phenomenon was assessed using a patient-reported Raynaud´s Questionnaire assessing the median duration of Raynaud´s attacks in minutes, median pain intensity on a scale from 0 to 100, and number of Raynaud's attacks per day at three consecutive days at baseline as well as three months after CD19-CAR T-cell therapy. Lung function parameters including forced vital capacity (FVC) and diffusing capacity for carbon monoxide (DLCO) were assessed at baseline and at least every six months throughout the follow up period. Capillaroscopy was performed at baseline and one, three, six months after therapy, assessing the number of capillaries per millimeter in multiple pictures taken from 2-4 fingers at each time point. Optical coherence tomography (OCT) was performed on the back of the hand before and two months after the application of CD19-CAR T-cell therapy using the multi-beam OCT scanner VivoSight Dx (Michelson Diagnostics Ltd., Maidstone, Kent, UK). A scan of an area of 6 mm×6 mm provides 120 cross-sectional images (6 mm×2 mm) with a lateral image resolution of <7.5 μm, an axial resolution of <10 μm and a depth of 1.5 mm. Blood vessels were visualized using the dynamic mode, which exploits repeated scans of the same area to detect changes, which are then automatically color coded and correspond to blood flow in dermal vessels. Patients receiving standard-of-care treatments were regularly assessed every three months according to the EUSTAR recommendations.

**⁶⁸Ga-FAPI-04/ ⁶⁸Ga-FAPI-46 PET-CT examination**
Radiosynthesis and formulation of $^{68}$Ga-FAPI-04 and $^{68}$Ga-FAPI-46 was performed as previously described[43,44]. Both FAPI tracers, FAPI-46 and FAPI-04 show a similar distribution and pharmacokinetics[87]. Briefly, the precursor FAPI-46/ FAPI-04 was kindly provided by Prof. Uwe Haberkorn (University Hospital Heidelberg / German Cancer Research Center (DKFZ), Germany) and by iTheranostics (Dulles, VA, USA). The GMP-compliant radiosynthesis of $^{68}$Ga-FAPI-46/68Ga-FAPI-04 was set up under clean room conditions as previously described[43]. Patients received intravenous infusions of $^{68}$Ga-FAPI-46 at a mean radioactivity of 89 MBq (range: 47-211 MBq). In average the scan was acquired 38 min (range 13-100 Min. p.i.) after the infusion. First, a diagnostic computational tomography (CT) scan was acquired with 100 kV tube voltage and 40 mAs using the CARE Dose4D algorithm (Siemens, Erlangen, Germany) and 16×1.2-mm slice collimation at 0.5-s rotation time and a pitch of 1.0. CT was reconstructed with FBP (filtered back projection) using Br40 and Br59 kernels at slice thicknesses of 1,0 and 5,0 mm. The positron emission tomography (PET) scan started at the level of the thyroid and went down to the bottom of the liver. It was commenced immediately after the CT. PET data were reconstructed using an OSEM algorithm and time-offlight technology with 4 iterations/5 subsets and 2 mm full-width-at-half-maximum Gaussian post-smoothing.

All PET datasets were analyzed by visual interpretation of coronal, sagittal and transverse slices by two nuclear medicine specialists and one radiologist in consensus. Both nuclear medicine specialists were blinded to clinical information including pulmonary function testing and to the high-resolution computational tomography (HRCT) imaging results. Both attenuation- and non-attenuation corrected images were reviewed visually to ensure that areas of higher-density lung parenchyma were not inducing artifacts related to attenuation correction. For visually identified FAPpositive lung parenchyma, a VOI (diameter 10 mm) was placed to determine the mean and maximum standardized uptake value (SUVmean and SUVmax).

**Histological analyses**
Formalin-fixed, paraffin-embedded skin sections were stained with masson-trichrome staining as previously described[88–90]. Images were acquired at 200-fold magnification using a Nanozoomer S60v2MD slide scanner (Hamamatsu Photonics, Herrsching am Ammersee, Germany). Quantification of dermal papillae was performed by assessing the papillae numbers per mm and papillae heights at five different sites in each skin section in a blinded manner using the NDPView2 software (Hamamatsu Photonics)[14], and follow-up measurements were normalized to the corresponding baseline sample. To this end, papillae bases were set as the line connecting the two rete ridges surrounding each papilla. The papillae height was then determined as the perpendicular to the papillae base.

The alignment of collagen fibers, ranging from 0 to 180°, and alignment coefficients were assessed using Curvealign V4.0 Beta

(MATLAB). Using three regions of interest of 0,02 mm² per skin section, the alignment coefficient was measured through the CT-Fiber analysis method following the developer's manual (https://eliceirilab.org/software/curvealign/). The alignment coefficient was determined on a scale from 0 (indicating low parallelism between collagen fibers) to 1 (indicating high parallelism between collagen fibers).

## Immunofluorescence staining

Paraffin-embedded skin sections were fixed with 4% paraformaldehyde[91]. Epitope retrieval was carried out using a heat-induced method: sections were incubated with pre-heated citrate buffer (10 mM sodium citrate, pH 6.0) and Tris-EDTA buffer (10 mM Tris, 1 mM EDTA, 0.05% Tween 20, pH 9.0). Sections were blocked with PBS (Phosphate buffered saline) containing 2% BSA and 5% horse serum. Primary antibodies were incubated overnight at 4 °C. After washing, secondary antibodies were incubated for one h at room temperature. The sections were then counterstained with 4′,6-diamidino-2-phenylindole (DAPI) (1:800, #sc-3598, Santa Cruz Biotechnology, Heidelberg, Germany). The primary antibodies used targeted CD31 (1:200, #AF3628; R&D Systems, Minneapolis, USA), Fibroblast activation protein α (FAP) (1:200, #AF3715; R&D Systems, Minneapolis, USA), prolyl-4-hydroxylase β (P4Hβ) (1:200, AF4236, R&D Systems, Minneapolis, USA). Conjugated secondary antibodies (1:200, Alexa Fluor, Thermo Fisher Scientific, Dreieich, Germany) were used afterwards. The staining was analyzed using a Nikon Eclipse 80i microscope (Nikon, Tokyo, Japan) at 200x magnification.

The CD31 fluorescence intensity was quantified at five distinct sites in each skin section from each patient using the ImageJ software (NIH, version 1.46). The CD31 fluorescence signal was used to identify and define the boundaries of blood vessels, which allowed for the measurement of length of vessel fragments through the ImageJ software (NIH, version 1.46). A semi-quantitative analysis of FAP-positive cells was performed at three different sites per skin section in each patient in a blinded manner using the Nikon NIS-Element software platform (Nikon, Tokyo, Japan).

## RNA isolation and bulk RNA sequencing

RNA isolation from skin biopsies was performed using the NucleoSpin RNA kit from Macherey-Nagel according to the manufacturer's instructions (Macherey-Nagel, Düren, Germany). Skin biopsies were homogenized in Precellys Ceramic tubes using a Precellys 24 tissue homogenizer. Total RNA was extracted from each full-thickness skin sample using the NucleoSpin RNA kit (Macherey-Nagel) and its quality was determined using a TapeStation (Agilent). A library of cDNA fragments was created from 100 ng of each sample using the Illumina Stranded mRNA Kit. The libraries were sequenced paired-end with 2×150 bp length on a NovaSeq-6000 platform (Illumina). The raw data were converted into reads and stored together with a quality score (bcl2fastq v2.17).

## Analysis of RNA-sequencing

Quality control on the Illumina raw data was done using FastQC and TrimGalore. The processed paired-end reads were then aligned to the reference human genome (GRCh38 release 111) using the aligner, STAR, and the mapping outputs were exported to transcriptome coordinates for quantification with Salmon. Using the tximport R package, transcript level counts for each sample were read and converted to gene-level counts to produce a length-scaled counts matrix. Using the counts matrix and sample metadata, we performed differential expression (DEG) analysis using DESeq2. We also used sva to assess hidden factors contributing to technical variance and included them in the design matrix. For DEG analysis, we modeled patient and time to extract desired contrasts (post-treatment samples against the reference, before CAR treatment samples). Significant DEG were

exported at an adjusted P-value cutoff <0.05 and fold change cutoffs (>1.5 and <0.67) for each comparison.

## Gene set enrichment analyses (GSEA) and Gene Ontology Analysis

Gene set enrichment analyses were performed in GSEA 4.3.3[92]. Five gene sets were considered: a Wikipathways data set of angiogenesis-related genes under the reference number WP1539, a data set of epidermis development-related genes annotated by the GO term GO:0008544, a data set of extracellular matrix regulator genes under the accession number M3468[93], a data set of extracellular matrix degradation-related genes under the accession number M587, and a data set of extracellular matrix elasticity-conferring constituents annotated by the GO term GO:0030023. A normalized enrichment score (NES) was computed for each gene set of interest, and a false discovery rate (FDR)-adjusted q-value of <0.05 was considered statistically significant. GSEA plots were generated on R version 4.4.1 using the ggplot2 package[94].

## Imaging Mass Cytometry (IMC)

Formalin-fixed paraffin-embedded (FFPE) skin sections of 5 μm thickness were used. A hematoxylin-eosin (H&E) and trichrome histology staining were performed on consecutive sections (images scanned using Nanozoomer S60 slide scanner) for pathological analysis and selection the regions of interest (ROIs) to be scanned and analysed by Imaging Mass Cytometry (IMC). No scanned sample was removed from the analysis.

All non-metal conjugated antibodies were first validated by immunofluorescence staining on human skin sections using serial dilutions and a staining without primary antibody as negative control (Supplementary Methods, table 1). The non-metal conjugated antibodies that passed the validation step were labeled with indium (In) or lanthanide metals (Ln) using the Maxpar® X8 Labeling kit (Standard Biotools (South San Francisco, Ca, USA), #201300) following manufacturer's instructions. After the conjugation, the antibodies were stored at a concentration of 0.5 mg/ml in Antibody Stabilizer PBS (Boca Scientific (Dedham, MA, USA), #131050) supplemented with 0.05% sodium azide. Finally, all antibodies (pre-conjugated and self-conjugated) were re-validated and titrated by IMC on FFPE human skin sections. The dilutions that presented high signal and the highest signal-to-noise ratio were selected.

IMC staining was performed as previously described[58,95]. Briefly, FFPE 5 μm thickness sections were incubated at 65 °C in a dry oven for 40 minutes to melt the paraffin before deparaffinization with Xylol (3 ×10 minutes). The samples were rehydrated through serial ethanol dilutions (100%, 90% and 80%) and washed with double-distilled water (ddH₂O). Slides were incubated for 30 minutes at 96 °C in Tris-EDTA solution (10 mM Tris, 1 mM EDTA, 0.05% Tween 20, pH 10.0) for antigen retrieval. Afterwards, the samples were placed in a humidified chamber and non-unspecific antibody bindings were blocked by incubating with 2% BSA in PBS for 1 h at room temperature. Subsequently, the antibody staining was performed with 100 μl/sample of antibody cocktail prepared in 0.5% BSA in DPBS and incubated overnight at 4 °C. The next day, the slides were washed with PBST (DPBS + 0.02% Tween 20) and DNA staining was performed with Ir-intercalator (Standard Biotools (South San Francisco, Ca, USA), #201192 A) 1:400 for 5 minutes at room temperature. Finally, after washing with DPBS and ddH₂O, the slides were dried and stored at room temperature.

According to the H&E and trichrome images, the ROIs to be scanned were selected in order to include similar proportions of epidermis, papillary and reticular dermis and subcutaneous tissue. When possible, an area of 2.5 mm² (1 ×2.5 mm) was ablated. The samples were measured using a cytometry by time-of-flight (CyTOF) (Helios,

Standard Biotools, South San Francisco, Ca, USA) coupled to the Hyperion Imaging System (Standard Biotools, South San Francisco, Ca, USA). Tissue ablation was performed at a laser frequency of 200 Hz and in a pixel resolution of 1 μm² using the CyTOF software (version 7.0.8493, Standard Biotools, South San Francisco, Ca, USA). Calibration and quality control of the device was performed daily using a metal-coated tuning slide (Standard Biotools (South San Francisco, Ca, USA), #201088) following the manufacturer's instructions.

To ensure the quality of the staining and its consistency across all acquired samples, each individual.mcd file for each ROI was examined. In addition, heatmaps, multidimensional scaling (MDS) plots and uniform manifold approximation and projection (UMAPs) plots were generated to compare the protein expression in each ROI. The staining quality was also visualized by plotting the signal intensity against the signal-to-noise ratio.

Cell segmentation was performed using the publicly available Steinbock pipeline and its DeepCell implementation[96]. Cells were identified using Ir-intercalator and Histone 3 as nuclear signal and the commercial cell segmentation kit (Standard Biotools (South San Francisco, Ca, USA), #TIS-00001) as cytoplasm/membrane staining. The single-cell data generated included the mean signal intensity of all pixels belonging to the same cell, their spatial localization and neighbor relationships.

An inverse hyperbolic sine transformation (arcsinh) with a coefficient of 1 was applied to the data. Afterwards, the data was normalized by calculating a z-score using the whole dataset as reference. A second z-score was calculated within the fibroblasts. Finally, the data was exported as.fcs and.csv files for further analysis.

Single-cell phenotyping was performed. Firstly, erector muscles were removed by manual gating using FlowJo software (version 10.8.1, BD). After, the main cell types were identified by manual gating also using FlowJo software and comparing the spatial localization of the defined cells with the raw images. The following cell types were identified (supplementary methods, Fig. 1):

–Epithelial cells: E-cadherin+
  • Epidermis (defined by their spatial localization)
  • Hair follicle (defined by their spatial localization)
–Non-epithelial cells: E-cadherin-
  • Immune cells: CD45 + ,E-cadherin-
  • Endothelial cells: CD31 + ,E-cadherin-

  • Endothelial cells LYVE1 + : LYVE1 + ,CD31 + ,E-cadherin-
  • Endothelial cells LYVE1-: LYVE1-,CD31 + ,E-cadherin-
  • Fibroblasts: CD45-,CD31-,E-cadherin-
  • Vascular smooth muscle cells: SM22^high,αSMA^high,E-cadherin-
  • Fibroblasts: SM22^low/-,αSMA^low/-,E-cadherin-

We used the R package imcRtools[96] to perform dimensionality reduction (UMAP) and unsupervised PhenoGraph clustering of immune cells and fibroblasts. Fibroblasts were clustered using a K value of 40 and the z-score normalized protein expression of the markers: PU.1, TFAM, PDGFRαa, GLI2, αSMA, PDGFRβb, GLI1, S1PR, CD90, Collagen, ADRP, ADAM12, PI16, P4Hβb and FAP. For the clustering of immune cells, we employed a K value of 20 and the markers: CD20, CD29, CD45, CD4, LYVE1, CD8, CD3 and CD68. The similarities of the obtained clusters were studied using UMAPs, MDS plots, protein expression, Pearson correlation, frequencies over time-points and spatial localization. Similar clusters showing minor differences were merged to recover the biological meaning, to obtain non-overlapping fibroblast subpopulations and to avoid over clustering. We annotated these clusters according to their defining markers and protein expression.

Subsequently, the frequency of each cluster has been studied for each patient individually. For statistical analysis, paired t-tests were used to account for the repeated measurements.

For spatial analysis, the border between papillary and reticular dermis has been previously defined by the superficial vascular plexus[14] which corresponds to 100 μm from the epidermal basal membrane. Therefore, using the function minDistToCells() the distance from the epidermis basal membrane to each fibroblast was calculated, and subsequently, each non-epidermal cell was classified as papillary (distance <100 μm) or reticular (distance > 100 μm).

These data were merged with the healthy data of another dataset with 22 shared protein markers[58]. All shared markers (PU.1, CD31, SM22, TFAM, PDGFRa, αSMA, PDGFRb, CD45, E-cadherin, Gli1, Podoplanin, CD90, Collagen, ADRP, vWillebrand, ADAM12, S1PR, Ki67, PI16, p16, FAP and CD68) were used to generate a UMAP using the R package Specter[97] As the UMAP showed high degree of overlap between the two datasets, a k-nearst neighbor (Knn)-based label transfer (K = 10) was performed to assign fibroblast annotation to the healthy dataset.

### Cyclic in-situ hybridization (cISH) and data analysis
**Tissue preparation and cISH.** Deparaffinization and decrosslinking of FFPE tissues were performed based on the Xenium In Situ protocol (10X Genomics, CG000580 Rev E). Slides with 5-μm thick FFPE skin sections were first incubated at 60 °C for 30 minutes, followed by two rounds of xylene treatment and graded ethanol washes for deparaffination and rehydration. Decrosslinking was carried out at 80 °C for 30 minutes, then at room temperature for 10 minutes, before rinsing in PBS-T.

Subsequent processing of the slides followed the general workflow outlined in the Xenium Prime In Situ Gene Expression protocol (10X Genomics, CG000760 Rev B). The procedure began with priming setup, which improves the specificity and efficiency of the mRNA probe hybridization using priming oligos and RNase treatment. mRNA-targeting padlock probes were then hybridized at 50 °C for 17 h. The genomic sequences targeted by the probes are provided in the supplementary data 6. After hybridization, a series of washes were performed prior to ligation. Rolling circle amplification was then carried out to enhance the sensitivity of the assay. Tissue sections were subsequently stained using Xenium Cell Segmentation Kit (10X Genomics) for multimodal cell segmentation, and autofluorescence was minimized using a quenching reagent (10X Genomics) before counterstaining with DAPI and loading into the Xenium Analyzer.

cISH and multi-cycle imaging were performed according to the Xenium Analyzer User Guide (10X Genomics, CG000584 Rev G). Images processing, transcript decoding, and multimodal cell segmentation were automatically executed using Xenium Onboard Analysis (10X Genomics, version 3.0). Cell segmentation was further refined by using Xenium Ranger (10X Genomics, version 3.1) with DAPI threshold raised to 2000 and Baysor transcript-based segmentation with prior segmentation confidence set to 0.99 (https://github.com/kharchenkolab/Baysor)[98]. Followed by validating the cell segmentation, the gene count matrices and spatial coordinates were transformed and aggregated into a Seurat object for subsequent analysis[98].

**Data preprocessing.** Cells were filtered based on cell area, the proportion of negative control genes, and the number of detected features. Area thresholds varied between samples and were manually adjusted to exclude segments with extremely small or large areas.

The data was processed following recent published recommendations for Xenium datasets[99]. In brief, gene expression data were normalized using a shifted logarithmic transformation with a scaling factor of 300. Scaling was performed using all genes, and each sample was scaled independently to account for sample-specific variability.

**Cell phenotyping by clustering.** Principal component (PC) analysis was performed using all genes detected, and the top 30 PCs were used for clustering. Cells were annotated based on known marker genes from literature and further confirmed using Scimilarity (https://github.

com/Genentech/scimilarity)[100]. The combined results were used to assign major cell types, and sub-clusters were identified using previously published marker genes, with a specific focus on fibroblast and keratinocyte subpopulations for downstream analysis[101].

**Cell composition analysis.** To identify changes in cell composition after CD19-CAR T-cell therapy, we utilized the sccomp R package (https://github.com/MangiolaLaboratory/sccomp), which employs a Bayesian modeling framework based on sum-constrained Beta-binomial distributions[102]. The default sccomp thresholds of a minimum effect size of ±0.1 and an FDR cutoff of 0.05 were used to determine statistically significant compositional changes. Patient-specific effects were included as random intercepts to account for variation between individuals.

**Gene signature scores and functional analysis.** Collagen scores were calculated using Seurat's *AddModuleScore* function based on collagen genes retrieved from the Matrisome Project. The CAR T-cell response score and TGFβ pathway activity was inferred using decoupleR R package (https://github.com/saezlab/decoupleR) with a multivariate linear model on the differentially expressed genes, comparing 1-month follow-up to baseline by bulk-RNA-Seq, and the top 2000 most highly genes from PROGENy, respectively[45,103,104].

The CAR T-cell treatment response score was computed from the DEG analysis on bulk RNA-Seq data (1-month follow-up compared to baseline) by multivariate linear modeling using the decoupleR R package, with DEGs (adjusted *p*-value < 0.05) and log2-fold changes input as mode of regulation (Supplementary Data 5). To visualize the distribution of CAR T-cell treatment response score, we first used the sf R package to visualize spatial polygons of cell segmentation. The weighted spatial Gaussian kernel density estimates of fibroblasts and CAR T-cell treatment response score were calculated using spatialEco R package (https://github.com/jeffreyevans/spatialEco) and overlaid with the segmentation polygons to visualize their spatial distributions.

Functional analysis across different cell populations and time points were performed using the fgsea R package[105]. Differential expression analysis was performed for each pairwise comparison between populations and time points using Seurat's *FindMarkers* function with minimal log2 fold change set to 0.1 and genes that were detected in at least 1% of the populations. The fold changes of the DEGs were then set as the ranks for the FGSEA analysis with a curated gene set obtained from Bader Lab (version Human_GOBP_AllPathways_noPFOCR_no_GO_iea_March_01_2025)[106].

To investigate temporal changes in pathway activity over time, we used the AUCell method implemented in the decoupleR R package to score selected pathways using pseudobulk expression profiles. Generalized Additive Models (GAMs) were then applied to model the temporal trends of pathway activities.

**Validation of cISH-identified fibroblast populations.** We validated our cISH-identified fibroblast populations by scANVI-based integration of the cISH data and the recently published skin fibroblast atlas using scvi-tools Python package (https://github.com/scverse/scvi-tools)[107]. The fibroblast atlas was downloaded from the web portal of Haniffa Lab (https://cellatlas.io/studies/skin-fibroblast). We removed the data obtained from cancer, benign tumors, or granulomatous conditions, and we filtered the dataset to include only genes in the cISH panel.

We first trained the filtered atlas data using scVI with the following hyperparameters: 2 layers of neural network, 30 latent dimensions, model gene likelihood with zero-inflated negative binomial distribution, and GEO accession number as batch variable. The scANVI reference model was trained based on the previously trained scVI model and the cell type labels provided in the atlas. We then updated the scANVI model with the cISH data and trained 100 epochs with the

"weight_decay" parameter set to 0 for reference mapping. The label transfer was performed using *scvi.model.SCANVI.predict()* function. The correspondence between the cISH annotation and the atlas labels was then visualized in sankey plots using the ggsankey R package. Spurious correspondences, defined as pairs containing fewer than 50 cells or representing less than 10% of a given cISH label, were removed from the visualization.

**Spatially informed ligand-receptor analysis.** Spatially informed cell-cell communication was analyzed using the CellChat v2 R package[108] with ligand–receptor data from the CellChatDB v2 database. Gene expression was summarized using the truncated mean approach (10% trimming at both ends). For spatial constraints, maximum distances were set to 250 μm for secreted signaling and 20 μm for contact-dependent signaling. Only interactions involving at least 10 cells were considered for downstream analysis.

**Identification of dermal compartments and papillary vasculature.** As epidermal layer consists of compact keratinocytes with high density, we first identify cellular patches of keratinocytes with at least 10 keratinocytes per patch by using imcRtools R package. The keratinocyte patches located above the dermis were then annotated as the epidermal layer. The papillary and reticular dermis were defined through the distances to epidermal layer as described above for IMC analysis.

For the identification of the papillary vessels, endothelial cells located within the papillary dermis were first extracted for further analysis. Individual vessel structures were identified as cellular patches with at least three endothelial cells in the neighborhood, defined by knn (k = 10). Since the papillary vessels tend to be perpendicular to the epidermal layer, we next computed the angle of each papillary vascular patch respected to the epidermis using principal component analysis (PCA) on spatial coordinates of the cells. The first PC in this setting fits the long axis of the vascular patch or the epidermal layer. The angle between fitted lines of vascular patch and epidermis was then computed. Patches with an angle >45° relative to the epidermis were highlighted with red color in Fig. 4F.

**Other statistical analysis.** Data are displayed as median with interquartile range or mean with standard deviation with individual data points presented as dots. Comparisons between experimental groups were analyzed with non-parametric Mann-Whitney-U-test or t-tests depending on the distribution using GraphPad Prism 8.3.0. Differences of proportion in categorial datasets with two groups were tested using Fishers Exact test. In analysis including multiple tests, the *p*- value of each test (pi) was adjusted using the Bonferroni correction (pi= p/n, *n*= number of tests, *p*=overall significance level).

### Reporting summary
Further information on research design is available in the Nature Portfolio Reporting Summary linked to this article.

## Data availability
The clinical parameters and differential gene analysis used in the study have been provided in supplementary data files. The transcriptomic datasets generated in this study are not publicly available due to legal restrictions. The processed data is available upon request to the corresponding author. Access will be granted subject to approval by the corresponding Ethics Committees. Data and documents related to the CASTLE study (EudraCT identifier: 2022-001366-35) are published elsewhere[109]. Source data are provided with this paper.

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

## Acknowledgements

We thank Dr. Engel (Siemens Healthineers) for 3D reconstruction of PET/CT scans using cinematic rendering. Moreover, we thank Wolfgang Espach for excellent technical assistance. The analysis was supported by a de.NBI Cloud project (YNL) within the German Network for Bioinformatics Infrastructure (de.NBI) and ELIXIR-DE (Forschungszentrum Jülich and W-de.NBI-001, W-de.NBI-004, W-de.NBI-008, W-de.NBI-010, W-de.NBI-013, W-de.NBI-014, W-de.NBI-016, W-de.NBI-022) and the Center for Information and Media Technology at Heinrich Heine University Düsseldorf. In addition, we thank all members of the Spatial and Functional Screening Core Facility, Medical Faculty of Heinrich Heine University. The CASTLE study was started and initially sponsored by Friedrich-Alexander-Universität Erlangen-Nürnberg. Sponsorship was transferred to "Miltenyi Biomedicine GmbH" in November 2024. This study was supported by the Deutsche Forschungsgemeinschaft (DFG) through the Leibniz Award (GS), CRC1483, CRC/TRR305 and CRC/TRR221 (AM), BE 7036/5-1, DI 1537/23-1 (JHWD), CRU5024 (PG), NOTICE 493624887, CRC1755 (CASCAID-55029805- Z01,TP01, TP02, TP06, TP07, TP08, TP15). FM is supported by the German Cancer Aid Grant-No. 70113695). Further funding has been obtained from the Bundesministerium für Forschung, Technologie und Raumfahrt (BMFTR) through BMFTR #01EO2105 (CB, Advanced Clinician Scientist Program iIMMUNE), a EUSTAR grant to center 106, the Staedtler Foundation and donations from the Bendel family and the Bleyl family, by the Hiller-Foundation through an unrestricted research grant (JHWD); by the Research Committee of the Medical Faculty of the Heinrich-Heine-University Düsseldorf through grant 2023-31 (JHWD). MGR is supported by the Interdisciplinary Center for Clinical Research (IZKF) Erlangen through the grants P049 and J106. The project was further supported by a database improvement grant to center 106. P.Gu. was supported by the AI-PREDICT project (grant no. 01ZU2502) funded by the BMFTR, as well as by Deutsche Forschungsgemeinschaft (DFG, German Research Foundation) through KFO 5024 (Project-ID 505539112, Project Z01), TRR 417 (Project-ID 540805631, Project S03). The present work was performed in (partial) fulfillment of the requirements for obtaining the degree PhD for JA, ZL and CT.

## Author contributions

A.R.R., M.Xu and Z.L., S.C.F. contributed equally. Yi.-N.L. and C.B. supervised equally. A.R.R., Y.-N.L. and C.B. designed the study. A.R.R., M.X., C.S.F., Z.L., J.A., P.G., A.Z., M.K., M.G.R., C.T., P.G., Y.-N.L. and C.B. were involved in the acquisition and analysis of data. A.R.R., M.X., C.S.F., Z.L., J.A., Y.-N.L. and C.B. were involved in the interpretation of data. A.R.R., M.X., C.S.F., Y.-N.L. and C.B. created the original draft and data visualization. A.R.R., M.X., C.S.F., Y.-N.L. and C.B. acquired resources and oversaw this study. A.R.R., M.X., C.S.F., Z.L., J.A., P.G., A.Z., M.K., M.G.R., C.T., T.F., P.G., M.E., F.M., A.A., M.R., A.E., R.S., A.W., M.H., S.B., T.K., R.E.H., C.B., R.G.B., A.R., P.G., A.B., A.M., J.H.W.D., G.S., Y.-N.L. and C.B. were involved in manuscript preparation and proofreading. A.R.R., M.X., C.S.F., Y.-N.L. and C.B. take full responsibility for the overall content as guarantors.

## Funding

## Competing interests

J.H.W.D. has consulted for Active Biotech, Anamar, ARXX, AstraZeneca, Bayer Pharma, Boehringer Ingelheim, Bristol Myers Squibb, Callidatas,

Calluna, Galapagos, GSK, Janssen, Kyverna, Novartis, Pfizer, Quell Therapeutics and UCB; has received research funding from Anamar, ARXX, BMS, Boehringer Ingelheim, Cantargia, Celgene, CSL Behring, Exo Therapeutics, Galapagos, GSK, Incyte, Inventiva, Kiniksa, Kyverna, Lassen Therapeutics, Mestag, Sanofi-Aventis, SpicaTx, RedX, UCB and ZenasBio; is the CEO of 4D Science and scientific lead of FibroCure. JA was supported by the Kyverna Therapeutics Travel. CB received speaker fees from Novartis. The other authors declare no competing interests.

## Additional information

https://doi.org/10.1038/s41467-026-72817-7).

[1]Department of Internal Medicine 3 - Rheumatology and Immunology, Friedrich-Alexander-Universität Erlangen-Nürnberg and Uniklinikum Erlangen, Erlangen, Germany. [2]Deutsches Zentrum Immuntherapie (DZI), Friedrich-Alexander-Universität Erlangen-Nürnberg and Uniklinikum Erlangen, Erlangen, Germany. [3]Department of Rheumatology, University Hospital Düsseldorf, Medical Faculty of Heinrich Heine University, Düsseldorf, Germany. [4]Hiller Research Center, University Hospital Düsseldorf, Medical Faculty of Heinrich Heine University, Düsseldorf, Germany. [5]Department of Rheumatology, Fondazione Policlinico Universitario A Gemelli, IRCSS, Catholic University of Sacred Heart, Rome, Italy. [6]Core Unit for Bioinformatics, Data integration and Analysis (CUBiDA), Medizinisches Zentrum für Informations- und Kommunikationstechnik (MIK), Universitätsklinikum, Erlangen, Germany. [7]Department of Pathology, Friedrich-Alexander-Universität Erlangen-Nürnberg and Uniklinikum Erlangen, Erlangen, Germany. [8]Department of Internal Medicine 5, Hematology and Oncology, Friedrich-Alexander-Universität Erlangen-Nürnberg and Uniklinikum Erlangen, Erlangen, Germany. [9]Department of Nuclear Medicine, Friedrich-Alexander-Universität Erlangen-Nürnberg and Uniklinikum Erlangen, Erlangen, Germany. [10]Department of Dermatology, Friedrich-Alexander-Universität Erlangen-Nürnberg and Uniklinikum Erlangen, Erlangen, Germany. [11]Institute of Human Genetics, Friedrich-Alexander-Universität Erlangen-Nürnberg and Uniklinikum Erlangen, Erlangen, Germany. [12]Department of Plastic and Hand Surgery, Laboratory for Tissue Engineering and Regenerative Medicine, Friedrich-Alexander-Universität Erlangen-Nürnberg and Uniklinikum Erlangen, Erlangen, Germany. [13]Department of Pediatrics, Friedrich-Alexander-Universität Erlangen-Nürnberg and Uniklinikum Erlangen, Erlangen, Germany. [14]Department of Stem Cell Biology, Friedrich-Alexander-Universität Erlangen-Nürnberg and Uniklinikum Erlangen, Erlangen, Germany. [15]Spatial & Functional Screening Core Facility, Medical Faculty of Heinrich Heine University, Düsseldorf, Germany. [16]These authors contributed equally: Aleix Rius Rigau, Meilin Xu, Ziyuan Liu, Sara Chenguiti Fakhouri. [17]These authors jointly supervised this work: Yi-Nan Li and Christina Bergmann. ✉e-mail: christina.bergmann@uk-erlangen.de

