## [Transparent Peer Review file · Nature Communications]

Deep phenotyping of skin tissue remodeling in patients with systemic sclerosis treated with CD19-CAR T cells

Corresponding Author: Dr Christina Bergmann

Version 0:

Reviewer comments:

Reviewer #1

(Remarks to the Author)

The manuscript presents an important and timely investigation into the effects of CAR T therapy on skin pathology, with a focus on histological and molecular changes over the course of treatment and follow-up. The topic is highly relevant and the authors employ a combination of imaging, histology, and single-cell profiling to explore these dynamics. The breadth of data, including multiple time points and supplementary figures, provides a potentially rich resource for understanding disease progression and treatment effects.

While the study is ambitious and addresses a clinically meaningful question, certain aspects of the presentation and figure interpretation require clarification to ensure that the results can be accurately interpreted by readers.

Figure 1 and 5 - Images of CAR T and NDG skin at baseline look different from each other, especially in the papillary dermis. This impairs an immediate comparison of the skin changes following CAR T treatment with the skin modifications happening during the natural course of the disease.

Obviously, they are taken from different patients and may have been taken at different stages of the disease according to Supplementary Table 1. The NDG baseline shown here is from the patient with 5 months disease duration or 11 years? Moreover, 12 months FU and SOC images have a different intensity of blue staining: are they thinner sections?

Figure 2D is not clear. Does the left image represent the spatial distribution of fibroblast density at baseline?

Supplementary Figure 1: representative images of immunofluorescence staining of CD19+ and CD20+ B cells in the skin should be shown.

Supplementary Figure 2 A: does the UMAP plot depict clustering of major cell types detected by cISH at baseline or at 12 months FU?

Supplementary Figure 15: does it refer to epithelial populations detected in skin upon CAR T treatment? At which time point FU?

Finally, I suggest removal of 20 years' old reference 1 since it is quite outmoded and replaceable with updated reviews based on novel clinical and pathogenic concepts and study methods.

Reviewer #2

(Remarks to the Author)

This is a worthwhile and timely study, addressing a long-standing unmet need in systemic sclerosis: the possibility of reversing fibrotic tissue damage through targeted immunotherapy. The work is rich in content and provides numerous insights into the possible cellular and molecular remodeling of the skin following CD19-CAR T-cell therapy. The integration

of transcriptomic, proteomic, spatial, and imaging data is impressive and adds depth to the interpretation. However, the exploratory nature of the study should be more explicitly acknowledged. Several analyses rely on small and uneven sample sizes, and statistical robustness is limited in many comparisons. While the findings are promising, they should be interpreted with caution and framed as hypothesis-generating. Clearer articulation of the study's scope and limitations would help contextualize the results and guide future confirmatory research.

Major

- 1) According to the stated inclusion criteria, at least one patient in the treatment group (n.4), two in the SOC group (n.2 and 8), and one in the NDG group (n.2) appear to exceed the maximum disease duration allowed by criterion #1. Patients n.10 and 11 are also borderline. Are these numbers correct or there are whatsoever typos? If the provided numbers are correct, the exclusion of these patients is not trivial, as prolonged disease duration may shift the pathological balance toward atrophic rather than inflammatory or fibrotic processes, which are more amenable to immunomodulatory interventions. Despite the compelling results presented, I remain unconvinced that CD19-CAR T-cell therapy is suitable across all disease stages and/or for all patients. Strict adherence to inclusion criteria would help isolate the therapeutic effect in a well-defined subset of patients—those at higher risk of progression and typically enrolled in randomized clinical trials. Given the substantial burden and cost of CAR T-cell therapy, clearer guidance on patient selection is essential to ensure appropriate clinical translation.
- 2) Ancillary to comment #1, it would be helpful to explicitly list the main clinical indications that led to CD19-CAR T-cell therapy administration, particularly in relation to inclusion criterion #4 (e.g., progressive skin involvement, worsening interstitial lung disease, cardiac fibrosis). Furthermore, the authors may wish to clarify whether the therapy was effective in these specific domains. This information would enhance the clinical interpretability of the results and help define the therapeutic window for CD19-CAR T-cell therapy in systemic sclerosis. It is acknowledged that some patients may present with multiple organ domains affected, but domain-specific outcomes remain essential for guiding future patient selection.
- 3) In general, in all the representative figures, please indicate the baseline characteristics of the patient shown (in the shortest form: patient number, more verbosely: details about organ involvement and activity). Please, exclude patients listed in Major comment #1 and reformat the figure, if necessary.
- 4) Effects of CD19-CAR T-cell therapy on fibroblast function and phenotype (Figure 2E–2F and Results): The numbers of samples across time-points are uneven (BL n=7; 1-month FU n=8; 6-months FU n=4; 12-months FU n=4). The authors should justify this choice. More importantly, for robust longitudinal assessment it would be preferable to restrict the analysis to patients with repeated measures across all time-points. Alternatively, if the group-level statistics are retained as currently shown, the addition of individual trajectories (e.g., spaghetti plots or paired plots) where available would provide crucial insight into inter-individual variability and personalized responses to therapy.
- 5) Comment 4 is also applicable to any UMAP-based analysis. Pooling cells from all time-points is acceptable for descriptive visualization and pattern recognition. However, for statistical interpretation, the analysis should prioritize matched longitudinal data or incorporate models that account for repeated measures. The authors are encouraged to complement pooled UMAP plots with individual-level trends wherever possible.

Minor

- 1) Page 4, rows 98-99 “We hypothesize that B-cell driven autoimmunity may support this remodeling of the skin in SSc [16-20].” The authors may wish to briefly summarize the reference and their hypothesis and to better explain why this mechanism would be relevant in the present context.
- 2) Effects of CD19-CAR T-cell therapy on fibroblast function and phenotype: The authors present pulmonary FAPI PET data from a single patient (Figure 2A) to illustrate the reduction in fibroblast activation. Was this patient affected by progressive interstitial lung disease at baseline? To strengthen the interpretation, the authors may consider quantifying pulmonary FAPI uptake across the cohort and presenting group-level statistics, if available.
- 3) Effects of CD19-CAR T-cell therapy on fibroblast function and phenotype: Supplementary Figure 2 presents valuable single-cell data across different time points. However, it would be helpful to summarize these findings using descriptive statistics (e.g., means, standard deviations) and to perform group-level statistical comparisons. This would enhance interpretability and allow readers to better assess the consistency and significance of the observed changes.
- 4) Rows 328-333: despite R values, p values are merely explorative. This seems due to the small sample size, yet it is advisable to exercise caution in the interpretation of these findings.
- 5) Why capillaroscopy was not re-performed at 12 months? That would have given further insights about long-term effects of CART-T therapy. The lack of functional data beyond 3 months also limits the long-term interpretation and clinical implication of results. These aspects should be well underlined and discussed.
- 6) The increase in vessel fragment length and CD31⁺ cell counts may reflect structural changes rather than true neoangiogenesis.
- 4) IMC has been employed to validate fibroblast and endothelial cell populations at the protein level, no equivalent validation has been performed for epithelial cells. This methodological gap raises concerns regarding the robustness of the functional subdivision of epidermal cell types derived solely from cISH. Consequently, the interpretation of epidermal remodeling should be considered with caution, and this limitation ought to be explicitly acknowledged in the results section.
- 5) While the study provides compelling transcriptomic and spatial evidence of epidermal remodeling, further investigation using (animal) functional model may be warranted to validate the regenerative potential of epithelial changes. The authors may wish to elaborate on how the observed epidermal dynamics could inform future mechanistic studies or therapeutic strategies.

(Remarks to the Author)

The manuscript describes changes in the skin (and in a limited fashion also in the lung and heart) after CAR-T therapy in patients with systemic sclerosis. Multiple sophisticated approaches are applied and strongly suggest an effect of the treatment on skin at the level of pathology and molecular gene expression - detected at both bulk and single cell levels. As such, this represents the best manner for pursuing a translational analysis of a treatment effect for early-stage clinical development. The analytical approaches are also sound. Unfortunately, the fibroblast populations are not identified accurately leading to some faulty interpretation of the results. This issue is described in detail below and makes it difficult to follow the remainder of the manuscript. I have some other relatively minor concerns.

Supplemental Table 1 shows standard of care and natural disease course patients but the difference between these groups is unclear until reading the supplemental methods-it would be good to point to this in the supplemental table legend. It's great that they authors have included a control group but it's worth noting at some point that the control group has, as a group, lower skin scores.

Supplementary Figure 1 is confusing and does not convincingly show depletion of B cells in the skin. Patients 2, 5 and 6 appear on two graphs of CD20+ cells, with different scales that do not match across the graphs. This is a very important point. Past studies have shown that mononuclear cell including B cell infiltration in scleroderma skin is quite variable and B cell numbers minimally increased in the majority of skin biopsies. This appear to be the case here, where most of the skin appear to have minimal numbers of B cells and only two of the biopsies (patients 2 and 4) showing largely increased numbers, and only one of these show depletion in the right most graphic (patient 2).

Figure 1D suggests that B cell depletion has occurred. However, the increase in tissue remodeling and extracellular structure organization suggests that matrix gene expression might paradoxically be increased. However, Supplemental Figure 4 shows that key matrix gene expression associated with scleroderma, i.e. COL1A1, COL1A2, CTHRC1 and SERPINE1 are changed after treatment Unfortunately Suppl Figure 3 and 4 lack a legend so that the direction of change is not discernable. If red is higher, then these genes are showing increased expression after treatment—a surprising result. Supplemental data indicates these genes are decreased after treatment. Please indicate the number of skin samples included in the RNA-seq/GSEA analysis.

Please provide the genes making up the “CAR T response score”.

Myofibroblasts are typically seen first in the deep reticular dermis. ACTA2 and TAGLN + cells are likely marking the dermal sheath cells. Although ACTA2 is indeed a marker for myofibroblasts, it is more highly expressed by dermal sheath fibroblasts and smooth muscle/pericytes. The cells expressing FAP, COMP, SFRP4 and COL8A1 are the myofibroblasts (Nat Commun 12, 4384 (2021)). The TNN, COCH cells likely are also expressing CRABP1, a well-documented marker for dermal papilla. APCDD1 is a marker for papillary fibroblasts (or the major population of these fibroblasts). Superficial fibroblasts should probably be referred to as papillary fibroblasts (Journal of Investigative Dermatology Volume 138, Issue 4, April 2018, Pages 811-825). These designations are more consistent with their beautiful spatial localization data (Suppl Figure 5) which clearly shows the cells they refer to as myofibroblasts around hair follicles and blood vessels, Whereas the FAP+COL8A1+ cells are located in the deep dermis. The diffuse staining of PI16+ cells in the dermis is consistent with its expression by the most common SFRP2+ fibroblast population, which appear adjacent to the FAP+COL8A1+ myofibroblasts (Figure 2E) and are likely progenitors of myofibroblasts. Unfortunately, ref 33 paper lacks a significant number of myofibroblasts, making this a poor reference for identification of this important fibroblast population. Supplemental figure 5 shows the localization of these populations consistent with the descriptions.

The manuscript emphasis and results and discussion need to be re-considered in view of the proper identification of fibroblast subpopulations and clarification on the effect on matrix and biomarker gene expression. It appears that the message will change to one indicating that CAR-T depletion is associated with decreasing numbers of myofibroblasts and possibly matrix gene expression.

Reviewer #4

(Remarks to the Author)

This study investigates whether deep B-cell depletion via CD19-targeted CAR T-cell therapy can not only halt but also reverse tissue fibrosis in systemic sclerosis (SSc). The authors performed sequential forearm skin biopsies in 11 patients with diffuse cutaneous SSc before treatment and at 1, 6, and 12 months after CAR T infusion, with additional biopsies from control SSc cohorts (receiving standard-of-care therapy or observed over the natural disease course). Remarkably, CAR T therapy induced structural regeneration of the skin: the density and height of dermal papillae (rete ridges) in SSc patients significantly increased, approaching levels seen in healthy skin. This regrowth of dermal papillae was accompanied by reduced dermal fibrosis – collagen fibers became less aligned (a sign of fibrosis regression) – and by improvement in clinical skin scores (mRSS) over 12 months. Multi-modal molecular analyses further showed that CAR T treatment shifted the tissue toward a more “healthy-like” phenotype: pro-fibrotic gene signatures were downregulated, while pathways for angiogenesis, extracellular matrix remodeling, wound healing, and epidermal development were upregulated in post-treatment skin. In summary, the key finding is that profound B-cell depletion via CD19 CAR T cells led to an unprecedented reversal of SSc-associated skin fibrosis, with restoration of more normal skin architecture, cellular composition, and gene expression profiles. Overall, the manuscript is very well written, and the data strongly support the authors' conclusion that CAR T therapy can remodel fibrotic skin toward a regenerative state.

However, some issues, limitations, and questions remain:

1. Introduction

a. The authors state that this is the first time an intensive therapy has restored near-normal skin anatomy. This claim should be made more cautiously, as some studies have suggested that autologous stem cell transplantation (ASCT) or other therapies can also lead toward normalization of scleroderma skin histology. For example:

- i. Autologous hematopoietic stem cell transplantation modifies specific aspects of systemic sclerosis-related microvasculopathy. PMID: 35368373
- ii. Autologous hematopoietic stem cell transplantation reverses skin fibrosis but does not change skin vessel density in patients with systemic sclerosis. PMID: 26300240
- iii. B cell depletion therapy upregulates Dkk-1 skin expression in patients with systemic sclerosis: association with enhanced resolution of skin fibrosis. PMID: 27208972

Therefore, the authors should be more cautious in stating that current medical therapies do not halt disease progression or induce tissue regeneration. It may be that previous therapies have not been studied as deeply as in this work, and while CAR T cells might indeed perform better than other intensive treatments, that remains to be proven.

b. Provide additional context on B cells in SSc (both circulating and in tissues). For instance, briefly summarize how B cells contribute to SSc pathogenesis – e.g., through production of pro-fibrotic cytokines, autoantibodies, or interactions with fibroblasts – to frame why targeting B cells could impact fibrosis.

2. Methods & Results

a. Sample size and data handling: The sample size is inherently limited ($n = 11$ treated patients), as this is a pilot study in a rare disease. While understandable, this does impose caution – statistical power is modest, especially for subgroup analyses at later time points (only 4 patients had 12-month biopsies). The data presentation raises some questions. The authors report 11 patients at baseline, 10 at 1 month, 6 at 6 months, and 4 at 12 months (Figure 1). Are the 4 patients with 12-month biopsies a subset of those evaluated at earlier time points? If so, is it valid to analyze longitudinal changes without restricting to the 4 patients who had data across all time points? The manuscript should clarify how missing data were handled. For example, if a patient missed the 6-month biopsy, were they excluded from the 6-month vs. baseline comparisons? The authors must clarify the biopsy sampling and analysis plan. It would help to explicitly state how many patients had biopsies at each follow-up and whether analyses (e.g., comparisons to baseline) are based on paired samples. For instance, the Figure 1 legend indicates $n=11$ at 1 and 6 months, but $n=4$ at 12 months, whereas the spatial transcriptomic data at 12 months included 3 samples (Fig. 3). These discrepancies should be explained (perhaps some 12-month samples were used for histology but not for spatial transcriptomics due to quality or timing differences). Including a simple statement in the Methods or Results about how many patients contributed to each dataset and why certain data are missing would prevent confusion.

b. Baseline characteristics: The study included predominantly severe diffuse cutaneous SSc patients (Supplementary Table 1), likely young- to middle-aged adults with high mRSS and significant internal organ involvement. The authors should briefly note any key baseline similarities or differences between the CAR T group and the control cohorts (e.g., average mRSS, disease duration) to contextualize the outcomes. If any significant baseline imbalances existed (for instance, if the CAR T patients had more severe skin scores or shorter disease duration than the standard-of-care group), these should be mentioned as they could affect the interpretation of the results.

c. Analytical approaches and cell composition: The analytical approaches (immunohistology, spatial transcriptomics, high-dimensional cytometry, etc.) are appropriate and state-of-the-art. The GSEA results clearly support the narrative of decreased B-cell/plasma-cell activity and increased tissue regeneration pathways after therapy. The longitudinal analysis of cell composition changes (comparing baseline vs. 12-month samples, with intermediate time points analyzed in Supplementary Fig. 7) is presented clearly, showing significant decreases in fibrotic and inflammatory fibroblast subsets and increases in pro-regenerative cell types. One suggestion: if not already done, the authors could perform paired (within-patient) analyses for cell composition changes to account for baseline differences between patients, which would strengthen the findings.

d. “CAR T response score”: Using spatial transcriptomics (e.g., RNA in situ hybridization), the authors computed a “CAR T response score” for each cell, based on a signature of differentially expressed genes from the bulk RNA data. This metric is used in the analysis, but the manuscript should clarify how this “score” is defined and calculated, and what a high or low score signifies biologically. Providing a clear definition of the score would help readers better understand these results.

e. Serological and immunological data: It would be informative to include or mention serological data related to B-cell activity. For example, did autoantibody levels (such as anti-Scl-70) change over the course of therapy? Also, were B cells starting to reconstitute by 12 months post-CAR T? Even a brief note on these points would give a fuller picture of the systemic immunological “reset” achieved and help gauge the durability of response.

f. Patient-level response variability: Consider providing patient-level response data (perhaps in a table or supplementary figure). Summarizing each CAR T-treated patient’s clinical course – including mRSS changes, any improvement in organ involvement, and relevant laboratory values – would allow readers to appreciate the variability or consistency of responses. Did all patients show skin improvement, or were there any partial responders/non-responders? If a particular patient had less

improvement, what was distinctive about their case (e.g., longer disease duration, lower CAR T expansion, different autoantibody profile)? Detailing such individual outcomes could yield insights into predictors of response.

g. CAR T-cell expansion and persistence: If data on CAR T-cell counts in blood (e.g., from flow cytometry or PCR) are available, the authors should mention whether there was any correlation between CAR T expansion/persistence and clinical outcomes. For instance, did patients with higher CAR T-cell expansion or longer persistence tend to have greater skin score improvements? Such correlations, if present, would support the mechanistic role of CAR T cells in driving the observed effects.

h. Durability of response: The follow-up duration was up to 12 months for some patients (shorter for others). The manuscript should comment on the durability of the clinical and histological improvements. Have any patients experienced relapses or SSc flares after treatment within the follow-up period? Noting this (even qualitatively) would inform how sustained the benefits of CAR T therapy are, pending longer observation.

i. Transient inflammation and safety: The gene expression data showed that inflammatory pathways (e.g., interferon signaling) were elevated at 1 month post-CAR T but had normalized by 12 months. Could this transient early inflammation be related to CAR T-cell activity (for example, cytokine release during B-cell depletion)? Was it clinically significant? It would be helpful for the authors to note that, indeed, most patients experienced only mild cytokine release syndrome (CRS) after CAR T infusion. Published experiences with CAR T in autoimmune diseases (e.g., SLE) have also reported predominantly grade 1 CRS. Clarifying that any early inflammatory signal did not translate into serious clinical adverse events would reassure readers about the safety profile of this approach.

j. Disease subsets and timing: All patients in this study had diffuse cutaneous SSc (most were Scl-70 or RNA polymerase III autoantibody positive). The authors might discuss whether the outcomes could differ in other subsets of SSc. For instance, would limited cutaneous SSc (centromere-positive disease) respond similarly, or might it have a different trajectory? Also, might disease duration influence the degree of reversibility (e.g., could long-established fibrosis be less reversible)? While the current study may not have data to answer these questions, acknowledging them could highlight important considerations for future applications of CAR T therapy in SSc.

3. Discussion

a. The Discussion section currently reads as largely a recap of the results, with relatively little exploration of mechanisms or integration with existing literature. Rather than simply repeating the findings, the authors should expand the discussion to interpret the results in depth and to propose mechanistic hypotheses. This will better contextualize their findings within the broader understanding of SSc pathophysiology and therapy.

b. B-cell depletion and fibrosis reversal: A major point for discussion is that CD19 CAR T cells primarily act by depleting B cells, yet the manuscript provides limited discussion on how B-cell removal leads to fibrosis reversal. The authors should delve deeper into potential mechanisms. Do they think the CAR T cells acted mainly by depleting autoreactive B cells in the circulation, or did they also eliminate B cells within the affected tissues? (It is not clear how many B cells were present in skin biopsies before and after treatment, and whether CAR T cells themselves trafficked to the skin – the authors could clarify this if data are available.) The discussion should consider how B-cell depletion might translate to antifibrotic effects. For example, does the removal of autoantibody-producing B cells and plasmablasts alleviate chronic inflammatory signaling (such as interferon or TGF- β) in the tissue microenvironment? The data showed a transient spike in interferon-responsive genes at 1 month (perhaps due to initial immune activation by CAR T cells) that subsided by 12 months, consistent with an early inflammatory response that later resolves. A more detailed discussion of these dynamics would be valuable. The authors might hypothesize, for instance, that B cells produce cytokines and growth factors that sustain myofibroblasts, and that their elimination creates a milieu permissive for fibrosis resolution and tissue regeneration. Additionally, the observed increase in Schwann cells in regenerating skin is intriguing – the authors could comment on the potential interplay between nerve fibers and fibrosis (neurogenic factors in tissue repair) as part of the regeneration process.

c. The authors should further hypothesize how, mechanistically, B cells drive fibrosis and how their depletion reverses it. Possible mechanisms to discuss include: autoantibody-mediated stimulation of fibroblasts or endothelial cells; B cell production of pro-fibrotic cytokines (like IL-6, IL-13, or TGF- β) that directly activate fibroblasts; and roles of B cells in modulating T cell or macrophage phenotypes toward pro-fibrotic states. By framing the findings in terms of these known mechanisms, the discussion would better link the B-cell depletion to the observed downstream tissue changes.

d. Immune reconstitution after B-cell depletion: The authors should also consider the fate of the immune system after B-cell removal. What “fills the void” after pathogenic B cells are eliminated? The authors might draw parallels to recent lupus CAR T trials, where B-cell aplasia lasted only a few months and was followed by reemergence of naive B cells and a reset immune repertoire, without return of disease. In SSc, it would be interesting to know if new B cells began to return by 12 months, and if so, whether they appeared less autoreactive (though this may be beyond the scope of the current data). The transient nature of CAR T-induced B-cell depletion (lasting on the order of 3–4 months in other reports) suggests that the therapy provides an immune “reset” rather than permanent B-cell eradication. The discussion could speculate on how the immune system might reconstitute in a healthier balance post-CAR T, potentially with restored tolerance.

e. Comparison to HSCT: It would strengthen the discussion to compare CAR T therapy with autologous hematopoietic stem cell transplantation (HSCT), which is the most intensive therapy currently used for severe SSc. HSCT can induce remission of skin fibrosis and improve survival, but at the cost of high treatment-related mortality and significant toxicity due to complete

immune ablation. In contrast, CD19 CAR T-cell therapy targets only B cells and thus far appears to have a much more favorable safety profile (in this study and others in autoimmunity, there were no treatment-related deaths and only mild CRS, etc.). The authors could point out that CAR T might achieve some of the immune “reset” benefits of HSCT (halt of autoimmunity and fibrosis reversal) without the same risks, positioning CAR T as a potentially safer alternative to HSCT in the future for patients with refractory SSc.

f. In summary, the Discussion should be revised and expanded substantially. The authors need to go beyond restating their results and engage with these mechanistic questions and comparisons to existing literature. A more in-depth discussion, as outlined above, will better highlight the significance of their findings and address potential concerns or curiosities a reader might have.

Version 1:

Reviewer comments:

Reviewer #1

(Remarks to the Author)

The authors have addressed all my points.

Reviewer #3

(Remarks to the Author)

The authors have addressed most of my concerns. I personally don't find the designation of “universal fibroblast” a useful term but accept that there is not yet a consensus on how to name the fibroblast populations in the skin.

I have a few minor comments.

In Supplementary figure 2 it is now clear that most of the patients' studied had 0 B cells in their skin at baseline. I don't believe it diminishes the impact of the observations, but it suggests that local tissue B cell depletion is not behind the observed clinical course. It also raises the question of whether the two patients with B cells responded any differently. I realize this is a small study, but it is relevant to understanding whether B cell depletion is working locally or systemically. Tissue B cell depletion in skin was seen also with rituximab without an evident clinical response (PMC2637937), suggesting that it is the deletion of autoantibodies or other systemic effect that is driving the therapeutic benefit.

The cluster of cells expressing PU.1/SPI1 are most likely macrophage-fibroblast doublets (see supplemental file 4 in Tabib T et al PMID: PMC8289865). This is strongly supported by panel 2C showing that the fibroblasts are not forming a discrete cluster. PU.1/SPI1 is a clean marker of macrophages and known to regulate their differentiation. S100A4 is expressed more diffusely by both macrophages and all fibroblast subpopulations. The FLEX scRNA-seq technology applied to tissue is known to lead to larger populations of doublets, which explains the high number of macrophage doublets in the fibroblast cluster. This is a small point but will be confusing to macrophage experts. In my lab algorithms designed to remove these doublets (DoubletFinder, scDbtFinder etc) have unfortunately not enabled removal of these cells. So, my suggestion is to mention this in the text and to otherwise ignore these cells. For future studies, it may help to limit formalin fixation to overnight and then place biopsies in 70% ethanol and process for single cell studies as soon as feasible.

Although the authors indicate that Supplemental Figure 3 and 4 heatmaps have been removed, the bottom panel of figure 4G and supplemental Figure 5 are heatmaps that look similar and still lack a legend to show what color is up and what is down. If red is indicating higher expression, then it does seem odd that the collagen genes are upregulated after treatment. This appears particularly prominent in the 12-month samples. The authors explanation appears reasonable that fibroblasts numbers may be increasing during recovery. But if the authors are going to show these heatmaps they must provide the legends for interpretation even if it is not particularly supportive of the other data.

Reviewer #4

(Remarks to the Author)

The authors have satisfactorily addressed all of my comments.

**A-Response to the editorial comments:**

**Editorial comment (e-mail 01.10.2025):** Thank you again for submitting your manuscript "Deep
phenotyping of tissue remodeling in the skin of systemic sclerosis patients in response to CD19-CAR T-
cell therapy" to Nature Communications. We have now received reports from 4 reviewers and, after
careful consideration, we have decided to invite a major revision of the manuscript.

As you will see from the reports copied below, the reviewers raise important concerns. We find that
these concerns limit the strength of the study, and therefore we ask you to address them with
additional work.

In particular, it is essential to re-visit all your analyses to ensure correct identification of fibroblast
populations, as highlighted by reviewer 3. The conclusions of the study, as well as the discussion, will
need to be updated accordingly. It is also critical that you ensure adherence to all patient criteria set
forth in the study protocol, as noted by reviewer 2. In this regard, I also note that disease duration is
said to have been limited to 7 years in the manuscript, but 5 years in the study protocol. Please clarify
this point in the Methods section.

Please note that we do not find any point raised by our reviewers more important than the rest, and
we strongly encourage you to address all concerns with additional data and text edits, as specified by
the reviewers. Without substantial revisions, we will be unlikely to send the paper back to review.

Author response (in blue throughout the response letter):

Thank you for your careful evaluation of our manuscript, "*Deep phenotyping of tissue remodeling in*
*the skin of systemic sclerosis patients in response to CD19-CAR T-cell therapy,*" and for the opportunity
to re-submit it in revised form. We appreciate the effort invested by you and the reviewers in providing
detailed and constructive feedback that helped us to strengthen the manuscript. We have carefully
considered all comments raised by the four reviewers.

In the revised version, we re-analyzed all transcriptomic datasets with a new annotation workflow,
incorporating updated reference atlases. We revised all the respective figures, results, and
interpretations in the discussion section accordingly.

As noted by reviewer 2, we re-ensured that all patient criteria requested in the study protocol were
fulfilled by all participants and explained it on p.10, ll. 310ff. of this response letter. In response to the
editor's comment on the study protocol, we added the information on a study protocol amendment
(EudraCT-Nr. 2022-001366-35 CSP version 5.0 / 28.09.2023), that was approved on the 28.09.2023 and
included the extension of the disease duration since the first-non-Raynaud-phenomenon from "5
38 years" in the initial protocol to "7 years" in the amended protocol. This information is now added to
39 the Online Methods section (Online Methods, p.3, ll. 66-70). The present proof-of concept study
includes samples of patients who received CD19-CAR T-cell treatment within named patient use or the
CASTLE trial, in addition samples of patients who had received the state-of-the-art treatments as
recommended by national and international guidelines and an exploratory set of sample of a patient
who had stabilized without treatment were added as controls. This point is explained in more detail in
response to major comment 1 of reviewer 2 (p 10., ll. 310 ff. of this response letter) and in the methods
section (p.2., ll.39 ff., Online Methods section).

Beyond these key points, we address *all* reviewer's comments in a detailed point-by-point response
below. Revisions made to the manuscript text are highlighted in blue.

Editorial requests:

**Important:** In addition to the above, you must comply with the following editorial requests; we will
not be able to proceed with your revised manuscript otherwise. Please also see the *Nature*
*Communications* formatting instructions, which you may find useful while preparing your revised
manuscript. Please also ensure that you comply with our editorial policies.

We implemented the editorial formatting instructions as requested.

**POLICIES AND FORMS REQUIRED FOR RESUBMISSION**

* Please complete or update the following checklist(s) to verify compliance with our research ethics
and data reporting standards. Address all points on the checklist, revising your manuscript in
response to the points if needed.

The form(s) must be downloaded and completed in Adobe Reader rather than opened in a web
browser. Each form must be uploaded as a Related Manuscript file at the time of resubmission.

Reporting summary:

<https://www.nature.com/documents/nr-reporting-summary.pdf>

We added the reporting summary as requested as a separate file.

* Nature journals have recently announced an update to our guidance on reporting on sex and
gender in research studies (see here). We strongly encourage researchers to follow the 'Sex and
Gender Equity in Research – SAGER – guidelines' and to include sex and gender considerations for
studies involving humans, vertebrate animals and cell lines where relevant to the topic of study (an
overview can be found here). Authors should use the terms sex (biological attribute) and gender
(shaped by social and cultural circumstances) carefully in order to avoid confusing both terms.

When preparing your revised manuscript, please be aware of our guidance on Sex and Gender
reporting).

As requested, we double checked the manuscript for adherence to the SAGER guidelines and
guidance on Sex and Gender reporting.

Please note that we require that the following recommendations from the guidelines are followed:

1. If the research findings apply to only one sex or gender, that must be indicated in the title and/or
abstract.

Not applicable.

2a. For studies involving vertebrates animal and cell lines- The Reporting Summary should include
whether sex was considered in the study design.

Not applicable.

2b. For studies involving human research participants- The Reporting Summary should include
whether sex and/or gender was considered in the study design and whether sex and/or gender of
participants was determined based on self-report or assigned (and methodology used).

While females are at a higher risk of autoimmune diseases including systemic sclerosis, male sex is a
risk factor for a progressive disease course¹. This is reflected in the study cohort with 36% of
participants being female and 65% being male, based on self-reported sex and as documented in the
patients' health insurance information.

As requested, the respective information was added to the reporting summary.

3. Data should be reported disaggregated for sex and gender where this information has been
collected and consent has been obtained for reporting and sharing individual-level data;
disaggregated numbers for individual experiments must be provided in the source data as
appropriate whereas overall numbers may be provided in the Nature Portfolio Reporting Summary.

We added the analysis of dermal papillary quantifications as central outcome disaggregated by sex.
Herein, no significant sex-related differences were detected (see source data file). Based on this lack
of differences and given the sample size, data from both sexes were combined for all subsequent
analyses. This information was added to the reporting summary, the source data file and the results
section of the manuscript (p.6, ll. 165 ff.):

*“No significant sex-related differences were detected in dermal papillary measurements (data*
*shown in the source data file). Therefore, given the exploratory sample size, data from both*
*sexes were aggregated for all further analyses.”*

Information on the points above should be included in the revised manuscript and detailed in the cover
letter.

As requested, information on the points above was included in the revised manuscript and detailed
in the cover letter.

In addition, please note that if sex- and gender-based analyses have been performed a priori, results
should be reported regardless of positive or negative outcome. We discourage conducting post hoc
sex- and gender-based analysis if the study design is insufficient (for example, low sample size) to
enable meaningful conclusions.

If no sex- and gender-based analyses have been performed, please indicate the reasons for the lack
of these analyses in the Reporting Summary.

We addressed sex-and gender-based analyses as detailed in response to the previous comments 1-3.

**DATA AND CODE AVAILABILITY**

* All Nature Communications manuscripts must include a “Data Availability” section after the
Methods section but before the References. If any of the data can only be shared on request or are
subject to restrictions, please specify the reasons and explain how, when, and by whom the data can
be accessed. For more information on this policy and a list of examples, see:

<https://www.nature.com/documents/nr-data-availability-statements-data-citations.pdf>

* Please also include a “Code Availability” section after the “Data Availability” section. If the code can
only be shared on request, please specify the reasons. For more information on our code sharing
policy and requirements, please see:

[https://www.nature.com/nature-portfolio/editorial-policies/reporting-standards#availability-of-](https://www.nature.com/nature-portfolio/editorial-policies/reporting-standards#availability-of-computer-code)
[computer-code](https://www.nature.com/nature-portfolio/editorial-policies/reporting-standards#availability-of-computer-code)

We added a "data availability statement" and a "code availability statement" as requested (Main
manuscript, p.25, ll. 631 ff).

* To maximise the reproducibility of research data, we ask that you provide a Source Data file
containing the raw data underlying the following types of display items:

- Any reported means/averages in box plots, bar charts, and tables

- Dot plots/scatter plots, especially when there are overlapping points

- Line graphs

- Uncropped and unprocessed scans of all blots and gels including all quantified replicates. The edge
of membranes, molecular weight ladders and loading controls should be presented on all blots.

Where membranes have been cut, please ensure that at least one marker above and below is
present.

The data should be provided in a single Excel file with data for each figure/table in a separate sheet,
or in multiple labelled files within a zipped folder. Name this file or folder 'Source Data', and include a
brief description in your cover letter. The "Data Availability" section should also include the
statement "Source Data are provided with this paper."

We added the source data file as requested.

**AUTHOR CHANGES ON REVISION**

If there are any changes to the author list in the revised manuscript, please use this approval form
www.nature.com/documents/nr-author-list-change-form.pdf, arranging for all authors on your paper
to sign the statement confirming that they agree to the author list being changed, and add this
document to your resubmission.

An author changes form was added. The following authors were added to the manuscript: Ziyuan Liu,
Andrea Zoli, Paula Gehringer.

**B-Response to the reviewers' comments**

**Reviewer #1 (Remarks to the Author):**

The manuscript presents an important and timely investigation into the effects of CAR T therapy on
skin pathology, with a focus on histological and molecular changes over the course of treatment and
follow-up. The topic is highly relevant, and the authors employ a combination of imaging, histology,
and single-cell profiling to explore these dynamics. The breadth of data, including multiple time
points and supplementary figures, provides a potentially rich resource for understanding disease
progression and treatment effects.

While the study is ambitious and addresses a clinically meaningful question, certain aspects of the
presentation and figure interpretation require clarification to ensure that the results can be
accurately interpreted by readers.

We thank the reviewer for his/her encouraging comments.

Figure 1 and 5 - Images of CAR T and NDG skin at baseline look different from each other, especially
in the papillary dermis. This impairs an immediate comparison of the skin changes following CAR T
treatment with the skin modifications happening during the natural course of the disease.
Obviously, they are taken from different patients and may have been taken at different stages of the
disease according to Supplementary Table 1. The NDG baseline shown here is from the patient with 5
192 months disease duration or 11 years? Moreover, 12 months FU and SOC images have a different
intensity of blue staining: are they thinner sections?

We thank the reviewer for his/her comment.

The image previously shown for the NDG group was from patient 2 (11 years disease duration), yet
with more preserved dermal papillae. As suggested, we exchanged the image and inserted images of
NDG patient 1, whose disease duration matched the treatment groups. As further detailed in response
to major comment 1 of reviewer 2 (p.10, ll. 310 ff. of this revision letter), we removed NDG patient 2
from the manuscript due to the disease duration.

The different intensities of blue staining of the 12-month FU and SOC images might be explained by
the issue that the sections were stained in two series. We re-stained all sections and changed the
images.

Figure 1A and figure 5A now look as follows:

**Figure 1A:** Representative images of Masson-Trichrome staining are shown (CAR T patient 5, NDG
patient 1 and SOC patient 4).

**Figure 5A:** Representative images of Masson-Trichrome staining are shown (CD19-CAR T patient 3,
NDG patient 1, SOC patient 2).

Figure 2D is not clear. Does the left image represent the spatial distribution of fibroblast density at
baseline?

We thank the reviewer for his/her comment and clarify that the image on the left side of former figure
2 D visualizes the spatial distribution of the fibroblast density at 1 mo FU as reference for the spatially
resolved CAR T response score at 1 mo. This is now explained in the figure legend:

**Figure 1 C** Kernel density maps showing the spatial distribution of fibroblast density and CAR T-cell
response score in the skin tissue obtained from patient 9 at 1-month follow-up (1-mo FU).

Supplementary Figure 1: representative images of immunofluorescence staining of CD19+ and CD20+
B cells in the skin should be shown.

We thank the reviewer for his her/comment. Immunohistochemistry of CD19+ and CD20+ was
performed and representative images are now visualized in Supplementary Figure 2 as shown below.
We apologize that CD3+ counts were mistakenly labeled as CD20+ cells in the previous version of the
figures. This error has been corrected.

Former Supplementary Figure 1, current Supplementary Figure 2, looks like:

**Supplementary Figure 2: A** Individual course of the numbers of CD19+, CD20+ B-cells and CD3+ T-
 cells before and after CD19-CAR T-cell treatment as analyzed by immunohistochemistry. **B C D**
 Representative immunochemistry staining images of the CD20 (B), CD19 (C) and CD3 (D) for the four
 timepoints. BL = Baseline; 1m = 1 month; 6m = 6 months; 12m = 12 months. All representative
 images are from patient 5 of the CD19-CAR T group.

Supplementary Figure 2 A: does the UMAP plot depict clustering of major cell types detected by cISH
at baseline or at 12 months FU? – figure legend

We apologize for the lack of clarity of former Supplementary Figure 2, now Supplementary Figure 3.
The UMAP (Supplementary Figure 3A) shows initial annotation of cells from all the samples detected
by cISH, including baseline and follow-up visits, which is now clarified in the figure legends: “UMAP
plot (right) depicts clustering of major cell types detected by cISH, including BL and all follow-up
samples (n = 23 samples).”

The changing trends of individual fibroblast populations were visualized as further detailed in
response to comment 4 of reviewer 2 (p.14, ll. 447 ff.) of this response letter and supplementary
figure 10B.

Supplementary Figure 15: does it refer to epithelial populations detected in skin upon CAR T
treatment? At which time point FU?

As requested, we clarified the figure legend of former Supplementary Figure 15, now Supplementary
Figure 18: “UMAP plot shows clustering of epithelial cells detected by cISH, including BL and all follow-
up samples (n = 23 samples). The color represents the identity of the epithelial population.”

Finally, I suggest removal of 20 years' old reference 1 since it is quite outmoded and replaceable with
updated reviews based on novel clinical and pathogenic concepts and study methods.

As suggested, we updated the references by more recent reviews: PMID: 36442487², PMID:
39953141³ and PMID 28413064⁴.

**Reviewer #2 (Remarks to the Author):**

This is a worthwhile and timely study, addressing a long-standing unmet need in systemic sclerosis:
the possibility of reversing fibrotic tissue damage through targeted immunotherapy. The work is rich
in content and provides numerous insights into the possible cellular and molecular remodeling of the
skin following CD19-CAR T-cell therapy. The integration of transcriptomic, proteomic, spatial, and
imaging data is impressive and adds depth to the interpretation.

However, the exploratory nature of the study should be more explicitly acknowledged. Several
analyses rely on small and uneven sample sizes, and statistical robustness is limited in many
comparisons. While the findings are promising, they should be interpreted with caution and framed
as hypothesis-generating. Clearer articulation of the study's scope and limitations would help
contextualize the results and guide future confirmatory research.

We thank the reviewer for his/her encouraging comment. We amended the introduction to articulate
the scope of the study more clearly. In addition, we explicitly stated the exploratory nature of the study
in the discussion and amended the section on limitations. The respective passages now read as follows:

Introduction, p.5.l. 137 ff.: *“The aim of the present study was to provide a proof-of-concept for*
*skin tissue changes on the histological, molecular and cellular level before and after CD19-*
*CAR T-cell therapy by using RNA sequencing, imaging mass cytometry (IMC) and cyclic in situ*
*hybridization (cISH).”*

Discussion, p 17., ll. 421 ff: *“We performed an in-depth exploratory analysis of skin tissue of*
*SSc patients treated with CD19-CAR T-cell therapy and provide first evidence that deep B-cell*
*depletion can affect several pathogenic hallmark processes in SSc: fibrotic tissue remodeling*
*and vasculopathy. “*

Discussion, section on limitations, p.22, ll. 566 ff: *“Acknowledging the novelty of CD19-CAR*
*T-cell treatment as an only recently emerging therapy, the sample sizes are still limited.*
*Moreover, sample sizes are uneven between the different follow up time points due to the*
*consecutive inclusion of the patients. Thus, further investigation and studies in larger cohorts*
*with extended longitudinal data are required to confirm the effects of CD19-CAR T-cells on*
*tissue changes in SSc. The results of the present study may help to generate hypothesis for the*
*further exploration of the effects of CD19-CAR T-cell therapy on different pathogenetic*
*hallmarks and organ manifestations in SSc and support the definition of patient selection and*
*outcomes in future CD19-CAR T-cell related studies.”*

Major

1) According to the stated inclusion criteria, at least one patient in the treatment group
(n.4), two in the SOC group (n.2 and 8), and one in the NDG group (n.2) appear to exceed the
maximum disease duration allowed by criterion #1. Patients n.10 and 11 are also borderline.
Are these numbers correct or there are whatsoever typos? If the provided numbers are
correct, the exclusion of these patients is not trivial, as prolonged disease duration may shift
the pathological balance toward atrophic rather than inflammatory or fibrotic processes,
which are more amenable to immunomodulatory interventions. Despite the compelling
results presented, I remain unconvinced that CD19-CAR T-cell therapy is suitable across all
disease stages and/or for all patients. Strict adherence to inclusion criteria would help isolate
the therapeutic effect in a well-defined subset of patients—those at higher risk of
progression and typically enrolled in randomized clinical trials. Given the substantial burden
and cost of CAR T-cell therapy, clearer guidance on patient selection is essential to ensure
appropriate clinical translation.

We thank the reviewer for his/her comment. We would like to highlight that criterion #1 is
one of four alternative criteria as reflected by the **“either [...] or”** formulation, which is now
highlighted more explicitly in the online methods section (p.2, ll. 48-51). The inclusion criteria
were strictly adhered to and they are fulfilled by all patients included. As also commented in
response to the editor's comments, we now clarify that a protocol amendment was made
during the recruiting period of the CASTLE study, which allowed the inclusion of patients with
disease duration up to seven years, which applies to patients 10 and 11 of the CD19-CAR T-
cell cohort (protocol version CSP version 5.0 / 28.09.2023). This information was added to the
methods section (p.3, ll. 66-70 ff.). Moreover, the inclusion criteria allow the inclusion of
patients who received CD19-CAR T-cell treatment under named patient use after seven years,
if they showed clear signs of progressive disease as reflected by criterion 4. Patients with
disease duration > 7 years and no disease progression, including stable or atrophic disease,
which is clinically distinct from progressive skin and lung disease, were excluded. This is now
clarified in the exclusion criteria. In the CAR T group, patient 4 was included due to ILD

progression after seven years of disease duration. At inclusion, he was 47 years old with an
FVC of 45% and high risk of further deterioration that may require lung transplant. He was
consulted on the alternative options of HSCT evaluation at a transplant center and decided for
CD19-CAR T-cell approach as named patient use first. Since the application of CD19-CAR T-cell
therapy, no more progression of skin and lung disease and even improvements of FVC and
mRSS were documented (supplementary data file 1), suggesting that CD19-CAR T-cell
treatment may also be beneficial in patients with progressive ILD after seven years of disease
duration. This is in line with the reviewer's suggestion to enrich patients with progressive
disease and is reflected in the inclusion criteria. We acknowledge that patients with ILD
progression beyond seven years are hardly represented in clinical trials⁵. Thus, the inclusion
criteria used in this study fulfill an unmet clinical need and may help to refine inclusion criteria
for future studies.

We agree with the reviewer that CD19-CAR T-cell therapy may not be suitable across all
disease stages and in all patients. In the present study, patients with diffuse, progressive SSc
refractory to standard- therapy at an early stage and a patient with progressive ILD > 7 years
were treated with CD19-CAR T cell therapy, thus first conclusions can be drawn on this target
population. However, other disease stages including early or very early SSc, limited disease or
late/stable, late/atrophic disease were not included here and we cannot conclude on these
patient groups based on our results. As requested by reviewer 4, this was added to the
limitations section of the discussion (p.22, ll. 550 ff):

*“The patients investigated in this study had refractory, diffuse cutaneous SSc with a progressive*
*disease course despite multiple treatments. Interestingly, we observed a clinical and molecular*
*response in a patient with progressive disease after seven years of disease duration, which may*
*open avenues for further studies of the subgroup of patients who experience disease progression*
*at later disease stages. However, other subgroups of SSc patients such as early SSc, or limited*
*cutaneous SSc with organ involvement were not included in the target population of this study*
*and may be investigated in future trials. However, such patients can only be treated with CAR*
*T-cells if showing high-level safety.”*

We further investigated whether the patients with a disease duration of more than five years
may mask the effects of CAR T-cell from the rest of the cohort. We performed additional
cellular abundance analysis in our cISH dataset, excluding patient 4, 10, 11 (Response letter
Figure 1, below). The results shown below indicate no major differences in cellular
composition, compared to Figure 2E, when patients with longer disease duration were
excluded.

Response letter Figure 1: Forest plot showing differential composition of non-epithelial component between baseline and 12-month follow-up (12mo FU) by sccomp with donor-specific random intercepts. Points represent the posterior mean estimates for each cell type (colored by cell type), with error bars indicating 95% of credible intervals (2.5%–97.5%). Red bars indicate statistically significant difference in cell proportion (FDR < 0.05).

The inclusion criteria also allow the inclusion of control samples that were acquired from
patients with progressive SSc during standard-of-care procedures. At the current stage, no
controlled CD19-CAR T-cell studies, including a standard of care-treatment arm, are available,
given the challenges of a-blinded control arm for CD19-CAR T-cell therapy. Thus, we feel that
the inclusion of patients from a real-life state-of-the-art group is a reasonable tradeoff. This
procedure is in line with recent FDA recommendations (FDA-2025-D-3403) that refer to study
design for cellular therapy products in small populations (Online methods, p.2, ll. 39-44 ff.).
Patients 2 and 8 of the SOC group received treatment change due to progressive skin and lung
disease, respectively. A file stating indications and treatments applied in the SOC group was
added to the supplements (Supplementary Data File 3).

Relatively little is known on longitudinal histologic outcomes in the natural disease course in
SSc. Thus, as reference, we added samples that had been collected from patients who refused
medication due to relative clinical stabilization upon progression and instead decided to be
closely monitored. We agree with the reviewer that the disease duration of former patient 2
of the NDG group imperfectly matches the inclusion criteria. The sample of his patient was
included during the first version of the manuscript given the difficulty to assess the
spontaneous disease course in dcSSc, however, we have now removed this patient's samples
from the manuscript and all related files.

2) Ancillary to comment #1, it would be helpful to explicitly list the main clinical indications that led
to CD19-CAR T-cell therapy administration, particularly in relation to inclusion criterion #4 (e.g.,
progressive skin involvement, worsening interstitial lung disease, cardiac fibrosis). Furthermore, the
authors may wish to clarify whether the therapy was effective in these specific domains. This
information would enhance the clinical interpretability of the results and help define the therapeutic

window for CD19-CAR T-cell therapy in systemic sclerosis. It is acknowledged that some patients may
present with multiple organ domains affected, but domain-specific outcomes remain essential for
guiding future patient selection.

We thank the reviewer for his/her comment. As requested, we added a table listing the main
indications that resulted in the administration of CD19-CAR T-cell therapy, particularly concerning
inclusion criterion #4 (Supplementary Data File 1). We also expanded the information on organ
involvement at baseline and the summary of organ specific outcomes during the available follow-up in
Supplementary Data File 1: mRSS, forced vital capacity (% predicted) and diffusion capacity of the lung
for carbon monoxide (DLCO, % predicted) and the course of the EUSTAR-activity index as composite
score including several additional disease domains such as digital ulcerations, inflammation (CRP), and
tendon friction rubs⁶. Moreover, as fibroblast function is a central aspect of this manuscript, we show
changes of pulmonary FAPI-uptake over time (Figure 2A). The course of mRSS is depicted in main figure
1C, the changes of lung function parameters are shown in supplementary figure 3D and the change of
pulmonary FAPI uptake is shown in Main Figure 2A. We may comment that the inclusion criteria and
outcomes were not specifically tailored towards the assessment of other organ-specific domains
including gastrointestinal manifestations and myocardial involvement. This is now discussed in the
limitations section of the discussion (p.23, ll. 589 ff.):

*“SSc is a heterogeneous disease with varying organ involvement. Here we present detailed data*
*on the effects of CD19-CAR T-cell therapy on the molecular level in the skin and results of*
*molecular imaging of the lung. However, the inclusion criteria and outcomes of this study were*
*not directed towards the detailed assessment of other organ manifestations such as*
*gastrointestinal system or myocardial involvement. The efficacy in these domains needs to be*
*assessed in future trials.”*

Safety as part of the therapeutic window definition is the primary outcome of the CASTLE study and is
reported elsewhere⁷.

3) In general, in all the representative figures, please indicate the baseline characteristics of the
patient shown (in the shortest form: patient number, more verbosely: details about organ
involvement and activity). Please, exclude patients listed in Major comment #1 and reformat the
figure, if necessary.

As requested, we added the patient numbers of patients included in all representative images, which
is now highlighted in blue in the figure legends. The baseline information on disease duration and
organ involvement and EUSTAR-activity index at baseline are detailed in Supplementary Figure 1 and
in Supplementary Table 1 for reference and Supplementary Data File 1.

We also reformatted the figures 1 and 5 as requested, excluding the patients listed in Major comment
1, which now look as follows:

Figure 1A: Representative images of Masson-Trichrome staining are shown (CAR T patient 5, NDG patient 1 and SOC patient 4).

**Figure 5A:** Representative images of Masson-Trichrome staining are shown (CD19-CAR T patient 5,
NDG patient 1, SOC patient 4).

4) Effects of CD19-CAR T-cell therapy on fibroblast function and phenotype (Figure 2E–2F and
Results): The numbers of samples across time-points are uneven (BL n=7; 1-month FU n=8; 6-months
FU n=4; 12-months FU n=4). The authors should justify this choice. More importantly, for robust
longitudinal assessment it would be preferable to restrict the analysis to patients with repeated
measures across all time-points. Alternatively, if the group-level statistics are retained as currently
shown, the addition of individual trajectories (e.g., spaghetti plots or paired plots) where available
would provide crucial insight into inter-individual variability and personalized responses to therapy.

We thank the reviewer for his/her comment. The samples included in the analysis were consecutively
acquired from the patients who received CD19-CAR T cell therapy either within named patient use or
the CASTLE study. Due to consecutive inclusion, more longitudinal samples are available in patients
who were treated earlier compared to patients who received the treatment more recently. As also
requested by reviewer 4, an overview of all acquired and analyzed samples and timepoints was added
(Supplementary Data File 1_clinical data and sample overview_31012026.xls). In addition, as
requested by reviewer 4, a sample analysis plan and a section on sample handling that also includes
missing data was added to the online methods (Online methods, p. 5, ll. 110 ff. and p.35, ll. 1025 ff. of
this response letter). All samples available at the time of analysis were included in the analysis. The

456 uneven sample size mostly results from the time-shifted inclusion of the patients. Besides, if a sample
 had to be excluded, e.g. due to not passing quality control, this was annotated in Supplementary Data
 File 1_clinicaldataandsampleoverview_31012026 and in the source data file. In this exploratory
 analysis, the most comprehensive view and the highest granularity of data can be achieved by including
 all available information/samples. As suggested by the reviewer, in addition to the current figures 2 D
 and E, we added the individual trajectories to the time-dependent boxplots of the fibroblast
 populations that shifted upon CD19-CAR T-cell therapy. These figures were added to the
 supplementary figures 10B and now look as follows:

 **Supplementary Figure 10B:** Boxplots showing the distribution of fibroblast frequencies at each
 timepoint. The x-axis represents the timepoints, and the y-axis represents the fibroblast frequency for
 the specified subtype. Each dot represents an individual sample, with colors distinguishing patient
 identity; within-patient measurements are connected to depict longitudinal trajectories.

Moreover, we added the frequencies of fibroblast subtypes per patient between different
 timepoints, which is shown in the response letter figure below:

**Response letter Figure 2:** Alluvial plots showing patient-specific changes in fibroblast subtype
 composition across sequential timepoints. For each plot, the x-axis denotes different timepoints and
 the y-axis indicates their proportional frequencies (normalized to 100%). The flows illustrate
 transitions in the relative abundance of fibroblast subtypes, with colors representing distinct subtypes.

5) Comment 4 is also applicable to any UMAP-based analysis. Pooling cells from all time-points is
 acceptable for descriptive visualization and pattern recognition. However, for statistical
 interpretation, the analysis should prioritize matched longitudinal data or incorporate models that
 account for repeated measures. The authors are encouraged to complement pooled UMAP plots
 with individual-level trends wherever possible.

We thank the reviewer for his/her comment.

As a first analysis step, all cells from all time-points have been pooled for easier visualization by UMAP
 as well as better alignment of the unsupervised clustering. On the other hand, the consequent analysis
 has been performed analyzing the resulting data for each patient independently. For statistical
 significance calculations, paired t-test or random effect modeling were used to account for individual-
 specific effects. Supplementary Figure 13 shows the changes in the frequencies of the different IMC-
 defined fibroblast subsets for each individual patient.

A more detailed description has been added in the IMC method section (Online Methods, p.13, ll.320-
 321): *“Subsequently, the frequency of each cluster has been studied for each patient individually. For
 statistical analysis, paired t-test was used to account for the repeated measurements.”*

As described in response to your previous comment, we added the individual trajectories of changing
 trends of fibroblast populations for every patient (Supplementary Figure 10B). Moreover, we

exemplified the UMAP of cISH-identified fibroblast subsets for each individual at baseline and 1-month
 follow-up below. The changing trend of FAP+COL8A1+ myofibroblast at 1-month follow-up are shown:

**Response letter Figure 3:** UMAPs for individual patients comparing fibroblast subtype distributions at
 baseline (BL) and 1-month follow-up (1m FU), mapped onto the shared fibroblast-subtype embedding
 shown in the right panel.

Minor

1) Page 4, rows 98-99 “We hypothesize that B-cell driven autoimmunity may support this remodeling
 of the skin in SSc [16-20].” The authors may wish to briefly summarize the reference and their
 hypothesis and to better explain why this mechanism would be relevant in the present context.

We summarized and amended the references to further explain why B-cell driven tissue remodeling
 is interesting in this context in the introduction section. The respective passage now reads as follows
 (p.4., ll.103 ff.):

“We hypothesize that B-cell driven autoimmunity may be a key driver of cutaneous remodeling
 in SSc, as growing evidence implicates B cells in orchestrating fibrotic tissue remodeling in
 SSc. Dysregulation of circulating B-cells is evident at several levels in SSc, including aberrant
 B-cell receptor (BCR) signaling characterized by enhanced activating and diminished
 inhibitory signals⁸⁻¹⁰ and disturbed effector function resulting in hypergammaglobulinemia
 and autoantibody secretion. Distinct autoantibodies associate with defined clinical SSc
 phenotypes^{3,11}, e.g. anti-scl70 antibodies are associated with a high risk of disease progression
 and inner organ involvement. Other autoantibodies exert direct pathogenic effects: agonistic
 antibodies targeting anti-platelet-derived growth factor receptor (PDGFR) or fibrillin promote
 fibroblast activation^{12,13}, while anti-endothelial cell antibodies, anti-endothelin-1-type A
 receptor, and anti-angiotensin-II type 1 receptor antibodies contribute to endothelial cell

*damage and vasculopathy*^{9,14,15}. *In addition to autoantibody-mediated mechanisms, B-cells*
*shape the fibrotic milieu through the secretion of pro-fibrotic and pro-inflammatory cytokines*
*such as IL6^{16,17}, whereas regulatory B-cell subsets that restrain inflammation, e.g. via IL10,*
*are decreased in SSc¹⁸. B-cells also infiltrate the organs affected in SSc and may mediate*
*tissue remodeling*^{9,19}: *A recent exploratory study using a three dimensional humanized skin*
*model showed, that infiltrating B-cells of SSc patients show a hyperactivated phenotype,*
*increased immunoglobulin production and the capacity to induce profibrotic transcriptomic*
*programs in resident fibroblasts, supporting a direct role for B-cells in tissue-level disease*
*propagation*²⁰.

*The rationale of targeting B-cells in SSc is further supported by clinical trials of the CD20-*
*depleting antibody rituximab (RTX), which have shown improvements in both skin fibrosis and*
*interstitial lung disease (ILD)²¹⁻²³ and exploratory analyses suggest that clinical response may*
*be linked to dermal remodeling of pathways such as the upregulation of dermal DKK-1²⁴.*
*However, despite encouraging results, many patients still progress under conventional B-cell-*
*targeted therapy, highlighting a critical limitation of current B-cell-directed therapies. CD20-*
*targeting antibodies incompletely deplete tissue-resident B-cells²⁵⁻²⁷ and fail to eliminate*
*antibody-producing plasmablasts. While ASCT may overcome this limitation, it carries*
*significant procedure-related risk. We recently demonstrated that CD19-directed chimeric*
*antigen receptor (CAR) T cells achieve deep and broad B-cell depletion across tissues*^{25,26}.
*Consistent with this, we and others have reported profound clinical responses in patients with*
*SSc treated with CD19-CAR T-cell therapy*²⁸⁻³⁰. *Based on these several lines of evidence, we*
*hypothesize that deep depletion of B-cells by CAR T-cell therapy may interfere with the B-cell*
*mediated fibrotic and vascular remodeling, thereby reshaping diseased tissue architecture in*
*SSc."*

2) Effects of CD19-CAR T-cell therapy on fibroblast function and phenotype: The authors present
pulmonary FAPI PET data from a single patient (Figure 2A) to illustrate the reduction in fibroblast
activation. Was this patient affected by progressive interstitial lung disease at baseline? To
strengthen the interpretation, the authors may consider quantifying pulmonary FAPI uptake across
the cohort and presenting group-level statistics, if available.

We thank the reviewer for his/her comment. The representative images shown stem from patient
number 4, as now indicated in the respective figure legend. As visualized in the Supplementary Data
File 1_clinical data and sample overview_31012026, this patient had been suffering from progressive
ILD with constant FVC decrease within five years before the application of CD19-CAR T-cell therapy the
553 years before despite medical therapy with nintedanib, mycophenolate and cyclophosphamide adding
up to a relative FVC decline by 31% within 5 years. As also detailed in response to major comment 1,
this patient had been consulted on the evaluation of lung transplant in a respective center. Also HSCT
was discussed and the patient was recommended to co-evaluation for HSCT in a transplanting center,
although he would not fulfill typical selection criteria due to reduced FVC³¹. The patient decided for
the CD19-CAR T-cell approach and has improved with lung function (FVC) as visualized in the table
since then. In addition to lung function deterioration at baseline, he had pronounced FAPI uptake,
which is associated with a higher risk of further deterioration^{32,33}. FAPI uptake was quantified and the
individual trajectories of FAPI uptake are shown in Figure 1A. During revision, the quantification of
additional FAPI scans were amended of all patients who received CD19-CAR T-cell therapy and

individual trajectories (Figure 2 A) as well as group level statistics (Supplementary Figure 4A) were
 added. The graph now looks as follows:

**Figure 2A:** Quantification of FAPI uptake across the time points are visualized as Spaghetti plots.

**Supplementary Figure 4A:** Visualization of FAPI uptake (TBR SUVmean) by time point. Quantification
 across the cohort is visualized as bar graphs, median with interquartile range is shown, respectively.
 Paired comparisons were performed using Mann-Whitney-U Test. P-values < 0.016 were considered
 significant after Bonferroni-correction.

Time (mo)	BL	3-6 mo FU	12 mo FU	24 mo FU
Number of values	11	11	10	5
Minimum	1,790	1,151	0,6394	0,8951
25% Percentile	2,046	1,662	0,7353	1,023
Median	2,813	2,110	1,055	1,471
75% Percentile	3,581	2,941	2,206	1,918
Maximum	4,668	3,964	2,941	2,174
Range	2,877	2,813	2,302	1,279
Mean	3,034	2,209	1,381	1,471
Std. Deviation	0,8971	0,8343	0,8272	0,4911
Std. Error of Mean	0,2705	0,2516	0,2616	0,2196

**Supplementary Figure 4B: Descriptive Statistics.**

3) Effects of CD19-CAR T-cell therapy on fibroblast function and phenotype: Supplementary Figure 2
 presents valuable single-cell data across different time points. However, it would be helpful to
 summarize these findings using descriptive statistics (e.g., means, standard deviations) and to
 perform group-level statistical comparisons. This would enhance interpretability and allow readers to
 better assess the consistency and significance of the observed changes.

We thank the reviewer for his/her comment. Supplementary figure 3 (former Supplementary Figure 2)
 summarizes the CAR T-cell response score across different cell types. We have aggregated the scores
 by patients and cell types. Fibroblasts consistently demonstrate higher CAR T response scores for all
 patients. We have included the mean CAR T response scores for each cell type and patient in
 Supplementary Data File 2 CAR T response score 31012026.xls.

 **Response letter Figure 4: Visualization of the mean CAR T-cell response score across cell types. Each**
 **point represents an individual patient, colored by patient identity. Lines connect points from the same**
 **patient, highlighting the within-patient variation in CAR T-cell response score across cell types.**

We also listed descriptive statistics (mean \pm SD) of the CAR T-cell response Score here:

Adipocyte (0.32 ± 0.29), endothelial cell (0.77 ± 0.31), fibroblast (1.09 ± 0.38), gland cell (0.60 ± 0.46),
 immune cell (0.49 ± 0.32), keratinocyte (-0.05 ± 0.12), melanocyte (0.31 ± 0.10), pericyte (0.73 ± 0.31),
 Schwann cell (0.44 ± 0.22), smooth muscle cell (-0.03 ± 0.34).

 4) Rows 328-333: despite R values, p values are merely explorative. This seems due to the small
 sample size, yet it is advisable to exercise caution in the interpretation of these findings.

We acknowledge that sample size and p-values are explorative. We thus deemphasized the description
 in the research section from "correlate" to "tend to associate" (p.13, ll. 321 ff):

"Next, we analyzed the *association* of the quantitative changes of fibroblast subpopulations with
 clinical improvement of the skin ($\Delta mRSS$) upon CD19-CAR T-cell treatment (Figure 3 H and
 Supplementary Figure 14): the relative reduction of PU.1+ fibroblasts *tended to* associated
 with the relative reduction in mRSS score ($R_s=0.67$, $p=0.22$). Moreover, the increase of
 P116+FAP+ fibroblasts ($R_s=-0.78$; $p=0.12$) and Collagen^{high}FAP- fibroblasts ($R_s=-0.67$;
 $p=0.22$) *tended to* associate with a reduction in mRSS score, consistent with dermal
 regeneration."

 As requested, we emphasized the exploratory character of the study more in the discussion as detailed
 in response to your first comment (p.10, ll. 309 ff. of this response letter).

5) Why capillaroscopy was not re-performed at 12 months? That would have given further insights
 about long-term effects of CART-T therapy. The lack of functional data beyond 3 months also limits the
 long-term interpretation and clinical implication of results. These aspects should be well underlined
 and discussed.

We added the information on all available capillaroscopy data and functional assessments of
 Raynaud's phenomenon as assessed by Raynaud-questionnaires performed at twelve months, that
 were acquired since first submission. The respective graphs now look as follows:

 **Figure 4B:** Results of Raynaud's Questionnaire assessing the median duration of Raynaud's attacks,
 median pain intensity and number of Raynaud's attacks per day before (n=4; patients 5; 8; 9; 12), 3
 616 months (n=5; patients 5; 6; 8; 9; 12) and 12 months (n=4; patients 5; 6; 7; 8) after CAR T therapy. Each

617 data point represents Raynaud diary of one day, diaries of two to three consecutive days were available
 and analyzed per patient and time.

**Figure 4C:** Capillaries/mm were counted in several capillaroscopy pictures taken from at least 3
 different fingers before (n=3; patients 8; 9 ;12), 1-3 months (n=3; patients 8; 9 ;12) and 6 months (n=1;
 patient 8) and 12 months (n=3; patients 8; 9; 12) after CAR T-cell therapy.

6) The increase in vessel fragment length and CD31⁺ cell counts may reflect structural changes rather
 than true neoangiogenesis.

We thank the reviewer for his/her comment. We agree that the increase of vessel fragments is
 descriptive and does not prove de novo vessel sprouting. We thus did not interpret or describe this
 observation as “neoangiogenic” throughout the manuscript but referred to it as “increase of vessel
 structures” (p.13, l.343). Interestingly, on the transcriptomic level and using FGSEA on the cISH data,
 we observe the upregulation of angiogenesis-related terms particularly prominent in GJA5-VEC cells.
 Consistently, increased cell communication between endothelial cells with Vascular-endothelial
 growth factor (VEGF) was observed, which is referred to as “biomarker of angiogenesis” [1]. Thus,
 angiogenesis may contribute to the increase of vessel fragments; however, functional experiments,
 e.g. serum stimulations, were not performed within this study. This is discussed (p.18, ll. 457-465):

*“In line with clinical improvement of vascular symptoms such as Raynaud’s phenomenon and*
 *increase of capillary density, CD19-CAR T-cell therapy partially restored vascular skin*
 *structures on the histologic level. Transcriptomic analyses showed increased enrichment of*
 *angiogenesis-related terms, which may be a central contributor to vessel structure*
 *regeneration. Consistently, increased cell-cell interaction with pro-angiogenetic signals such*
 *as VEGF and IGF were observed after CD19-CAR T-cell treatment. These results indicate*
 *vascular remodeling upon CD19-CAR T-cell therapy as part of the process of tissue*
 *regeneration; however, further functional studies, e.g. using vascular model systems, are*
 *warranted to further describe the mechanisms in detail.”*

4) IMC has been employed to validate fibroblast and endothelial cell populations at the protein level,
 no equivalent validation has been performed for epithelial cells. This methodological gap raises
 concerns regarding the robustness of the functional subdivision of epidermal cell types derived solely

from cISH. Consequently, the interpretation of epidermal remodeling should be considered with
caution, and this limitation ought to be explicitly acknowledged in the results section.

Due to the limited number of channels in the IMC analysis and the panel directed towards the analysis
of fibroblasts and endothelial cells, epithelial cells as defined by the marker E-cadherin were only used
for reference in the IMC-based analysis without the aim to differentiate epithelial cell subsets. The high
background derived from unspecific antibody binding at the epidermal region also limits the
robustness of IMC-based analysis in epithelial cells. However, the transcriptomic definition of epithelial
cell is well characterized [2, 3]. As suggested, a sentence regarding this limitation has been added to
the cISH results section (p.16, ll. 411-413):

*“Our results indicate major functional changes toward regeneration in epithelial cells upon*
*CAR T-cell treatment at the transcriptomic level. Further studies may be required for protein-*
*based validation of these phenotypes.*

“Moreover, this limitation is now mentioned in the discussion section (p 19, ll. 470-481):

*Moreover, our findings obtained in cISH analysis including the temporal shifts IFN-responsive*
*and ECM-related transcriptional programs, and dynamic BMP/HSPG, THBS, and MPZ*
*signaling—suggesting that keratinocytes may act as active modulators of dermal/epidermal*
*regeneration upon B-cell depletion. A limitation is that these findings were not validated on the*
*protein level; however, the transcriptomic data may generate several hypotheses that may be*
*validated on the functional level in future studies, e.g. (i) BMP/HSPG-driven communication*
*between epibasal and suprabasal layers may initiate epidermal re-stratification and (ii) MPZ-*
*associated crosstalk between Schwann cells, fibroblasts, and epidermal cells may contribute to*
*papillae regeneration. Such pathways may represent novel therapeutic entry points that*
*synergize with CD19-CAR T-cell therapy by enhancing epidermal–stromal repair. However,*
*models that can adequately recapitulate the human dermal/epidermal interface are warranted*
*to answer these research questions in the future.”*

5) While the study provides compelling transcriptomic and spatial evidence of epidermal remodeling,
further investigation using (animal) functional model may be warranted to validate the regenerative
potential of epithelial changes. The authors may wish to elaborate on how the observed epidermal
dynamics could inform future mechanistic studies or therapeutic strategies.

We thank the reviewer for his/her positive comment. We agree that the study is descriptive, and
functional experiments are warranted. We thus amended the discussion to better outline how our
findings may inform future research. We may comment that the shape and interaction of rete ridges
with dermal papillae is quite peculiar to the human dermis. Mice, for instance, do not have a papillary
dermal layer and rete ridges. Humanized models that recapitulate this aspect could overcome this
limitation in the future. The respective text passage now reads as cited in response to your previous
comment (manuscript p.19, ll. 470-481).

Reviewer #3 (Remarks to the Author):

The manuscript describes changes in the skin (and in a limited fashion also in the lung and heart) after
 CAR-T therapy in patients with systemic sclerosis. Multiple sophisticated approaches are applied and
 strongly suggest an effect of the treatment on skin at the level of pathology and molecular gene
 expression - detected at both bulk and single cell levels. As such, this represents the best manner for
 pursuing a translational analysis of a treatment effect for early-stage clinical development. The
 analytical approaches are also sound. Unfortunately, the fibroblast populations are not identified
 accurately leading to some faulty interpretation of the results. This issue is described in detail below
 and makes it difficult to follow the remainder of the manuscript. I have some other relatively minor
 concerns.

We thank the reviewer for his/her encouraging comments on this project and for pointing out the
 matter of fibroblast identification. We have relabeled the myofibroblasts as further detailed below
 (p.28, ll.797 ff. of this response letter). We further performed verification of all fibroblast populations
 by mapping our data to a recently published skin fibroblast atlas (Nat Immunol 2025 Oct;26(10):1807-
 1820) [4]. This manuscript intended to present a standardized fibroblast nomenclature applicable to
 healthy and diseased skin. This study had also been informed by published datasets acquired from SSc
 samples (GSE138669 [5], GSE249279). In summary, the results of the revised analysis show, that our
 cISH data recover the majority of fibroblast populations in the skin and matched the pre-described
 fibroblast populations except for the fascial fibroblast, which is likely not recovered in the samples
 consistent with biopsies reaching into the deep dermis and subcutis but do not reach the fascia.

Supplemental Table 1 shows standard of care and natural disease course patients but the difference
 between these groups is unclear until reading the supplemental methods-it would be good to point
 to this in the supplemental table legend. It's great that they authors have included a control group
 but it's worth noting at some point that the control group has, as a group, lower skin scores.

We added the definition of the standard of care and natural disease course group to the legend of
 supplementary table 1. We agree that the mRSS scores in control groups are biased towards lower
 baseline values. As also requested by reviewer 4, a summary of statistics on the baseline variables is
 now added to supplementary figure 1 a. We explicitly discussed the bias towards a lower baseline
 mRSS in the control cohort (p.23, ll. 575-588):

*“Given the novelty and complexity of the CD19-CAR T-cell approach, studies including a*
 *randomized, blinded control arm were not available at the time of this manuscript.*

*Thus, employing an external systemic sclerosis cohort represents a pragmatic alternative for*
 *contextualizing clinical outcomes, as also acknowledged in a recent recommendation of the*
 *Food and Drug Administration (FDA) for clinical trial design for cellular therapy products in*
 *small populations (FDA-2025-D-3403). Nonetheless, the lower baseline mRSS in the external*
 *cohort introduces a bias that may attenuate the apparent relative improvement attributable to*
 *treatment or influence the natural trajectory of skin scores, thereby requiring careful*
 *consideration when interpreting comparative efficacy.”*

Supplementary Figure 1 is confusing and does not convincingly show depletion of B cells in the skin.
 Patients 2, 5 and 6 appear on two graphs of CD20+ cells, with different scales that do not match
 across the graphs. This is a very important point. Past studies have shown that mononuclear cell
 including B cell infiltration in scleroderma skin is quite variable and B cell numbers minimally

increased in the majority of skin biopsies. This appears to be the case here, where most of the skin
appear to have minimal numbers of B cells and only two of the biopsies (patients 2 and 4) showing
largely increased numbers, and only one of these show depletion in the right most graphic (patient
2).

We apologize for the confusion on the graph labelling: A mistake happened and the third graph on the
right refers to the counts of CD3+ T-cells and was accidentally labelled wrong with CD19+. This also
explains the mismatch of axis scale. This error has been corrected. The graphs in the left and in the
middle show the counts of CD19+ and CD20+ B-cells, respectively. Consistent with the reviewer's
comment, the number of B-cells in the skin vary inter-individually, however, the counts show a clear
depletion of B-cells, in particular the patients who had elevated B-cell counts at baseline (patients 5
and 7). Furthermore, representative images have been added for all three immunohistochemistry
investigations (CD19, CD20 and CD3) to supplementary former Supplementary Figure 1 (current
Supplementary Figure 2), which now looks as follows:

**Supplementary Figure 2: A** Individual course of the numbers of CD19+, CD20+ B-cells and CD3+ T-cells
 before and after CD19-CAR T-cell treatment as analyzed by immunohistochemistry. **B C D**
 Representative immunochemistry staining images of the CD20 (B), CD19 (C) and CD3 (D) for the four
 timepoints. BL = Baseline; 1m = 1 month; 6m = 6 months; 12m = 12 months. All representative images
 are from patient 5 of the CART group.

Figure 1D suggests that B cell depletion has occurred. However, the increase in tissue remodeling and
 extracellular structure organization suggests that matrix gene expression might paradoxically be
 increased. However, Supplemental Figure 4 shows that key matrix gene expression associated with
 scleroderma, i.e. COL1A1, COL1A2, CTHRC1 and SERPINE1 are changed after treatment Unfortunately

Suppl Figure 3 and 4 lack a legend so that the direction of change is not discernable. If red is higher,
then these genes are showing increased expression after treatment—a surprising result. Supplemental
data indicates these genes are decreased after treatment. Please indicate the number of skin samples
included in the RNA-seq/GSEA analysis.

We thank the reviewer for his/her comment. Although the term “extracellular structure organization”
may suggest matrix gene expression, it mostly relates to extracellular matrix regulators, which we
interpret as part of the remodeling process.

The sample size of the analysis is now included in the legend of Figure 1D:

*“Dot plot showing the enriched pathways detected by gene set enrichment analysis (GSEA) of*
*RNASeq data from skin biopsies collected at 1 month (n=5) or 12 months (n=3) after CAR T-*
*cell therapy, compared to baseline (n=6).”*

As bulk sequencing does not reflect gene expression on the single cell level, we reannotated the
different fibroblast populations as described in response to your comments below and analyzed the
gene expression of ECM on the level of different fibroblast populations to achieve more granular data.
Here we observed the consistent downregulation of extracellular matrix gene expression across
several fibroblast populations, in particular FAP+, COL8A1+ myofibroblasts (figure 3A).

We apologize for the confusion of the heatmaps of former Supplementary Figures 3 and 4, which are
now removed.

Please provide the genes making up the “CAR T response score”.

We have now included the genes and the fold changes, which were used for computing CAR T
response score, in the Supplementary Data file 4.

Myofibroblasts are typically seen first in the deep reticular dermis. ACTA2 and TAGLN + cells are likely
marking the dermal sheath cells. Although ACTA2 is indeed a marker for myofibroblasts, it is more
highly expressed by dermal sheath fibroblasts and smooth muscle/pericytes. The cells expressing
FAP, COMP, SFRP4 and COL8A1 are the myofibroblasts (Nat Commun 12, 4384 (2021)). The TNN,
COCH cells likely are also expressing CRABP1, a well-documented marker for dermal papilla. APCDD1
is a marker for papillary fibroblasts (or the major population of these fibroblasts). Superficial
fibroblasts should probably be referred to as papillary fibroblasts (Journal of Investigative
Dermatology Volume 138, Issue 4, April 2018, Pages 811-825). These designations are more
consistent with their beautiful spatial localization data (Suppl Figure 5) which clearly shows the cells
they refer to as myofibroblasts around hair follicles and blood vessels, Whereas the FAP+COL8A1+
cells are located in the deep dermis. The diffuse staining of PI16+ cells in the dermis is consistent
with its expression by the most common SFRP2+ fibroblast population, which appear adjacent to the
FAP+COL8A1+ myofibroblasts (Figure 2E) and are likely progenitors of myofibroblasts. Unfortunately,
ref 33 paper lacks a significant number of myofibroblasts, making this a poor reference for
identification of this important fibroblast population. Supplemental figure 5 shows the localization of
these populations consistent with the descriptions.

We thank the reviewer for his/her comment and for the appreciation of our spatial data. As suggested,
we revisited the annotation of the respective populations: As commented by the reviewer, the cell
population that was annotated as myofibroblasts in the previous version of this manuscript expressed
the pericyte marker *RGS5* and did not show high myofibroblast signatures adapted from Steele et al.

(Nat Immunol 2025 Oct;26(10):1807-1820)³⁴ and Tabib et al (Nat Commun 12, 4384 (2021)³⁵, whereas
 the FAP+COL8A1+ indeed showed the most prominent expression of the myofibroblast signature (See
 Response letter Figure 5 below, myofibroblast signature gene set: ACTA2, COL5A1, COL8A1, POSTN,
 CTHRC1, LRRC15, SFRP4, ASPN, RUNX2). We have thus re-analyzed the cISH data after labelling this
 population as pericyte. We also co-labeled the FAP+COL8A1 fibroblasts as “myofibroblasts” based on
 the reviewer’s comment and the highlighted citations.

Unfortunately, we are unable to verify the expression of *CRABP1*, as it is not in our cISH gene panel.
 However, the co-expression of *TNMD* also confirmed the identity of TNN+COCH+ based on the
 previously reported spatial transcriptomics study (Nat Commun. 2024 Jan 3;15(1):210.)³⁶.

We confirm the expression of *APCDD1* in superficial fibroblasts in Figure 2D, which we re-labeled as
 “papillary fibroblasts” as both designations are used synonymously³⁶⁻³⁸.

**Response letter Figure 5:** The dot plot characterizes the original fibroblast subtypes based on the
 expression of marker genes for fibroblasts and pericytes. Dot size represents the proportion of cells
 expressing each gene, and dot color indicates the average expression level. The UMAP plot shows
 myofibroblast scores projected onto the same UMAP embedding of cell types as in the middle panel.
 Colors represent the myofibroblast signature, with blue indicating low expression and red indicating
 high expression.

We also confirmed the spatial distribution of the papillary fibroblasts in the upper dermis, the
 FAP+COL8A1+ myofibroblasts in the deeper dermis, and the diffused distribution of PI16+ fibroblasts
 as visualized in supplementary figure 7. Indeed, the PI16+ fibroblast also expressed high levels of
 *SFRP2*, which is now shown in the revised Figure 2D, as indicated below:

**Figure 2D:** Dot plot characterizes eight fibroblast subpopulations based on the expression of known
 marker genes. Dot size represents the proportion of expressing cells and dot color indicates the
 average expression level. FAP+COL8A1+ cells correspond to myofibroblasts.

The manuscript emphasis and results and discussion need to be re-considered in view of the proper
 identification of fibroblast subpopulations and clarification on the effect on matrix and biomarker
 gene expression. It appears that the message will change to one indicating that CAR-T depletion is
 associated with decreasing numbers of myofibroblasts and possibly matrix gene expression.

We have performed additional validation of cISH-identified fibroblast subpopulations by integrating
 our data to the skin fibroblast atlas created by Steele et al.³⁴ where the data from Tabib et al.³⁵ and
 834 Ma et al.³⁶ were included. The results of scANVI-based integration are now presented in the
 835 Supplementary Figure 8 and a section was added to the online methods (p.17, ll. 406 ff.). We observed
 high concordance between cISH-identified populations and the ones reported in the atlas published
 by Steele et al. For instance, the papillary (formally superficial) fibroblasts correspond to *F1: Superficial,*
 *also referred to as "papillary fibroblasts",* CCL19+ and PU.1+S100A4+ fibroblasts mapped to
 inflammatory *F3: FRC-like,* PI16+ fibroblasts mapped to *F2: Universal progenitor* population, and
 FAP+COL8A1+ myofibroblast to disease-specific myofibroblast populations, including *F6: Inflammatory*
 *myofibroblast, F7: Myofibroblast,* and *F8: Facia-like myofibroblast.*

A

B

**Supplementary Figure 8: Correspondence of cISH-identified fibroblasts to skin fibroblast atlas**

**A** UMAP showing scANVI-based integration of fibroblast populations identified by cISH with the
 populations reported in the skin fibroblast atlas (PMID: 40993240). **B** Sankey plot showing the
 correspondence between cISH-identified fibroblasts and the populations defined in Steele et al., as
 predicted by scANVI.

As requested, we revised the results section after re-annotating the fibroblasts and included more
 relevant studies on myofibroblast as reference. The revised description of fibroblast populations (p.8,
 850 l.217-p.9, l.233) now reads as follows:

*“Next, we focused on the detailed phenotyping of fibroblast populations using cISH and*
 *identified eight fibroblast subpopulations (Figure 2E). Aligned with the spatial genomic atlas*
 *³⁴, the papillary fibroblasts (expressing COL18A1 and COL23A1) and TNN+COCH+*
 *fibroblasts are located right beneath the skin epidermal layer and adjacent to hair follicles,*
 *respectively, confirming their spatial localization (Supplementary Figure 7). The expression*
 *profiles of papillary fibroblasts also resemble their reported phenotype (Figure 2D) ³⁸.*
 *Similarly, the P116+ fibroblasts exhibited a universal fibroblast progenitor phenotype*

*associated with regenerative functions*³⁹. We also recovered other fibroblast populations of
*SSc skin described in previous studies, including CCL19+*⁴⁰, *PU.1+S100A4+*⁴¹, *IGFBP4+*⁴²,
*CCL8+ (expressing ACKR3;*^{43,44}*), and FAP+COL8A1+, which co-expressed the myofibroblast*
*marker SFRP4*^{35,36} (Figure 2E). We further validated the *cISH-identified fibroblast populations*
*with the spatial genomics atlas by scANVI-based integration. The results indicated good*
*correspondence of the cISH-identified population to the reported phenotype. For instance,*
*papillary fibroblasts correspond to F1: Superficial in the atlas; CCL19+ and PU.1+S100A4+*
*fibroblasts mapped to inflammatory population F3: FRC-like; PII6+ fibroblasts mapped to*
*F2: Universal progenitor population; and FAP+COL8A1+ myofibroblasts to disease-specific*
*myofibroblasts (Supplementary Figure 8).”*

In the following, we referred to this revised annotation throughout the description of results relating
to fibroblast populations (p.9, l. 235-p.11, l 276).

We also revised the discussion accordingly. We particularly emphasized that FAP+COL8A1+ fibroblasts,
which show prominent reduction upon CD19-CAR cell therapy and reduced expression of ECM genes,
are consistent with myofibroblasts. The respective section (p.17, l.435-p.18-l.451) now reads as
follows:

*“Our results suggest several mechanisms of CD19-CAR T-cell-mediated skin remodeling on*
*the tissue level: First, the clinical improvement of mRSS is accompanied by a shift in fibroblast*
*population composition toward a more physiologic distribution both at the transcriptomic and*
*protein levels, as assessed using two complementary single-cell spatial imaging approaches*
*(IMC and cISH). Across both datasets, we observe decreased proportions of fibroblast subsets*
*associated with pro-fibrotic and pro-inflammatory activity, including reductions in PU.1+*
*fibroblasts and myofibroblasts, which showed a high collagen score and activation of TGFβ*
*related signaling cascades as analyzed by cISH and in line with the previous literature*^{35,45}.
*Conversely, fibroblast populations linked to more homeostatic and potentially antifibrotic*
*roles, such as PII6+ fibroblasts and TFAM+ fibroblasts—typically diminished in SSc—showed*
*numerical increases following CD19-CAR T-cell therapy. Of note, the shift towards*
*regenerative fibroblast populations was particularly pronounced in the papillary dermis.*
*In addition to changes of fibroblast composition, we observe changes of fibroblast function as*
*demonstrated by cISH: fibrosis-related FGSEA terms such as “extracellular matrix*
*organization” and “collagen formation” decline across several fibroblast populations,*
*particularly myofibroblasts.”*

**Reviewer #4 (Remarks to the Author):**

This study investigates whether deep B-cell depletion via CD19-targeted CAR T-cell therapy can not
only halt but also reverse tissue fibrosis in systemic sclerosis (SSc). The authors performed sequential
forearm skin biopsies in 11 patients with diffuse cutaneous SSc before treatment and at 1, 6, and 12
896 months after CAR T infusion, with additional biopsies from control SSc cohorts (receiving standard-of-
897 care therapy or observed over the natural disease course). Remarkably, CAR T therapy induced
structural regeneration of the skin: the density and height of dermal papillae (rete ridges) in SSc
patients significantly increased, approaching levels seen in healthy skin. This regrowth of dermal
papillae was accompanied by reduced dermal fibrosis – collagen fibers became less aligned (a sign of
fibrosis regression) – and by improvement in clinical skin scores (mRSS) over 12 months. Multi-modal
molecular analyses further showed that CAR T treatment shifted the tissue toward a more “healthy-

like" phenotype: pro-fibrotic gene signatures were downregulated, while pathways for angiogenesis,
extracellular matrix remodeling, wound healing, and epidermal development were upregulated in
post-treatment skin. In summary, the key finding is that profound B-cell depletion via CD19 CAR T cells
led to an unprecedented reversal of SSc-associated skin fibrosis, with restoration of more normal skin
architecture, cellular composition, and gene expression profiles. Overall, the manuscript is very well
written, and the data strongly support the authors' conclusion that CAR T therapy can remodel fibrotic
skin toward a regenerative state.

We thank the reviewer for his/her encouraging comment.

However, some issues, limitations, and questions remain:

1. Introduction

a. The authors state that this is the first time an intensive therapy has restored near-normal skin
anatomy. This claim should be made more cautiously, as some studies have suggested that
autologous stem cell transplantation (ASCT) or other therapies can also lead toward normalization of
scleroderma skin histology. For example:

i. Autologous hematopoietic stem cell transplantation modifies specific aspects of systemic sclerosis-
related microvasculopathy. PMID: 35368373

ii. Autologous hematopoietic stem cell transplantation reverses skin fibrosis but does not change skin
vessel density in patients with systemic sclerosis. PMID: 26300240

iii. B cell depletion therapy upregulates Dkk-1 skin expression in patients with systemic sclerosis:
association with enhanced resolution of skin fibrosis. PMID: 27208972

Therefore, the authors should be more cautious in stating that current medical therapies do not halt
disease progression or induce tissue regeneration. It may be that previous therapies have not been
studied as deeply as in this work, and while CAR T cells might indeed perform better than other
intensive treatments, that remains to be proven.

We thank the reviewer for his/her comment and apologize for the misunderstanding. We did not
intend to state that CD19-CART cell therapy is the only intensive therapy that can slow down disease
progression or impact other pathophysiological aspects in SSc towards normalization. However, to the
best of our knowledge, this is the first study to show regeneration of the papillary skin structure and
associated structures including vessels, fibroblasts and keratinocytes, although other aspects such as
the normalized fibrosis score were described upon ASCT. In addition, the references named by the
reviewer suggest that ASCT may lead to normalization of capillaroscopy as assessed in two patients
(PMID: 35368373) [51], while skin vessel density remained unchanged in an exploratory analysis in
another study (PMID: 26300240) [50].

In the third suggested reference (PMID: 27208972), the authors demonstrate that rituximab treatment
results in restoration of dermal DKK-1 expression in 4 out of 8 RTX treated SSc patients and is
associated with the increased mRSS response.

Following the reviewer's suggestions, we rephrased the respective passages, which now read as
follows:

Passage 1 (p.4., ll. 93 ff) :

*“In SSc, the papillary dermal structures disappear with progression of the disease, reflecting*
*both vascular loss and fibrotic tissue remodeling, as the skin transitions to a “reticularized”*
*phenotype^{46,47} with accumulation and increased alignment of collagen fibers throughout the*
*dermis⁴⁷⁻⁴⁹. These skin changes and papillary loss as characteristic hallmarks of SSc skin have*
*been considered permanent for a long time. Studies performed on skin biopsies in patients who*
*had received autologous stem-cell transplantation (ASCT) have shown that broad*
*immunosuppression can reduce the extent of skin fibrosis, as assessed by the semi-quantitative*
*normalized fibrosis score⁵⁰, and obtain improvements of capillaroscopy results⁵¹. However,*
*reversal of the dermal phenotype with recovery of the dermal papillae, has not been*
*investigated so far.”*

**Passage 2 (p.5, ll. 122 ff):**

*“The rationale of targeting B-cells in SSc is further supported by clinical trials of the CD20-*
*depleting antibody rituximab (RTX), which have shown improvements in both skin fibrosis and*
*interstitial lung disease (ILD)³²⁻³⁴ and exploratory analyses suggest that clinical response may*
*be linked to dermal remodeling of pathways such as the upregulation of dermal DKK-1³⁵.”*

b. Provide additional context on B cells in SSc (both circulating and in tissues). For instance, briefly
summarize how B cells contribute to SSc pathogenesis – e.g., through production of pro-fibrotic
cytokines, autoantibodies, or interactions with fibroblasts – to frame why targeting B cells could impact
fibrosis.

We thank the reviewer for his/her comment. We amended the introduction and detailed more on
the role of B-cells in SSc, as also requested by reviewer 2, minor comment 1. The new passage now
reads as follows (p.4, ll.103 ff.):

*“We hypothesize that B-cell driven autoimmunity may be a key driver of cutaneous remodeling*
*in SSc, as growing evidence implicates B cells in orchestrating fibrotic tissue remodeling in*
*SSc. Dysregulation of circulating B-cells is evident at several levels in SSc, including aberrant*
*B-cell receptor (BCR) signaling characterized by enhanced activating and diminished*
*inhibitory signals⁸⁻¹⁰ and disturbed effector function resulting in hypergammaglobulinemia*
*and autoantibody secretion. Distinct autoantibodies associate with defined clinical SSc*
*phenotypes^{3,11}, e.g. anti-scl70 antibodies are associated with a high risk of disease progression*
*and inner organ involvement. Other autoantibodies exert direct pathogenic effects: agonistic*
*antibodies targeting anti-platelet-derived growth factor receptor (PDGFR) or fibrillin promote*
*fibroblast activation^{12,13}, while anti-endothelial cell antibodies, anti-endothelin-1-type A*
*receptor, and anti-angiotensin-II type 1 receptor antibodies contribute to endothelial cell*
*damage and vasculopathy^{9,14,15}. In addition to autoantibody-mediated mechanisms, B-cells*
*shape the fibrotic milieu through the secretion of pro-fibrotic and pro-inflammatory cytokines*
*such as IL6^{16,17}, whereas regulatory B-cell subsets that restrain inflammation, e.g. via IL10,*
*are decreased in SSc¹⁸. B-cells also infiltrate the organs affected in SSc and may mediate*
*tissue remodeling^{9,19}: A recent exploratory study using a three dimensional humanized skin*
*model showed, that infiltrating B-cells of SSc patients show a hyperactivated phenotype,*
*increased immunoglobulin production and the capacity to induce profibrotic transcriptomic*

*programs in resident fibroblasts, supporting a direct role for B-cells in tissue-level disease*
*propagation*²⁰.

*The rationale of targeting B-cells in SSc is further supported by clinical trials of the CD20-*
*depleting antibody rituximab (RTX), which have shown improvements in both skin fibrosis and*
*interstitial lung disease (ILD)*²¹⁻²³ *and exploratory analyses suggest that clinical response may*
*be linked to dermal remodeling of pathways such as the upregulation of dermal DKK-1*²⁴.
*However, despite encouraging results, many patients still progress under conventional B-cell-*
*targeted therapy, highlighting a critical limitation of current B-cell-directed therapies. CD20-*
*targeting antibodies incompletely deplete tissue-resident B-cells*²⁵⁻²⁷ *and fail to eliminate*
*antibody-producing plasmablasts. While ASCT may overcome this limitation, it carries*
*significant procedure-related risk. We recently demonstrated that CD19-directed chimeric*
*antigen receptor (CAR) T cells achieve deep and broad B-cell depletion across tissues*^{25,26}.
*Consistent with this, we and others have reported profound clinical responses in patients with*
*SSc treated with CD19-CAR T-cell therapy*²⁸⁻³⁰. *Based on these several lines of evidence, we*
*hypothesize that deep depletion of B-cells by CAR T-cell therapy may interfere with the B-cell*
*mediated fibrotic and vascular remodeling, thereby reshaping diseased tissue architecture in*
*SSc.”*

2. Methods & Results

a. Sample size and data handling: The sample size is inherently limited (n = 11 treated patients), as
this is a pilot study in a rare disease. While understandable, this does impose caution – statistical
power is modest, especially for subgroup analyses at later time points (only 4 patients had 12-month
biopsies). The data presentation raises some questions. The authors report 11 patients at baseline,
10 at 1 month, 6 at 6 months, and 4 at 12 months (Figure 1). Are the 4 patients with 12-month
biopsies a subset of those evaluated at earlier time points? If so, is it valid to analyze longitudinal
changes without restricting to the 4 patients who had data across all time points? The manuscript
should clarify how missing data were handled. For example, if a patient missed the 6-month biopsy,
were they excluded from the 6-month vs. baseline comparisons? The authors must clarify the biopsy
sampling and analysis plan. It would help to explicitly state how many patients had biopsies at each
follow-up and whether analyses (e.g., comparisons to baseline) are based on paired samples. For
instance, the Figure 1 legend indicates n=11 at 1 and 6 months, but n=4 at 12 months, whereas the
spatial transcriptomic data at 12 months included 3 samples (Fig. 3). These discrepancies should be
explained (perhaps some 12-month samples were used for histology but not for spatial
transcriptomics due to quality or timing differences). Including a simple statement in the Methods or
Results about how many patients contributed to each dataset and why certain data are missing
would prevent confusion.

*We thank the reviewer for his/her comments. The samples analyzed were obtained consecutively from*
*patients who received CD19-CAR T-cell therapy. Because of this consecutive enrollment, patients*
*treated earlier, contribute more longitudinal samples than those treated more recently. As requested,*
*we added an overview of all samples acquired and analyzed (Supplementary Data file 1_ clinical data*
*and sample overview_31012026). Except the baseline biopsy of patient 1 (sampling error,) no biopsy*
*was missed by any patient. In case a sample did not pass quality control, the samples were excluded*
*from the respective analyses, information on samples that did not pass quality control are added in*
*Supplementary Data File 1. Samples were included consecutively to the analysis workflow according*

to prespecified procedures. As requested, we clarified the sample analysis plan in the methods section
 and added a visualization of the sample analysis workflow (Online methods, p 5., ll. 110 ff.):

***“Sampling and analysis plan***

*Skin punch biopsies (three-millimeter diameter) were taken at the involved forearm; successive*
 *biopsies were taken at one cm from the baseline biopsy spot. In the CAR T cell group, baseline*
 *biopsies were taken two to four weeks before leukapheresis, during this interval, previous*
 *immunosuppression was paused. Follow-up biopsies were taken 1 month, six months and twelve*
 *months upon receipt of CD19-CAR T-cells. In the SOC and NDG groups, samples were*
 *acquired 2-4 weeks before the initiation of the novel treatment and a follow-up biopsy was taken*
 *during clinical monitoring after six months. All samples were consecutively included in the*
 *stepwise analysis workflow depicted in online methods figure 1. As most obvious changes were*
 *observed in the CAR T- group in step 1, we focused on this group in the consecutive steps. In*
 *case a sample was missing or did not pass quality, the sample was excluded from the further*
 *consecutive analyses. An overview of all acquired and analyzed samples is summarized in*
 *supplementary data file 1.”*

**Online methods figure 1: Samples analysis workflow**

Moreover, the number of the patients investigated in each analysis were added to the figure legends.

Given the exploratory nature of the study, we argue that highest granularity of data can be
 achieved by inclusion of all available samples per time point. However, to allow a clearer view, inter
 subject trajectories were added in each graph in addition to group level statistics as depicted in
 response to reviewer 2 (p.15., ll. 447 ff. of this response letter).

b. Baseline characteristics: The study included predominantly severe diffuse cutaneous SSc patients
 (Supplementary Table 1), likely young- to middle-aged adults with high mRSS and significant internal

organ involvement. The authors should briefly note any key baseline similarities or differences
 between the CAR T group and the control cohorts (e.g., average mRSS, disease duration) to
 contextualize the outcomes. If any significant baseline imbalances existed (for instance, if the CAR T
 patients had more severe skin scores or shorter disease duration than the standard-of-care group),
 these should be mentioned as they could affect the interpretation of the results.

We thank the reviewer for this thoughtful comment. We have added a table summarizing key baseline
 characteristics of the CAR T and SOC cohorts to Supplementary Figure 1A:

Baseline characteristics	CD19-CAR T	SOC	p-value
age (mean (SD))	41.5 (14.8)	46.6 (14.1)	0.46
Sex			
male (n (%))	7 (63.6)	5 (62.5)	0.94
female (n (%))	4 (36.4)	3(37.5)	
Disease Duration (mo, mean (SD))	45.6 (37.4)	47.3(59.0)	0.5
auto-antibody status			0.25
anti-Scl70 (n (%))	9(81.8)	5(62.5)	
anti-RNAP III (n (%))	1(9.1)	0(0)	
other (n (%))	1(9.1)	3 (27.2)	
mRSS (mean (SD))	23.91(6.2)	15.6(4.6)	0.0072
Lung Involvement (n (%))	11(100)	8 (100)	>0.99
Heart Involvement (n (%))	5 (45.5)	2 (25)	0.633
EUSTAR AI (mean (SD))	5.5(2.0)	4.2(1.6)	0.13
(SD) number of previous treatments (mean	3.1 (1.5)	0.9 (0.4)	0.0018

**Supplementary Figure 1 A:** Summary statistics of patient characteristics at baseline. Significance of
 difference of proportions in categorical variables comparing two groups were analyzed using Fisher's
 exact test. Continues parameters were summarized as mean and standard deviation, and significance
 testing was done using t-tests. P values below 0.005 are considered significant after Bonferroni
 correction.

As also noted by Reviewer 3, the SOC group had a bias towards a lower baseline mRSS, indicating a
 bias toward less severe skin involvement at similar disease duration. Moreover, patients treated with
 CAR T cell therapy had a higher average number of prior immunomodulatory therapies compared with
 controls, which may reflect a more severe and refractory disease at baseline. This imbalance may
 influence skin outcomes through cumulative treatment effects or altered tissue responsiveness.
 However, despite this potential disadvantage, CAR- T-cell therapy was associated with pronounced
 skin changes, supporting the treatment-specific effect. Nevertheless, residual confounding by prior
 therapies cannot be fully excluded and should be considered when interpreting comparative effects.

We now explicitly acknowledge these baseline imbalances in the discussion sections. The respective
passage now reads as follows (p.23, ll. 575 ff.):

*“Given the complexity of the CD19-CAR T-cell approach, studies including a randomized, blinded control*
*arm were not available at the time of this manuscript. Thus, employing an external systemic sclerosis*
*cohort represents a pragmatic alternative for contextualizing clinical outcomes, as also acknowledged*
*in a recent recommendation of the Food and Drug Administration (FDA) for clinical trial design for*
*cellular therapy products in small populations (FDA-2025-D-3403). Nonetheless, baseline differences*
*between cohorts warrant careful consideration. In addition to a lower average baseline modified*
*Rodnan Skin Score (mRSS) in the external cohort—which may attenuate the apparent relative*
*improvement attributable to treatment or influence the natural trajectory of skin scores—patients*
*receiving CD19-CAR T cells had a higher burden of prior immunomodulatory therapies, potentially*
*reflecting more severe or refractory disease. This imbalance could affect skin responsiveness through*
*cumulative treatment effects or altered tissue biology and may introduce residual confounding in*
*comparative analyses. Therefore, comparative efficacy should be interpreted cautiously, taking these*
*baseline differences into account.”*

c. Analytical approaches and cell composition: The analytical approaches (immunohistology, spatial
transcriptomics, high-dimensional cytometry, etc.) are appropriate and state-of-the-art. The GSEA
results clearly support the narrative of decreased B-cell/plasma-cell activity and increased tissue
regeneration pathways after therapy. The longitudinal analysis of cell composition changes (comparing
baseline vs. 12-month samples, with intermediate time points analyzed in Supplementary Fig. 7) is
presented clearly, showing significant decreases in fibrotic and inflammatory fibroblast subsets and
increases in pro-regenerative cell types. One suggestion: if not already done, the authors could
perform paired (within-patient) analyses for cell composition changes to account for baseline
differences between patients, which would strengthen the findings.

We thank the reviewer for his/her comment. As also suggested by reviewer 2, we added the within-
patient analysis to the group comparisons of fibroblast populations, that changed upon CD19-CAR T-
cell treatment. The new graphs are depicted in response to comment 4 of reviewer 2 (p.15, ll. 447 ff.)
and in Supplementary Figure 10B.

1108 d. “CAR T response score”: Using spatial transcriptomics (e.g., RNA in situ hybridization), the authors
computed a “CAR T response score” for each cell, based on a signature of differentially expressed
genes from the bulk RNA data. This metric is used in the analysis, but the manuscript should clarify
how this “score” is defined and calculated, and what a high or low score signifies biologically.
Providing a clear definition of the score would help readers better understand these results.

We thank the reviewer for pointing out this missing information in the original manuscript. The aim of
the CAR T response score was to compare the transcriptomic changing trends between different cell
types in dermal skin following CD19-CAR T-cell treatment and to decipher the response derived from
bulk RNA Seq data by cell type using the spatially informed cISH dataset. Thus, the word “response”
refers primarily to molecular response by cell type and does not aim to predict a clinical outcome. To
address this, we computed the CAR T response score by multivariate linear modeling. Specifically, we
created a gene set that models the tissue response upon CAR T using the differentially expressed genes
(DEGs) obtained from the bulk RNA-Seq data. Then, we infer the cellular activity using this gene set,
i.e., the response score. Indeed, we are able to pinpoint which cell types exhibit high levels of this
response signature in the tissue from our cISH data. As we aimed to decipher the transcriptomic

response by cell type, we did not primarily associate the scores with global clinical outcomes such as
mRSS. However, we added this analysis upon request and did not observe an association.

We reasoned that the genes used for computing the response score contain various biological
functions. Additional deconvolution and/or factorization analysis may provide further insights on
biological phenotypes. We have now described our approach in the Method section as follows (Online
methods, p.16, ll. 386 ff.):

*"The CAR T-cell treatment response score is computed from the DEG analysis on bulk RNA-*
*Seq data (1-month follow-up compared to baseline) by multivariate linear modeling using the*
*decoupleR R package, with DEGs (adjusted p-value < 0.05) and log2-fold changes input as*
*mode of regulation (Supplementary Data File 2 CAR T response score 31012026)."*

e. Serological and immunological data: It would be informative to include or mention serological data
related to B-cell activity. For example, did autoantibody levels (such as anti-Scl-70) change over the
course of therapy? Also, were B cells starting to reconstitute by 12 months post-CAR T? Even a brief
note on these points would give a fuller picture of the systemic immunological "reset" achieved and
help gauge the durability of response.

We thank the reviewer for his/her comment. We added the autoantibody levels of anti-Scl70 antibody
titers and peripheral B-cell counts (both shown below) to Supplementary Figure 1 and added a
description in the results section which now reads as follows (p.6, ll.151 ff.):

*"In line with previous studies, peripheral B-cells were completely depleted within 9 days in all*
*patients after the application of CD19-CAR T-cells. Their reconstitution started between d62*
*and d253 (Supplementary Figure 1B). Interestingly, while B-cells counts were heterogenous at*
*baseline, no CD19+ B-cells or CD20 + B-cells were detected in any of the available follow up*
*skin samples (Supplementary Figure 2). The serum-titers of anti-Scl70 antibodies declined*
*throughout the observation (Supplementary Figure 1D)."*

**Supplementary Figure 1 B:**

**Supplementary Figure 1D:**

**Supplementary Figure 1B and 1D. B:** Numbers of B cells in peripheral blood before and after CD19
CAR T-cell therapy. **D:** Course of anti – Scl70 titers in patients 2 -11.

We also referred to these results in the discussion as further detailed in response to your comments
“3. Discussion” (p.46, ll.1341 ff. of the response letter).

f. Patient-level response variability: Consider providing patient-level response data (perhaps in a
table or supplementary figure). Summarizing each CAR T-treated patient’s clinical course – including
mRSS changes, any improvement in organ involvement, and relevant laboratory values – would allow
readers to appreciate the variability or consistency of responses. Did all patients show skin
improvement, or were there any partial responders/non-responders? If a particular patient had less
improvement, what was distinctive about their case (e.g., longer disease duration, lower CAR T
expansion, different autoantibody profile)? Detailing such individual outcomes could yield insights
into predictors of response.

We thank the reviewer for his/her comment. As also requested by reviewer 2, we added a table
summarizing central parameters of the clinical course including modified Rodnan skin score (mRSS),
forced vital capacity (FVC), diffusion capacity (DLCO) and the course of pulmonary FAPI uptake
measurements (Supplementary Data File 4). In addition, the individual trends of mRSS are visualized
in main figure 1c as Spaghetti plots and the course of individual FVC changes (% predicted) and DLCO
changes (% predicted) are visualized in supplementary figure 3D.

As this analysis focusses on the description of molecular responses of the skin, mRSS is the central
clinical parameter in this context. The minimally important clinically difference for mRSS response is
commonly considered improvement of 5 units within 12 months in dcSSc patients as described based
on data from the Scleroderma lung study 1 and 2⁵², a definition commonly used to define response in
clinical trials^{53,54}. On this ground, all patients responded to the CD19-CAR T-cell treatment in this study.
Unfortunately, the sample size is currently too limited to define further subcategories such as “low” or
“high” response.

1199 g. CAR T-cell expansion and persistence: If data on CAR T-cell counts in blood (e.g., from flow
cytometry or PCR) are available, the authors should mention whether there was any correlation
between CAR T expansion/persistence and clinical outcomes. For instance, did patients with higher
CAR T-cell expansion or longer persistence tend to have greater skin score improvements? Such
correlations, if present, would support the mechanistic role of CAR T cells in driving the observed
effects.

The data on CD19-CAR T-cell expansion in the blood of the patients included in this study are shown in
Response letter Figure 6 below and in Supplementary Figure 1C. As suggested and given the
exploratory sample size of this study, we added a preliminary association CAR T-cell expansion
(maximum cell number at peak between d 7-9) with mRSS change between baseline and 12 months
follow-up: We observe a relatively weak negative association, which was not significant. Similarly, the
CAR-cell persistence (measure by d from first to last detection of CD19-CAR T-cells in the blood) did
not significantly associate with mRSS change. This is largely explained by the relatively homogenous
CAR T-expansion across the cohort as visualized below, making correlation/association analysis
difficult. We reason, this correlation analysis may not be an appropriate methodology to assess the
mechanistic implication of CD19-CAR T-cells and thus we had not included it in the a-priori analysis
plan.

The functionality of CD19-CAR T-cell therapy *in vivo* is proven by depletion of the peripheral B-cells and
the depletion of CD19⁺-cells in the lymph node, which was previously described [25,26]. In extension,
we here describe depletion of B-cells in the skin which lasts longer than B-cell depletion in the blood.
In addition, we investigated the effects of B-cell depletion on the skin tissue using a comprehensive set
of analyses including histology, cISH and IMC. The results of this exploratory study are hypothesis-
generating for further mechanistic validation studies, as discussed (p.16, ll. 400 ff.).

**Response letter Figure 6. A** expansion of CAR T-cells in the peripheral blood (also depicted in
 Supplementary Figure 1C). **B** Association of peak number of peripheral CAR T-cells with mRSS change
 from baseline to 12 month follow up. **C** Association of CAR T-cell persistence with mRSS change from
 baseline to 12 month follow up.

1227 h. Durability of response: The follow-up duration was up to 12 months for some patients (shorter for
 others). The manuscript should comment on the durability of the clinical and histological
 improvements. Have any patients experienced relapses or SSc flares after treatment within the
 follow-up period? Noting this (even qualitatively) would inform how sustained the benefits of CAR T
 therapy are, pending longer observation.

As described in response to your comment 2a, we added a file with consecutive clinical outcomes
 (Supplementary Data File 1_clinical data and sample overview_31012026) to visualize the individual
 course of mRSS and lung function parameters, which are also summarized in Figure 1c and
 supplementary figure 3D. In summary, we observed continuous improvement of mRSS and lung
 function throughout the described observational time. In line with the clinical response we observe
 sustained improvement of papillae measurements (Figure 1A), counts of rete ridges (Figure 5A), and
 vessel counts (Figure 4A/E). In addition, we observed the ongoing reduction of FAP+ fibroblasts in the
 skin (Figure 2B). During the revision, the respective existing quantifications shown in Figures 1A, 2B, 4
 A+E and 5 A were amended by additional follow samples acquired during the revision and included the
 6-and 12 months follow-up samples from patients 7, 8, 9, 10, 11. In addition to the figures, the
 measurements are specified in the source data file.

The respective quantifications now look as follows:

**Revised Figure 1A:** Papillae quantifications (papillae/mm and papillary height) per section as well as
 collagen alignment coefficient are shown as bar graphs. Median with interquartile range (IQR) are
 shown. p-values < 0.0125 were considered significant after Bonferroni correction. Baseline (n=18), CAR
 1249 T: 1-month follow-up (n=10), CAR T: 6-month follow-up (n=9), CAR T: 12-month follow-up (n=10),
 natural disease course group: follow up (n=1) and standard-of-care treatments: follow up (n=7).

**Revised Figure 2B:** Quantification of FAP+ fibroblasts as percent of total fibroblasts is shown as bar
 graph. Before CAR T (n=10), 1 mo-follow up (n=10), 6 mo-follow up (n=9), 12 mo-follow up (n=9),
 before start of standard of care therapy (SOC) (n=4), follow up SOC (n=5).

**Revised Figure 4E:** E Fold change of total vessel length per section and fold change of mean vessel
 length per section per section are shown as bar graphs. Measurements were normalized to the
 respective baseline sample. Baseline (n=18), CAR T: 1-month follow-up (n=10), 6-month follow-up
 (n=9), 12-month follow-up (n=10), natural disease course group (NDG, n=1) and standard of care (SOC,
 n=7).

**Revised Figure 5A:** Quantifications of rete ridges (rete ridges/mm) per section are shown as bar graphs.
 Median with interquartile range (IQR) are shown. P-values < 0.0125 were considered significant after
 Bonferroni correction. Baseline (n=18), 1-month follow-up (n=10), 6-month follow-up (n=9), 12-month
 follow-up (n=10), natural disease course group (n=1) and standard-of-care treatments (n=7).

We also amended a sentence to the discussion section, which now reads as follows (p.17, ll. 431-433):

*“The improvement of histological outcomes persisted during the time of study observation in*
 *line with clinical response as assessed by mRSS.”*

i. Transient inflammation and safety: The gene expression data showed that inflammatory pathways
 (e.g., interferon signaling) were elevated at 1 month post-CAR T but had normalized by 12 months.
 Could this transient early inflammation be related to CAR T-cell activity (for example, cytokine
 release during B-cell depletion)? Was it clinically significant? It would be helpful for the authors to
 note that, indeed, most patients experienced only mild cytokine release syndrome (CRS) after CAR T
 infusion. Published experiences with CAR T in autoimmune diseases (e.g., SLE) have also reported
 predominantly grade 1 CRS. Clarifying that any early inflammatory signal did not translate into
 serious clinical adverse events would reassure readers about the safety profile of this approach.

We thank the reviewer for his/her comment. We elaborated on transient inflammation and safety in
 the discussion section (p.19, ll. 483 ff.):

*“Interestingly, we observe an upregulation of terms associated with immune response*
 *regulation in fibroblasts, endothelial cells and keratinocytes early (1 month) after CD19-CAR*
 *T-cell therapy including “T-cell activation”, “interferon signaling” and “leukocyte*
 *chemotaxis”. At the time of 1-month follow up biopsy, B-cells were not detectable in the blood*
 *and skin tissue. Moreover, the peak expansion of CAR T cells had already subsided, and*
 *peripheral CAR T-cells were no longer detectable, making a local manifestation of cytokine*
 *release syndrome unlikely. We thus speculate that these transcriptomics changes may be a*
 *reflection of the recently described local immune effector cell-associated toxicity syndrome*
 *(LICATS)⁵⁵, which is characterized by local organ specific inflammatory symptoms that occur*
 *after CD19-CAR T-cell infusion and are usually transient and mild in their nature. In this*
 *cohort of SSc patients, a transient reddening of the skin was observed in one patient, so we*
 *assume the molecular findings occurred largely beyond the clinical detection threshold. The*
 *pathogenesis of LICATS is not fully defined, however, activation of autoreactive T cells that are*
 *reinfused as non-transduced T cells and subsequently expand alongside the CAR T cells or*
 *effects mediated by immune cells that repopulate after lymphodepletion, including—but not*
 *limited to—fludarabine-induced changes in myeloid cells and regulatory T cells may play a*
 *role. By twelve months, this activation of inflammatory responses in fibroblasts and endothelial*

*cells is no longer detectable, and in contrast, the respective signatures are even*
*downregulated.”*

j. Disease subsets and timing: All patients in this study had diffuse cutaneous SSc (most were Scl-70 or
RNA polymerase III autoantibody positive). The authors might discuss whether the outcomes could
differ in other subsets of SSc. For instance, would limited cutaneous SSc (centromere-positive disease)
respond similarly, or might it have a different trajectory? Also, might disease duration influence the
degree of reversibility (e.g., could long-established fibrosis be less reversible)? While the current study
may not have data to answer these questions, acknowledging them could highlight important
considerations for future applications of CAR T therapy in SSc.

We thank the reviewer for his/her comment. The inclusion criteria of the present study enrich for
patients with diffuse SSc with progressive and refractory disease being at risk for further progression.
None of the patients co-expressed anti-centromere antibodies. Thus, unfortunately, a data-informed
discussion on patients with limited SSc or with anti-centromere A/B positivity is not possible based on
the results of this study. If the authors may speculate on the responsiveness of patients with limited
SSc to CD19-CAR T cell therapy, the subgroup of lc SSc, short disease duration and positive anti-Scl70
or anti-RNAP III antibodies would share most similarities with the target population of this study,
however, future studies will need to clarify this research question.

As also described in response to comment 1, reviewer 2, we investigated samples of a patient with
progressive disease beyond seven years disease duration, who received CD19-CAR T-cell therapy as
named patient use with clinical and molecular response as further detailed above (p.10, ll.309 ff. of
this response letter). Thus, we speculate that fibrotic remodeling that has already been accumulating
over several years may not be fully reversible upon CD19-CAR T-cell therapy, at least not within the
current follow-up intervals. However, further progression even at an advanced disease stage can
potentially be stopped. Interestingly, we observed relative FVC improvement by 11% in this patient by
12 months after CD19-CAR T-cell therapy, which exceeded our expectations. As suggested by the
reviewer, we amended the discussion (p.22, ll. 550-557):

*“The patients investigated in this study had refractory, diffuse cutaneous SSc with a progressive*
*disease course despite multiple treatments. Interestingly, we observed a clinical and molecular*
*response in a patient with progressive disease after seven years of disease duration, which may*
*open avenues for further studies of the subgroup of patients who experience disease progression*
*at later disease stages. However, other subgroups of SSc patients such as early SSc, or limited*
*cutaneous SSc with organ involvement were not included in the target population of this study*
*and may be investigated in future trials. However, such patients can only be treated with CAR*
*T-cells if showing high-level safety.”*

3. Discussion

a. The Discussion section currently reads as largely a recap of the results, with relatively little
exploration of mechanisms or integration with existing literature. Rather than simply repeating the
findings, the authors should expand the discussion to interpret the results in depth and to propose
mechanistic hypotheses. This will better contextualize their findings within the broader understanding
of SSc pathophysiology and therapy.

We thank the reviewer for his/her comment. As also requested by reviewer 2, we revised the
discussion to emphasize the exploratory and hypothesis generating nature of the study (p. 17, l. 416-
p.24, l. 600 of the manuscript). Herein, we particularly expanded on potential mechanistic hypothesis
and integrated our findings more with the existing literature, e.g. current knowledge on immune
alteration upon hematopoietic stem cell transplantation in SSc and as further detailed below.

b. B-cell depletion and fibrosis reversal: A major point for discussion is that CD19 CAR T cells primarily
act by depleting B cells, yet the manuscript provides limited discussion on how B-cell removal leads
to fibrosis reversal. The authors should delve deeper into potential mechanisms. Do they think the
CAR T cells acted mainly by depleting autoreactive B cells in the circulation, or did they also eliminate
B cells within the affected tissues? (It is not clear how many B cells were present in skin biopsies
before and after treatment, and whether CAR T cells themselves trafficked to the skin – the authors
could clarify this if data are available.) The discussion should consider how B-cell depletion might
translate to antifibrotic effects. For example, does the removal of autoantibody-producing B cells and
plasmablasts alleviate chronic inflammatory signaling (such as interferon or TGF- β) in the tissue
microenvironment? The data showed a transient spike in interferon-responsive genes at 1 month
(perhaps due to initial immune activation by CAR T cells) that subsided by 12 months, consistent with
an early inflammatory response that later resolves. A more detailed discussion of these dynamics
would be valuable. The authors might hypothesize, for instance, that B cells produce cytokines and
growth factors that sustain myofibroblasts, and that their elimination creates a milieu permissive for
fibrosis resolution and tissue regeneration. Additionally, the observed increase in Schwann cells in
regenerating skin is intriguing – the authors could comment on the potential interplay between
nerve fibers and fibrosis (neurogenic factors in tissue repair) as part of the regeneration process.

As requested, we discussed the potential mechanisms of CD19-CAR T-cell mediated anti-fibrotic effects
and tissue regeneration based on the results of this study addressing all issues suggested by the
reviewer's comment. As also described in response to the comments of reviewers 1 and 3 (p.7, ll.224
ff. and p. 26, ll. 738 ff. of this response letter), we clarified the B-cell counts in the skin, which are now
clearer visualized in Supplementary Figure 2. Unfortunately, tracing CAR T-cells to the skin is technically
not possible at the current stage. We thoroughly discussed the potential mechanisms, by which CD19-
CAR T-cell mediated B-cell depletion translates into tissue remodeling in a data informed manner and
based on the data acquired by investigating fibroblasts, endothelial cells and epidermal cells and
speculate on the potential upstream mechanisms that may mediate these aspects. Moreover, we
indulged on the transient spike of interferon related signaling 1 month after CD19-CAR T-cell therapy
in the skin. We also discussed the aspect of nerve-fibroblast interaction (p.17, l. 435-p.21, l.534):

[revised manuscript text omitted]

c. The authors should further hypothesize how, mechanistically, B cells drive fibrosis and how their
depletion reverses it. Possible mechanisms to discuss include: autoantibody-mediated stimulation of
fibroblasts or endothelial cells; B cell production of pro-fibrotic cytokines (like IL-6, IL-13, or TGF- β) that
directly activate fibroblasts; and roles of B cells in modulating T cell or macrophage phenotypes toward
pro-fibrotic states. By framing the findings in terms of these known mechanisms, the discussion would
better link the B-cell depletion to the observed downstream tissue changes.

We thank the reviewer for his/her comment. As suggested, we discussed the potential mechanisms
of B-cell/fibroblast and B-cell/endothelial cell interactions, as skin vessels and fibroblasts were two
central aspects of this study and also B-cell/keratinocyte interaction (p.20, l.502-p21, l.534 (see
response to your previous comment) and p.19, l 467-481, respectively). The effect of B-cell depletion
on other inflammatory cell subsets has not been investigated in this study which focused on the
mesenchymal compartment and was thus added to the limitations section. The respective passages
now read as follows:

Discussion, p.22, ll. 599-566:

*“Our study has some limitations: while we provide detailed clinical, histological, proteomic,*
*and transcriptomic evidence for skin regeneration toward a physiologic phenotype—including*
*restoration of fibroblast, vascular, and epidermal compartments— a detailed analysis of the*
*immunologic mechanisms behind the observed tissue changes including effects of B-cells on*
*other immune cells has not been performed as part of this study. Furthermore, the question, in*
*how far B-cell depletion in the tissue versus systemic immunologic reprogramming mediate the*
*effects of CD19-CAR T cell treatment need to be addressed in future studies with longer follow*
*up and detailed investigation of immune cells given the persisting absence of B-cells in the skin*
*during the available observational time line.”*

1499 d. Immune reconstitution after B-cell depletion: The authors should also consider the fate of the
1500 immune system after B-cell removal. What “fills the void” after pathogenic B cells are eliminated? The
1501 authors might draw parallels to recent lupus CAR T trials, where B-cell aplasia lasted only a few months
and was followed by reemergence of naive B cells and a reset immune repertoire, without return of
disease. In SSc, it would be interesting to know if new B cells began to return by 12 months, and if so,
whether they appeared less autoreactive (though this may be beyond the scope of the current data).
The transient nature of CAR T-induced B-cell depletion (lasting on the order of 3–4 months in other
reports) suggests that the therapy provides an immune “reset” rather than permanent B-cell
eradication. The discussion could speculate on how the immune system might reconstitute in a
healthier balance post-CAR T, potentially with restored tolerance.

We thank the reviewer for his/her comment. Although the question of immune reconstitution is very
intriguing, it was not the focus of this study and the methodology was not primarily tailored to answer
the question of immune reconstitution across compartments. The studies cited in SLE refer to
peripheral blood, while the presented study here focusses on the skin. Based on our results, we may
postulate that immune reconstitution in the skin occurs with delay, as B-cells remain depleted
throughout the observational time period, while blood B-cells recover earlier. The peripheral B-cell
counts were added as requested and are depicted in supplementary 1B and are discussed as detailed
in response to your comment e, section Methods and Results (p.39., ll. 1138 ff. of this response letter).

A section on the investigation of the immune reset was added to the discussions section (p.22, ll. 559-
566):

*“Our study has some limitations: while we provide detailed clinical, histological, proteomic,
and transcriptomic evidence for skin regeneration toward a physiologic phenotype—including
restoration of fibroblast, vascular, and epidermal compartments— a detailed analysis of the
immunologic mechanisms behind the observed tissue changes including effects of B-cells on
other immune cells has not been performed as part of this study. Furthermore, to which extent
B-cell depletion in the tissue versus systemic immunologic reprogramming mediates the effects
of CD19-CAR T-cell treatment needs to be addressed in future studies with longer follow up
and detailed investigation of immune cells.”*

e. Comparison to HSCT: It would strengthen the discussion to compare CAR T therapy with autologous
hematopoietic stem cell transplantation (HSCT), which is the most intensive therapy currently used for
severe SSc. HSCT can induce remission of skin fibrosis and improve survival, but at the cost of high
treatment-related mortality and significant toxicity due to complete immune ablation. In contrast,
CD19 CAR T-cell therapy targets only B cells and thus far appears to have a much more favorable safety
profile (in this study and others in autoimmunity, there were no treatment-related deaths and only
mild CRS, etc.). The authors could point out that CAR T might achieve some of the immune “reset”
benefits of HSCT (halt of autoimmunity and fibrosis reversal) without the same risks, positioning CAR
T as a potentially safer alternative to HSCT in the future for patients with refractory SSc.

As requested, we added a section comparing CART-therapy with autologous hematopoietic stem cell
transplantation to the discussion. The respective section now reads as follows (p.21, l.529-p22, l.549):

*“These observations, together with the transient peripheral B-cell depletion, may argue for a
deep immune alteration resulting in regenerative tissue remodeling despite the persistence of
anti-Scl70 antibodies, yet with declining titers. Similar results on the persistence of anti-Scl70
antibodies were also observed upon hematopoietic stem cell transplantation, which is
considered a highly effective therapy in SSc: An analysis of the autoantibody repertoire of SSc-
patients treated within the SCOT trial showed transient reduction of the anti-Scl70 antibody
profile in two out of five Scl70 positive patients⁶⁰. Despite the persistence of anti-Scl70
antibodies, hematopoietic stem cell transplantation provides proof-of concept for the
elimination of autoreactive immune-cells resulting in the restoration of self-tolerance. Several
studies showed the downregulation of interferon-related signatures in the peripheral blood
upon HSCT, similar to our results on skin-related signatures, and the shift from pro-
inflammatory to tolerant immune phenotypes⁶⁰⁻⁶². In parallel, beneficial effects on skin tissue
remodeling with reduced skin thickness and collagen deposition were described^{50,63,64}. Overall,
AHSCT is an effective therapy for SSc that can reduce the rate of disease-related death,
however, it remains a complex procedure with treatment related risks⁶⁵. CD19 CAR T-cell
therapy also eliminates B cells deeply and appears to have a rather favorable safety profile
1555^{28,66,67}. Given its clinical effects and the profound tissue remodeling described in this study,
CAR T-cell therapy might achieve immune mediated tissue regeneration with aspects similar to
HSCT at a more favorable risk profile. Comparisons between the effects of CD19-CAR T-cell
therapy with AHSCT would be needed to further clarify this point.”*

f. In summary, the Discussion should be revised and expanded substantially. The authors need to go
beyond restating their results and engage with these mechanistic questions and comparisons to

existing literature. A more in-depth discussion, as outlined above, will better highlight the
significance of their findings and address potential concerns or curiosities a reader might have.

We thank the reviewer for his/her comment. As requested, we extensively revised and expanded the
manuscript. We discussed potential mechanisms of CD19-CAR T-cell-mediated effects on SSc based on
our data and formulated mechanistic hypotheses. We referred to existing literature relating to CD19-
CAR T-mediated effects, autologous hematopoietic stem cell transplantation and immune mechanisms
in SSc.

**References – Response letter**

[revised manuscript text omitted]

Author response to the comments of Reviewer 3

Reviewer #3 (Remarks to the Author):

The authors have addressed most of my concerns. I personally don't find the designation of "universal fibroblast" a useful term but accept that there is not yet a consensus on how to name the fibroblast populations in the skin.

We thank the reviewer for sharing this matter. We acknowledge that the term "universal fibroblast" is used among others. We de-emphasized the term "universal fibroblast" and referred to this population as PL16+ fibroblasts with a "universal fibroblast progenitor phenotype" (manuscript, p. 9, l 224).

I have a few minor comments.

In Supplementary figure 2 it is now clear that most of the patients' studied had 0 B cells in their skin at baseline. I don't believe it diminishes the impact of the observations, but it suggests that local tissue B cell depletion is not behind the observed clinical course. It also raises the question of whether the two patients with B cells responded any differently. I realize this is a small study, but it is relevant to understanding whether B cell depletion is working locally or systemically. Tissue B cell depletion in skin was seen also with rituximab without an evident clinical response (PMC2637937), suggesting that it is the deletion of autoantibodies or other systemic effect that is driving the therapeutic benefit.

We thank the reviewer for his/her comments and for highlighting the reference by Lafyatis et al [1](PMC2637937). In this series, the authors describe clinical and serological effects in 15 dcSSc patients as well as B-cell counts in the skin from baseline to six-months follow-up upon rituximab therapy. Clinically, no significant mRSS change was observed, and lung function parameters remained stable. Auto-antibody titers showed discrete and variable response. B-cell counts in the skin were limited before therapy and reduced in the skin after therapy. While the reviewer argues no clinical response was evident in terms of improvement, other experts may argue that no worsening occurred either, which would be expected in some patients of an at-risk population. However, the described clinical changes were milder compared to the mRSS and FVC improvements observed upon CD19-CAR T cell therapy [2](PMID: 41501497). Interestingly, CD19-CAR T cell therapy resulted in slow but constant reduction of systemic autoantibodies and in pronounced depletion of CD19+ cells in the lymph nodes, which was not observed upon rituximab. These observations in context with the modest number of B-cells in the skin before therapy raise the hypothesis that systemic effects may at least play a central role, while local effects of B-cells cannot be excluded. In response to the comments of previous reviewer four, we added a detailed discussion of potential local and systemic mechanisms of CD19-CAR T-cell induced tissue remodeling (manuscript p.17, ll.436 ff.). We further expanded this discussion in response to the current comment of reviewer 3 and the amended text now reads as follows (manuscript p.21, ll. 538 ff.):

"Both, local and systemic immunologic effects may contribute to the mediation of CD19-CAR T cell mediated tissue responses. While the reduction of systemic autoantibodies and depletion of CD19+ cells in the lymph nodes suggest systemic effects, the role of local mediators cannot fully be excluded. Further mechanistic studies are needed to clarify these contributions."

As acknowledged in the discussions section, while we performed clinical, histologic, proteomic and transcriptomic investigations including the analyses of mesenchymal and vessel structures, the detailed analyses of immunologic mechanisms were not part of the study (discussion section, manuscript p. 22, ll. 568 ff.).

The cluster of cells expressing PU.1/SPI1 are most likely macrophage-fibroblast doublets (see supplemental file 4 in Tabib T et al PMID: PMC8289865). This is strongly supported by panel 2C showing that the fibroblasts are not forming a discrete cluster. PU.1/SPI1 is a clean marker of macrophages and known to regulate their differentiation. S100A4 is expressed more diffusely by

both macrophages and all fibroblast subpopulations. The FLEX scRNA-seq technology applied to tissue is known to lead to larger populations of doublets, which explains the high number of macrophage doublets in the fibroblast cluster. This is a small point but will be confusing to macrophage experts. In my lab algorithms designed to remove these doublets (DoubletFinder, scDbfFinder etc) have unfortunately not enabled removal of these cells. So, my suggestion is to mention this in the text and to otherwise ignore these cells. For future studies, it may help to limit formalin fixation to overnight and then place biopsies in 70% ethanol and process for single cell studies as soon as feasible.

We thank the reviewer for bringing the issue of SPI1 expression in the named file to our attention. In supplementary file 4 in Tabib T et al, SPI1 is expressed across several cell populations with the most prominent expression in the cell cluster identified as macrophages. The SPI1 expression is described in different cell populations, including fibroblasts [3-5].

We also tested doublet removal in our cISH data using scDbfFinder without success, as reported by the reviewer. We thus cannot exclude the possibility that some these cells are fibroblast-macrophage doublets. However, we also detected a similar reduction of PU.1 expressing fibroblasts (CD45-negative population) upon CAR T cells in our IMC data. The results further suggest that PU.1+S100A4+ population detected in cISH is, at least in part, fibroblast. We further tuned down our interpretation of PU.1+S100A4+ fibroblasts by adding the following text in the Discussion section (p.18, ll. 444 ff.):

"Despite the previous scRNA-Seq study suggesting exclusive PU.1 expression in macrophages (Tabib T et al), IMC and cISH consistently detected reduction of PU.1+ fibroblasts. However, we could not exclude the possibility that the population was derived from fibroblast-macrophage doublets. Additional studies on fibroblast-macrophage interaction may further elucidate whether these signals reflect a genuine cellular phenotype or a close spatial association."

We appreciate the suggestion of preventing over-fixation, which has been considered when preparing the samples for the present study.

Although the authors indicate that Supplemental Figure 3 and 4 heatmaps have been removed, the bottom panel of figure 4G and supplemental Figure 5 are heatmaps that look similar and still lack a legend to show what color is up and what is down. If red is indicating higher expression, then it does seem odd that the collagen genes are upregulated after treatment. This appears particularly prominent in the 12-month samples. The authors explanation appears reasonable that fibroblasts numbers may be increasing during recovery. But if the authors are going to show these heatmaps they must provide the legends for interpretation even if it is not particularly supportive of the other data.

We thank the reviewer for his comment and apologize for the lacking legend, which was now included according to the data presented in the source data file.

References:

1. Lapyatis R, Kissin E, York M, Farina G, Viger K, Fritzler MJ, Merkel PA, Simms RW. *Arthritis Rheum.* 2009.10.1002/art.24249
2. Müller F, Hagen M, Wirsching A, Kharboutli S, Aigner M, Völkl S, Kretschmann S, Tascilar K, Taubmann J, Bucci L, Raimondo MG, Bergmann C, Rothe T, Corte G, Tur C, Muñoz L, Böltz S, Schuster L, Hartmann F, Garantziotis P, Spörl S, Vasova I, Gerbitz A, Spriewald B, Kiener H, Giannarelli D, Locatelli F, D'Agostino M-A, Hanssens L, Miltenyi S, Bozec A, Grieshaber-Bouyer R, Mackensen A, Schett G. *Nature Medicine.* 2026.10.1038/s41591-025-04185-6
3. Wohlfahrt T, Rauber S, Uebe S, Lubert M, Soare A, Ekici A, Weber S, Matei AE, Chen CW, Maier C, Karouzakis E, Kiener HP, Pachera E, Dees C, Beyer C, Daniel C, Gelse K, Kremer AE, Naschberger E, Stürzl M, Butter F, Sticherling M, Finotto S, Kreuter A, Kaplan MH, Jüngel A, Gay S, Nutt SL, Boykin DW, Poon GMK, Distler O, Schett G, Distler JHW, Ramming A. *Nature.* 2019.10.1038/s41586-019-0896-x
4. Tu J, Chen W, Fang Y, Han D, Chen Y, Jiang H, Tan X, Xu Z, Wu X, Wang H, Zhu X, Hong W, Li Z, Zhu C, Wang X, Wei W. *Ann Rheum Dis.* 2023.10.1136/ard-2022-222708
5. Zhang M, Zhang J, Hu H, Zhou Y, Lin Z, Jing H, Sun B. *J Transl Med.* 2024.10.1186/s12967-024-05398-y

The authors have addressed most of my concerns. I personally don't find the designation of "universal fibroblast" a useful term but accept that there is not yet a consensus on how to name the fibroblast populations in the skin.

I have a few minor comments.

In Supplementary figure 2 it is now clear that most of the patients' studied had 0 B cells in their skin at baseline. I don't believe it diminishes the impact of the observations, but it suggests that local tissue B cell depletion is not behind the observed clinical course. It also raises the question of whether the two patients with B cells responded any differently. I realize this is a small study, but it is relevant to understanding whether B cell depletion is working locally or systemically. Tissue B cell depletion in skin was seen also with rituximab without an evident clinical response (PMC2637937), suggesting that it is the deletion of autoantibodies or other systemic effect that is driving the therapeutic benefit.

The cluster of cells expressing PU.1/SPI1 are most likely macrophage-fibroblast doublets (see below from supplemental file 4 in Tabib T et al PMID: PMC8289865). This is strongly supported by panel 2C showing that the fibroblasts are not forming a discrete cluster. PU.1/SPI1 is a clean marker of macrophages and known to regulate their differentiation. S100A4 is expressed more diffusely by both macrophages and all fibroblast subpopulations. The FLEX scRNA-seq technology applied to tissue is known to lead to larger populations of doublets, which explains the high number of macrophage doublets in the fibroblast cluster. This is a small point but will be confusing to macrophage experts. In my lab algorithms designed to remove these doublets (DoubletFinder, scDblFinder etc) have

unfortunately not in our hands enabled removal of these cells. So, my suggestion is to mention this in the text and to otherwise ignore these cells. For future studies, it may help to limit formalin fixation to overnight and then place biopsies in 70% ethanol and process for single cell studies as soon as feasible.

Although the authors indicate that Supplemental Figure 3 and 4 heatmaps have been removed, the bottom panel of figure 4G and supplemental Figure 5 are heatmaps that look similar and still lack a legend to show what color is up and what is down. If red is indicating higher expression, then it does seem odd that the collagen genes are upregulated after treatment. This appears particularly prominent in the 12-month samples. The authors explanation appears reasonable that fibroblasts numbers may be increasing during recovery. But if the authors are going to show these heatmaps they must provide the legends for interpretation even if it is not particularly supportive of the other data.